# Equivariant Descriptor Fields: SE(3)-Equivariant Energy-Based Models for End-to-End Visual Robotic Manipulation Learning

**Hyunwoo Ryu**[1]    **Hong-in Lee**[1]    **Jeong-Hoon Lee**[2,3]    **Jongeun Choi**[1,3]
[1]Department of Artificial Intelligence, Yonsei University
[2]Samsung Research    [3]School of Mechanical Engineering, Yonsei University
`{tomato1mule,theorist17,jongeunchoi}@yonsei.ac.kr`
`jh_0921.lee@samsung.com`

## Abstract

End-to-end learning for visual robotic manipulation is known to suffer from sample inefficiency, requiring large numbers of demonstrations. The spatial roto-translation equivariance, or the $SE(3)$-*equivariance* can be exploited to improve the sample efficiency for learning robotic manipulation. In this paper, we present $SE(3)$-equivariant models for visual robotic manipulation from point clouds that can be trained fully end-to-end. By utilizing the representation theory of the Lie group, we construct novel $SE(3)$-equivariant energy-based models that allow highly sample efficient end-to-end learning. We show that our models can learn from scratch without prior knowledge and yet are highly sample efficient ($5\sim10$ demonstrations are enough). Furthermore, we show that our models can generalize to tasks with (i) previously unseen target object poses, (ii) previously unseen target object instances of the category, and (iii) previously unseen visual distractors. We experiment with 6-DoF robotic manipulation tasks to validate our models' sample efficiency and generalizability. Codes are available at: `https://github.com/tomato1mule/edf`

## 1 INTRODUCTION

Learning robotic manipulation from scratch often involves learning from mistakes, making real-world applications highly impractical (Kalashnikov et al., 2018; Levine et al., 2016; Lee & Choi, 2022). Learning from demonstration (LfD) methods (Ravichandar et al., 2020; Argall et al., 2009) are advantageous because they do not involve trial and error, but expert demonstrations are often rare and expensive to collect. Therefore, auxiliary pipelines such as pose estimation (Zeng et al., 2017; Deng et al., 2020), segmentation (Simeonov et al., 2021), or pre-trained object representations (Florence et al., 2018; Kulkarni et al., 2019) are commonly used to improve data efficiency. However, collecting sufficient data for training such pipelines is often burdensome or unavailable in practice.

Recently, roto-translation equivariance has been explored for sample-efficient robotic manipulation learning. Transporter Networks (Zeng et al., 2020) achieve high sample efficiency in end-to-end visual robotic manipulation learning by exploiting $SE(2)$-*equivariance* (planar roto-translation equivariance). However, the efficiency of Transporter Networks is limited to planar tasks due to the lack of the full $SE(3)$-*equivariance* (spatial roto-translation equivariance). In contrast, Neural Descriptor Fields (NDFs) (Simeonov et al., 2021) can achieve few-shot level sample efficiency in learning highly spatial tasks by exploiting the $SE(3)$-equivariance. Moreover, the trained NDFs can generalize to previously unseen object instances (in the same category) in unseen poses. However, unlike Transporter Networks, NDFs cannot be end-to-end trained from demonstrations. The neural networks of NDFs need to be pre-trained with auxiliary self-supervised learning tasks and cannot be fine-tuned for the demonstrated tasks. Furthermore, NDFs can only be used for well-segmented point cloud inputs and fixed placement targets. These limitations make it difficult to apply NDFs when 1) no public dataset is available for pre-training on the specific target object category, 2) when well-segmented object point clouds cannot be expected, or when 3) the placement target is not fixed.

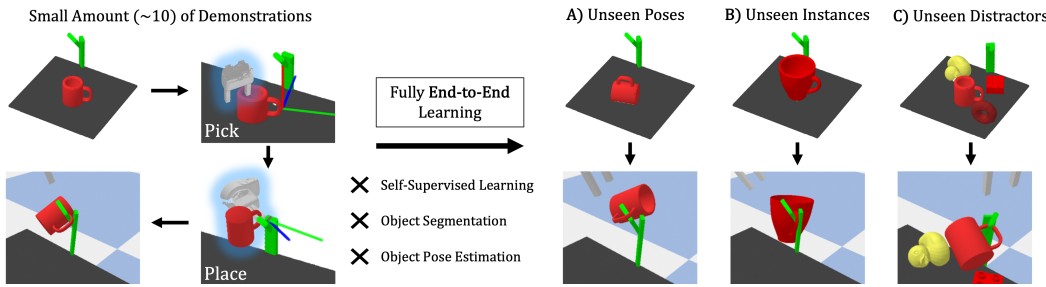

Figure 1: Given few (5∼10) demonstrations of a mug pick-and-place task, EDFs can be trained fully end-to-end without requiring any pre-training, object segmentation, or pose estimation pipelines. In addition, we show that EDFs can generalize to A) unseen poses, B) unseen instances of the target object category, and C) the presence of unseen visual distractors.

To overcome such limitations, we present *Equivariant Descriptor Fields* (EDFs), the first end-to-end trainable and $SE(3)$-equivariant visual robotic manipulation models. EDFs can be fully end-to-end trained to solve highly spatial tasks from only a few (5∼10) demonstrations without requiring any pre-training, object keypoint annotation, or segmentation. EDFs can generalize to previously unseen target object instances in unseen poses as NDFs. Furthermore, EDFs can generalize to unseen distracting objects and unseen placement poses (See Figure 1). Our contributions are as follows:

1. To enable end-to-end training, we reformulate the energy minimization problem of NDFs into a probabilistic learning framework with energy-based models on the $SE(3)$-manifold.

2. We generalize the *invariant* descriptors of NDFs into representation-theoretic *equivariant* descriptors. Using equivariant descriptors significantly improves generalizability owing to their orientational sensitivity.

3. We propose a novel energy function and end-to-end trainable query point models to achieve the $SE(3)$-equivariance regarding both the target object and placement target.

4. EDFs do not resort to non-local mechanisms to achieve the $SE(3)$-equivariance. This specific design enables our method to work well without object segmentation pipelines.

## 2 BACKGROUND AND RELATED WORKS

**Equivariant Robotic Manipulation**   Equivariant models have emerged as a promising approach for robotic manipulation learning, with growing evidence indicating they can significantly improve both sample efficiency and generalizability (Wang & Walters, 2022; Wang et al., 2022). Transporter Networks and their variants (Zeng et al., 2020; Seita et al., 2021) are end-to-end models for visual robotic manipulation tasks that exploit the planar roto-translation equivariance, or the $SE(2)$-equivariance for the sample efficiency. Equivariant Transporter Networks (ETNs) (Huang et al., 2022) exploit the representation theory of discrete rotation groups to further improve the sample efficiency. However, the efficiency of $SE(2)$-equivariant models is limited to planar tasks and cannot be extended to highly spatial tasks. Neural Descriptor Fields (NDFs) (Simeonov et al., 2021) overcome this limitation by leveraging the spatial roto-translation equivariance, or the $SE(3)$-equivariance.

**Energy-Based Models**   Energy-based models (EBMs) are probabilistic models that are derived from energy functions. EBMs are widely used for image and video generation (Zhu & Mumford, 1998; Xie et al., 2016; 2017; Du & Mordatch, 2019), 3D geometry generation (Xie et al., 2018b; 2021a), internal learning (Zheng et al., 2021), and control (Xu et al., 2022; Florence et al., 2022). Due to the intractability of the integral in the denominator of EBMs, Markov chain Monte Carlo (MCMC) methods are commonly used to estimate the gradient of the log-denominator to maximize the log-likelihood (Hinton, 2002; Carreira-Perpinan & Hinton, 2005). The *Metropolis-Hastings algorithm* (MH) (Hastings, 1970) and the *Langevin dynamics* (Langevin, 1908; Welling & Teh, 2011) are widely used MCMC methods for EBMs on Euclidean spaces. However, typical Langevin dynamics cannot be used for non-Euclidean manifolds such as the $SE(3)$ manifold. The Langevin dynamics on the $SE(3)$ group and general Lie groups are studied by Brockett (1997); Chirikjian

(2011); Davidchack et al. (2017). The Langevin dynamics and their convergence on general Riemannian manifold have been studied by Girolami & Calderhead (2011); Gatmiry & Vempala (2022). In this paper, we propose $SE(3)$-equivariant EBMs on the $SE(3)$ manifold, which should be distinguished from $SE(3)$-equivariant EBMs on Euclidean spaces (Jaini et al., 2021; Wu et al., 2021).

**Representation Theory of Lie Groups** A *representation* $\mathbf{D}$ of a group $G$ is a map from $G$ to the space of linear operators acting on a vector space $\mathcal{V}$ that has the following property:

$$\mathbf{D}(g)\mathbf{D}(h) = \mathbf{D}(gh) \quad \forall g, h \in G \tag{1}$$

Any representation of $SO(3)$ group $\mathbf{D}(\mathbf{R})$ for $\mathbf{R} \in SO(3)$ can be block-diagonalized into the direct sum of *(real) Wigner D-matrices* $\mathbf{D}_l(\mathbf{R}) \in \mathbb{R}^{(2l+1) \times (2l+1)}$ of *degree* $l \in \{0, 1, 2, \cdots\}$, which are orthogonal matrices (Aubert, 2013). The $(2l+1)$ dimensional vectors that are transformed by $\mathbf{D}_l(\mathbf{R})$ are called *type-l* (or *spin-l*) vectors. Type-0 vectors are invariant to rotations such that $\mathbf{D}(\mathbf{R}) = \mathbf{I}$. Type-1 vectors are the familiar 3-dimensional space vectors with $\mathbf{D}(\mathbf{R}) = \mathbf{R}$. Type-$l$ vectors are identical to themselves when rotated by $\theta = 2\pi/l$.

A type-$l$ vector field $\mathbf{f} : \mathbb{R}^3 \times \mathcal{X} \to \mathbb{R}^{2l+1}$ is $SE(3)$-*equivariant* if

$$\mathbf{D}_l(\mathbf{R})\mathbf{f}(\mathbf{x}|X) = \mathbf{f}(T\mathbf{x}|T \circ X) \quad \forall T = (\mathbf{R}, \mathbf{v}) \in SE(3), X \in \mathcal{X}, \mathbf{x} \in \mathbb{R}^3 \tag{2}$$

where $\mathcal{X}$ is some set equipped with a group action $\circ$ and $T\mathbf{x} = \mathbf{R}\mathbf{x} + \mathbf{v}$. *Tensor Field Networks* (TFNs) (Thomas et al., 2018) and $SE(3)$-*Transformers* (Fuchs et al., 2020) are used to implement $SE(3)$-equivariant vector fields in this work. We provide details of these networks in Appendix G.

## 3 PROBLEM FORMULATION

Let a colored point cloud with $M$ points given by $X = \{(\mathbf{x}_1, \mathbf{c}_1), \cdots, (\mathbf{x}_M, \mathbf{c}_M)\} \in \mathcal{P}$ where $\mathbf{x}_i \in \mathbb{R}^3$ is the position, $\mathbf{c}_i \in \mathbb{R}^3$ is the color vector of the $i$-th point, and $\mathcal{P}$ is the set of all possible colored point clouds. The action of $SE(3)$ on point clouds $\circ : SE(3) \times \mathcal{P} \to \mathcal{P}$ is then defined as

$$T \circ X = \{(T\mathbf{x}_1, \mathbf{c}_1), (T\mathbf{x}_2, \mathbf{c}_2), \cdots, (T\mathbf{x}_M, \mathbf{c}_M)\} \quad \forall T = (\mathbf{R}, \mathbf{v}) \in SE(3) \tag{3}$$

Consider a problem where a robot has to learn to grasp and place objects in specific locations from human demonstrations. To solve this problem, the placement pose $T \in SE(3)$ must be inferred such that the relative pose between the grasped object and placement target remains consistent with the demonstrations, regardless of changes in their postures. This can be achieved by requiring the *bi-equivariance* to $T$ such that 1) it follows changes in the posture of the placement target, and 2) it compensates for changes in the posture of the grasp (See Appendix C for more explanation). In contrast, uni-equivariant methods such as NDFs, which can only account for changes in one object, are not well-suited for problems where both postures may change.

Now let $X$ be the point cloud of the scene (where the placement target belongs), and $Y$ be the point cloud of the end-effector (with the grasped object) observed in the end-effector frame $T \in SE(3)$. The formal definition of bi-equivariance is as follows:

**Definition 1.** *A differential probability distribution $dP(T|X, Y)$ on $SE(3)$ conditioned by two point clouds $X, Y \in \mathcal{P}$ is bi-equivariant if for all Borel subsets $\Omega \subseteq SE(3)$,*

$$\int_{T \in \Omega} dP(T|X, Y) = \int_{T \in S\Omega} dP(T|S \circ X, Y) = \int_{T \in \Omega S} dP(T|X, S^{-1} \circ Y) \quad \forall S \in SE(3) \tag{4}$$

*where $S\Omega = \{ST | T \in \Omega\}$, $\Omega S = \{TS | T \in \Omega\}$, and $S^{-1}$ denotes the group inverse of $S$.*

**Definition 2.** *A scalar function $f : SE(3) \times \mathcal{P} \times \mathcal{P} \to \mathbb{R}$ is bi-equivariant if*

$$f(T|X, Y) = f(ST|S \circ X, Y) = f(TS|X, S^{-1} \circ Y) \quad \forall S \in SE(3) \tag{5}$$

**Proposition 1.** *A probability distribution $P(T|X, Y)dT$ is bi-equivariant if $dT$ is the bi-invariant volume form (See Appendix A) on the $SE(3)$ manifold and $P(T|X, Y)$ is a bi-equivariant probability density function (PDF).*

The proof of Proposition 1 can be found in Appendix F.1. The bi-equivariance condition of Definition 1 can be viewed as a generalization of Huang et al. (2022) to non-commutative groups like $SE(3)$ and a probabilistic generalization of Ganea et al. (2021). By Proposition 1, our goal boils down to constructing a bi-equivariant PDF $P(T|X, Y)$ such that

$$P(T|X, Y) = P(ST|S \circ X, Y) = P(TS|X, S^{-1} \circ Y) \quad \forall S \in SE(3) \tag{6}$$

A) Global Equivariance

B) Local Equivariance

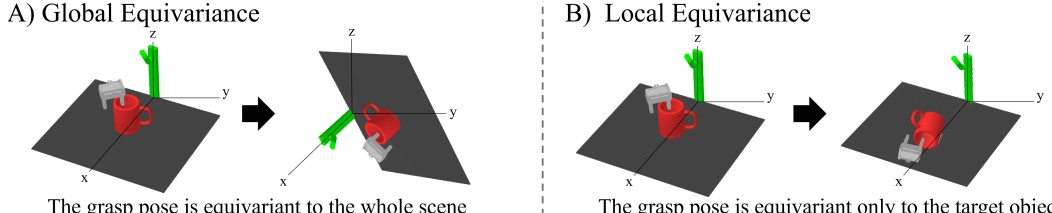

The grasp pose is equivariant to the whole scene

The grasp pose is equivariant only to the target object

Figure 2: A) The model is globally equivariant if the grasp pose is equivariant to the transformations of the whole scene (the target object and background). B) The model is locally equivariant to the target object if the grasp pose is equivariant to the localized transformations of the target object.

However, we want our policy to be not only *globally equivariant* as Equation (6) but also to be *locally equivariant*. That is, we want our models to be equivariant only to the target object, not the backgrounds. We illustrate the local equivariance and the global equivariance in Figure 2. To achieve the local equivariance, the model should only rely on *local mechanisms* to achieve the equivariance. For example, Transporter Networks achieve translational equivariance by using convolutional neural networks, which are well-known for their locality (Battaglia et al., 2018; Goodfellow et al., 2016). On the other hand, NDFs (Simeonov et al., 2021) rely on the centroid subtraction method to obtain translational equivariance, which is highly nonlocal. As a result, unlike Transporter Networks, NDFs cannot be used without object segmentation pipelines.

## 4    BI-EQUIVARIANT ENERGY BASED MODELS ON $SE(3)$

In this section, we present EDFs and the corresponding bi-equivariant energy-based models on $SE(3)$. We also provide practical implementations for the proposed models. We illustrate the overview of our method in Figure 3.

### 4.1    EQUIVARIANT DESCRIPTOR FIELD

We define the EDF $\boldsymbol{\varphi}(\mathbf{x}|X)$ as a direct sum of $N$ vector fields

$$\boldsymbol{\varphi}(\mathbf{x}|X) = \bigoplus_{n=1}^{N} \boldsymbol{\varphi}^{(n)}(\mathbf{x}|X) \tag{7}$$

where $\boldsymbol{\varphi}^{(n)}(\mathbf{x}|X) : \mathbb{R}^3 \times \mathcal{P} \to \mathbb{R}^{2l_n+1}$ is an $SE(3)$-equivariant type-$l_n$ vector field. Therefore, the EDF $\boldsymbol{\varphi}(\mathbf{x}|X)$ transforms according to a rigid body transformation $T \in SE(3)$ as

$$\boldsymbol{\varphi}(T\mathbf{x}|T \circ X) = \mathbf{D}(\mathbf{R})\boldsymbol{\varphi}(\mathbf{x}|X) \quad \forall\, T = (\mathbf{R}, \mathbf{v}) \in SE(3) \tag{8}$$

where $\mathbf{D}(\mathbf{R}) = \bigoplus_{n=1}^{N} \mathbf{D}_{l_n}(\mathbf{R})$ is the direct sum of the real Wigner D-Matrices of degree $l_n$ in the real basis. Therefore, $\mathbf{D}(\mathbf{R})$ is an orthogonal representation of the $SO(3)$ group.

Note that NDFs (Simeonov et al., 2021) only use type-0 descriptors, which are invariant to rotations such that $\mathbf{D}(\mathbf{R}) = \mathbf{I}$. In contrast, EDFs also use type-1 or higher descriptors, which are highly sensitive to rotations. As a result, NDFs require at least three non-collinear query points (and much more in practice) to represent the orientation of a rigid body, whereas EDFs require only one point.

### 4.2    EQUIVARIANT ENERGY-BASED MODEL ON $SE(3)$

Naively minimizing the energy function like Simeonov et al. (2021) cannot be used to simultaneously train the descriptors, as this would result in all the descriptors collapsing to zero (or some other constants). Therefore, we use the EBM approach for the end-to-end training of descriptors.

An energy-based model on the $SE(3)$ manifold conditioned by $X, Y \in \mathcal{P}$ can be defined as

$$P(T|X,Y) = \frac{\exp\left[-E(T|X,Y)\right]}{\int_{SE(3)} dT \exp\left[-E(T|X,Y)\right]} \tag{9}$$

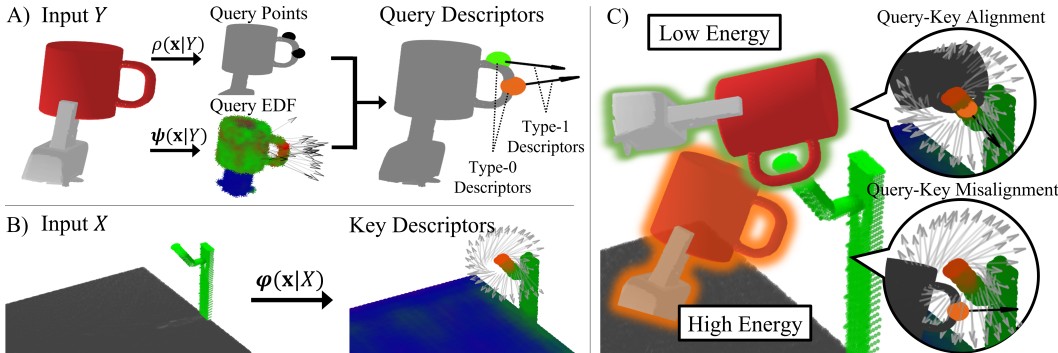

Figure 3: A) Query points and query EDF are generated from the point cloud of the grasp. Query EDF values at the query points are used as the query descriptors. We visualized three type-0 descriptors in colors (RGB) and type-1 descriptors as arrows. We only visualized type-1 descriptors in important locations. We did not visualize higher-type descriptors. B) The key descriptors are generated from the point cloud of the scene. C) The query descriptors are transformed and matched to the key descriptors to produce the energy of the pose. For simplicity, we only visualized the query descriptor for a single query point. Note that the query and key descriptors are better aligned in the low energy case than in the high energy case for both the type-0 and type-1 descriptors (The orange query points are near the orange region, and the black arrow is well aligned to the gray arrows).

**Proposition 2.** *The EBM $P(T|X, Y)$ in Equation (9) is bi-equivariant if the energy function $E(T|X, Y)$ is bi-equivariant.*

We prove Proposition 2 in Appendix F.2. We now propose the following energy function:

$$E(T|X, Y) = \int_{\mathbb{R}^3} d^3\mathbf{x}\, \rho(\mathbf{x}|Y) \|\boldsymbol{\varphi}(T\mathbf{x}|X) - \mathbf{D}(\mathbf{R})\boldsymbol{\psi}(\mathbf{x}|Y)\|^2 \tag{10}$$

where $\boldsymbol{\varphi}(\mathbf{x}|X)$ is the *key EDF*, $\boldsymbol{\psi}(\mathbf{x}|Y)$ is the *query EDF*, and $\rho(\mathbf{x}|Y)$ is the *query density*. Note that $T = (\mathbf{R}, \mathbf{v})$. The query density is an $SE(3)$-equivariant non-negative scalar field such that

$$\rho(\mathbf{x}|Y) = \rho(T\mathbf{x}|T \circ Y) \quad \forall T \in SE(3) \tag{11}$$

Intuitively, the energy function in Equation (10) can be thought as a query-key matching between the key EDF and the query EDF which is analogous to (Zeng et al., 2020; Huang et al., 2022).

**Proposition 3.** *The energy function $E(T|X, Y)$ in Equation (10) is bi-equivariant.*

We prove Proposition 3 in Appendix F.3. As a result, the EBM in Equation (9) with the energy function in Equation (10) is also bi-equivariant. Lastly, we provide an important consequence of Equation (11), whose proof can be found in Appendix F.4:

**Proposition 4.** *Non-constant query densities that satisfy Equation (11) must be grasp-dependent.*

### 4.3 IMPLEMENTATION

Our method consists of two models, viz. the *pick-model* and the *place-model*. The pick-model is a simplified version of the place-model. Therefore, we only demonstrate here the components of the place-model. The pick-model is demonstrated in Appendix E. We show in Appendix E that the energy function used in Simeonov et al. (2021) is a special case of our pick-model's energy function. For the following sections, we denote all the learnable parameters as $\boldsymbol{\theta}$. Therefore, all the functions with $\boldsymbol{\theta}$ as a subscript are to be understood as trainable models. We visualized the key EDF of a trained pick-model in Figure 4.

**Query Density** To make the integral in Equation (10) tractable, we model the query density as weighted query points by taking the weighted sum of Dirac delta functions $\delta^{(3)}(\mathbf{x}) = \prod_{i=1}^{3} \delta(x_i)$

$$\rho_{\boldsymbol{\theta}}(\mathbf{x}|Y) = \sum_{i=1}^{N_q} w_{\boldsymbol{\theta}}\left(\mathbf{q}_{i;\boldsymbol{\theta}}(Y)|Y\right) \delta^{(3)}\left(\mathbf{x} - \mathbf{q}_{i;\boldsymbol{\theta}}(Y)\right) \tag{12}$$

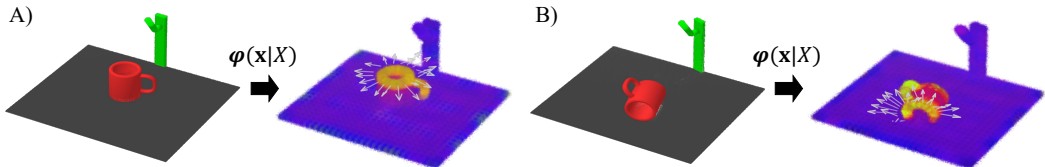

Figure 4: The key EDF of a trained pick-model is illustrated for the scenes with a mug in A) upright pose and B) lying pose. Note that the colors (type-0 descriptors) are invariant to the rotation of the mug. On the other hand, the arrows (type-1 descriptors) are equivariant to the rotation. We only visualized type-1 descriptors in important locations. Higher-type descriptors are not visualized.

where $\mathbf{q}_{i;\boldsymbol{\theta}}(Y) : \mathcal{P} \to \mathbb{R}^3$ is the $i$-th *query point function* and $w_{\boldsymbol{\theta}}(\mathbf{x}|Y) : \mathbb{R}^3 \times \mathcal{P} \to \mathbb{R}^+$ is the *query weight field*. These maps are $SE(3)$-equivariant such that

$$\mathbf{q}_{i;\boldsymbol{\theta}}(T \circ Y) = T\mathbf{q}_{i;\boldsymbol{\theta}}(Y)$$
$$w_{\boldsymbol{\theta}}(T\mathbf{x}|T \circ Y) = w_{\boldsymbol{\theta}}(\mathbf{x}|Y) \tag{13}$$

**Proposition 5.** *The query density $\rho_{\boldsymbol{\theta}}(\mathbf{x}|Y)$ in Equation (12) is $SE(3)$-equivariant.*

We prove Proposition 5 in Appendix F.5. Note that as a consequence of Proposition 4, the grasp dependence is inevitable for non-constant query point models like Equation (12). In this case, the integral in Equation (10) can be written in the following tractable summation form:

$$E_{\boldsymbol{\theta}}(T|X,Y) = \sum_{i=1}^{N_q} \widetilde{E}_{\boldsymbol{\theta}} \left(T|X,Y, w_{\boldsymbol{\theta}}\left(\mathbf{q}_{i;\theta}(Y)|Y\right), \mathbf{q}_{i;\theta}(Y)\right) \tag{14}$$

$$\widetilde{E}_{\boldsymbol{\theta}}(T|X,Y,w,\mathbf{q}) = w\|\boldsymbol{\varphi}_{\boldsymbol{\theta}}(T\mathbf{q}|X) - \mathbf{D}(\mathbf{R})\boldsymbol{\psi}_{\boldsymbol{\theta}}(\mathbf{q}|Y)\|^2 \tag{15}$$

In Appendix B, we provide practical implementations of $\mathbf{q}_{i;\boldsymbol{\theta}}(Y)$ and $w_{\boldsymbol{\theta}}(\mathbf{x}|Y)$ that are continuously parameterized and differentiable.

**EDFs** As was argued in Section 3, only the local operations should be used in our models for the local equivariance. We use Tensor Field Networks (TFNs) (Thomas et al., 2018) for the last layer and $SE(3)$-Transformers (Fuchs et al., 2020) for the other layers. The convolution operations that are used in these networks are highly local when their radial functions (See Appendix G) have short cutoff distances. We used simple radius clustering, which is locally $SE(3)$-equivariant within the clustering radius, to make the point clouds into graphs. For computational efficiency, only the last layer is used to evaluate the field values at the query points. All the other layers' outputs only depend on the point cloud and not the query points. Therefore, during the MCMC steps, only the last layer (TFN) has to be recalculated, and the outputs of the other layers ($SE(3)$-Transformers) can be reused. We use the E3NN package (Geiger et al., 2022) to implement the equivariant layers.

## 5 SAMPLING AND TRAINING

For the sampling, we first run the Metropolis-Hastings algorithm (MH) with $\mathcal{IG}_{SO(3)}$ distribution (Nikolayev & Savyolov, 1970; Savyolova, 1994; Leach et al., 2022) for the orientation proposal and typical Gaussian distribution for the translation proposal. Next, we run the Langevin dynamics on the $SE(3)$ manifold using the samples from MH as initial seeds. The Lie derivatives (Brockett, 1997; Chirikjian, 2011) for the Langevin dynamics are calculated in quaternion-translation parameterization as Davidchack et al. (2017) to avoid singularity issues. We provide details in Appendix D.

For the training, we estimate the gradient of the log-likelihood of Equation (9) at $T_{target}$ as

$$\nabla_{\boldsymbol{\theta}} \log P_{\boldsymbol{\theta}}(T_{target}|X,Y) \approx -\nabla_{\boldsymbol{\theta}} E_{\boldsymbol{\theta}}(T_{target}|X,Y) + \frac{1}{N}\sum_{n=1}^{N} [\nabla_{\boldsymbol{\theta}} E_{\boldsymbol{\theta}}(T_n|X,Y)] \tag{16}$$

where $T_n \sim P(T|X,Y)$ is the $n$-th negative sample (Carreira-Perpinan & Hinton, 2005). However, naively maximizing the log-likelihood is highly unstable. If the query EDF and the key EDF are

initially very different at some important sites, the learning algorithm tends to lower the query density of these sites rather than make the two EDFs closer. Therefore, all the query points in essential locations (such as contact points) are being pushed away. As a result, the training diverges.

To avoid this instability, we propose using the following surrogate query model during the early stage of training. We first decompose the EBM $P(T|X,Y)$ in Equation (14) into

$$P(T|X,Y) = \int d\mathbf{w} \int d\mathbf{Q} P(T|X,Y,\mathbf{w},\mathbf{Q}) P(\mathbf{w},\mathbf{Q}|Y) \tag{17}$$

$$P(T|X,Y,\mathbf{w},\mathbf{Q}) = \frac{\exp\left[-\sum_{i=1}^{N_q} \widetilde{E}(T|X,Y,w_i,\mathbf{q}_i)\right]}{\int_{SE(3)} dT \exp\left[-\sum_{i=1}^{N_q} \widetilde{E}(T|X,Y,w_i,\mathbf{q}_i)\right]} \tag{18}$$

$$P(\mathbf{w},\mathbf{Q}|Y) = \prod_{i=1}^{N_q} P_i(w_i,\mathbf{q}_i|Y) = \prod_{i=1}^{N_q} \left[\delta(w_i - w(\mathbf{q}_i|Y)) \times \delta^{(3)}(\mathbf{q}_i - \mathbf{q}_i(Y))\right] \tag{19}$$

where $\mathbf{Q} = (\mathbf{q}_1, \cdots, \mathbf{q}_{N_q})$ and $\mathbf{w} = (w_1, \cdots, w_{N_q})$. We temporarily hide $\boldsymbol{\theta}$ for brevity.

**Proposition 6.** *The marginal EBM $P(T|X,Y)$ in Equation (17) is bi-equivariant if*

$$P(\mathbf{w},\mathbf{Q}|Y) = P(\mathbf{w}, S\mathbf{Q}|S \circ Y) \ \ \forall S \in SE(3)$$

We prove Proposition 6 in Appendix F.6. We now relax this deterministic query model into a stochastic model by adding Gaussian noise to the logits of the query weights $l_i = \log w_i$ as follows.

$$\hat{P}(\mathbf{w},\mathbf{Q}|Y) = \prod_{i=1}^{N_q} \hat{P}_i(w_i,\mathbf{q}_i|Y) = \prod_{i=1}^{N_q} \frac{dl_i}{dw_i} \mathcal{N}(l_i; \log w(\mathbf{q}_i|Y), \sigma_H) \delta^{(3)}(\mathbf{q}_i - \mathbf{q}_i(Y)) \tag{20}$$

Now we propose the following surrogate query model

$$
\begin{aligned}
H(\mathbf{w},\mathbf{Q}|X,Y,T) &= \prod_{i=1}^{N_q} H_i(w_i,\mathbf{q}_i|X,Y,T) \\
H_i(w_i,\mathbf{q}_i|X,Y,T) &= \begin{cases} \hat{P}_i(w_i,\mathbf{q}_i|Y) & \text{if } d_{min}(T\mathbf{q}_i,X) < r \\ (dl_i/dw_i)\mathcal{N}(l_i;\alpha,\sigma_H)\delta^{(3)}(\mathbf{q}_i - \mathbf{q}_i(Y)) & \text{else} \end{cases}
\end{aligned} \tag{21}
$$

where $\sigma_H \in \mathbb{R}^+$, $r \in \mathbb{R}^+$, and $\alpha \in \mathbb{R}$ are hyperparameters and $d_{min}(\mathbf{x},X) : \mathbb{R}^3 \times \mathcal{P} \to \mathbb{R}^+$ is the shortest Euclidean distance between $\mathbf{x}$ and the points in $X$. We set $\alpha$ to be sufficiently small so that query points without neighboring points in $X$ can be suppressed.

To train our models using the surrogate query model in Equation (21), we maximize the following variational lower bound (Kingma & Welling, 2013) instead of the marginal log-likelihood.

$$\mathcal{L}_{\boldsymbol{\theta}}(T|X,Y) = \mathbb{E}_{\mathbf{w},\mathbf{Q}\sim H_{\boldsymbol{\theta}}}[\log P_{\boldsymbol{\theta}}(T|X,Y,\mathbf{w},\mathbf{Q})] - D_{KL}\left[H_{\boldsymbol{\theta}}(\mathbf{w},\mathbf{Q}|X,Y,T) \,\middle\|\, \hat{P}_{\boldsymbol{\theta}}(\mathbf{w},\mathbf{Q}|Y)\right] \tag{22}$$

**Proposition 7.** *The variational lower bound $\mathcal{L}_{\boldsymbol{\theta}}(T|X,Y)$ in Equation (22) is bi-equivariant.*

We provide the proof of Proposition 7 in Appendix F.7. The Kullback-Leibler divergence term in Equation (22) is provided in Appendix B.2. Once the query model has been sufficiently trained, we remove the surrogate query model and return to the maximum likelihood training in Equation (16).

## 6 EXPERIMENTAL RESULTS

We design the experiments to assess the generalization performance of our approach (EDFs) when the number of demonstrations is very limited. Similar to Simeonov et al. (2021), we evaluate the generalization performance of models with a mug-hanging task and bowl/bottle pick-and-place tasks. In the mug-hanging task, a mug has to be picked by its rim and then hung on a hanger by its handle. In the bowl/bottle pick-and-place task, a bowl/bottle should be picked and placed on a tray. As opposed to Simeonov et al. (2021), we randomize the pose of the hanger and tray to evaluate

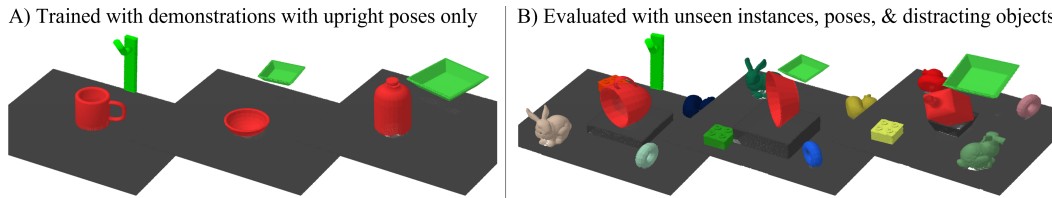

Figure 5: A) Only ten demonstrations with objects in upright poses are provided during the training. B) The models are evaluated with unseen object instances in unseen poses with unseen distractors.

Table 1: Pick-and-place success rates in various out-of-distribution settings.

| | Mug | | | Bowl | | | Bottle | | |
|---|---|---|---|---|---|---|---|---|---|
| | Pick | Place | **Total** | Pick | Place | **Total** | Pick | Place | **Total** |
| **Unseen Instances** | | | | | | | | | |
| SE(3)-TNs (Zeng et al., 2020) | 1.00 | 0.36 | 0.36 | 0.76 | 1.00 | 0.76 | 0.20 | 1.00 | 0.20 |
| EDFs (Ours) | 1.00 | **0.97** | **0.97** | **0.98** | 1.00 | **0.98** | **1.00** | 1.00 | **1.00** |
| **Unseen Poses** | | | | | | | | | |
| SE(3)-TNs (Zeng et al., 2020) | 0.00 | N/A | 0.00 | 0.00 | N/A | 0.00 | 0.00 | N/A | 0.00 |
| EDFs (Ours) | **1.00** | 1.00 | **1.00** | **1.00** | 1.00 | **1.00** | **0.95** | 1.00 | **0.95** |
| **Unseen Distracting Objects** | | | | | | | | | |
| SE(3)-TNs (Zeng et al., 2020) | 1.00 | 0.63 | 0.63 | 1.00 | 1.00 | 1.00 | 0.96 | 0.92 | 0.88 |
| EDFs (Ours) | 1.00 | **0.98** | **0.98** | 1.00 | 1.00 | 1.00 | **0.99** | **1.00** | **0.99** |
| **Unseen Instances, Arbitrary Poses & Distracting Objects** | | | | | | | | | |
| SE(3)-TNs (Zeng et al., 2020) | 0.25 | 0.04 | 0.01 | 0.09 | 1.00 | 0.09 | 0.26 | 0.88 | 0.23 |
| EDFs (Ours) | **1.00** | **0.95** | **0.95** | **0.95** | 1.00 | **0.95** | **0.95** | **1.00** | **0.95** |

with changing target placement poses. We use ten demonstrations of *upright poses* generated by probabilistic oracles for the training. We then evaluate the success rate for (1) unseen instances, (2) unseen poses (*lying poses*), (3) unseen distracting objects, and (4) unseen instances in arbitrary poses (50% lying, 50% upright but in arbitrary elevation) with unseen distracting objects. We illustrate the experimental setups in Figure 5. Details are provided in Appendix H.

For baselines, we use $SE(3)$-Transporter Networks ($SE(3)$-TNs) (Zeng et al., 2020) as the state-of-the-art method for end-to-end visual manipulation. However, for the mug hanging task, we find that $SE(3)$-TNs entirely fail due to the multimodality of the demonstrations. Therefore, we instead train $SE(3)$-TNs using unimodal, low-variance demonstrations only for the mug-hanging task. For fair comparison, we provide the result for EDFs trained with the same demonstrations in Table 3 of Appendix I. The experimental results for $SE(3)$-TNs and EDFs are summarized in Table 1. For EDFs, we provide the learning curves for the mug task in Figure 6.

Note that we do not directly compare EDFs with NDFs because (1) NDFs require the target placement poses to be fixed, and (2) NDFs require object segmentation pipelines (Performance of NDFs without object segmentation is provided in Appendix J). Instead, we perform an ablation study by using only type-0 descriptors as NDFs. One may consider this as the end-to-end trainable and bi-equivariant modification of NDFs (See Section 4.1 and Appendix E). We con-

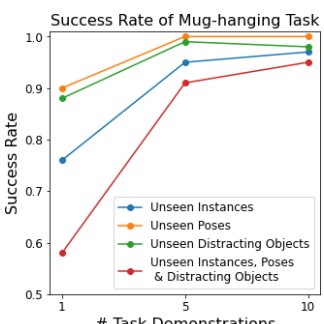

Figure 6: The success rate of EDFs for the mug-hanging task with respect to the number of demonstrations.

trol the ablated model's query point number such that the wall-clock inference time is similar to or slightly longer than EDFs. Note that the query point number serves a similar role to the batch size of inputs. Details on the ablated model and EDFs are provided in Appendix H. All the times were

Table 2: Success rate and inference time of the ablated model and EDFs. All the evaluations are done in the *unseen instances, poses & distracting objects* setting.

| Descriptor Type | Mug | | | Bowl | | | Bottle | | |
|---|---|---|---|---|---|---|---|---|---|
| | Pick | Place | **Total** | Pick | Place | **Total** | Pick | Place | **Total** |
| **NDF-like (Type-0 Only)** | | | | | | | | | |
| Inference Time | 5.7s | 8.6s | 14.3s | 6.1s | 9.9s | 16.0s | 5.8s | 17.3s | 23.0s |
| Success Rate | 0.84 | 0.77 | 0.65 | 0.60 | 0.95 | 0.57 | 0.66 | 0.95 | 0.63 |
| **EDFs (Type-0∼3)** | | | | | | | | | |
| Inference Time | 5.1s | 8.3s | 13.4s | 5.2s | 10.4s | 15.6s | 5.2s | 11.5s | 16.7s |
| Success Rate | **1.00** | **0.95** | **0.95** | **0.95** | **1.00** | **0.95** | **0.95** | **1.00** | **0.95** |

measured using an Intel i9-12900k CPU (P-core only) with an Nvidia RTX3090 GPU. Experimental results for the ablated model with more query points can be found in Table 4 of Appendix I.

**Analysis** As can be seen in Table 1, EDFs outperform $SE(3)$-TNs (Zeng et al., 2020) for all three tasks. It is obvious that EDFs generalize much better than $SE(3)$-TNs for unseen poses, as EDFs are $SE(3)$-equivariant, whereas $SE(3)$-TNs are only $SE(2)$-equivariant. We provide a qualitative example in Figure 7. EDFs also generalize much better than $SE(3)$-TNs for unseen instances and/or unseen distracting objects. For example, $SE(3)$-TNs often fail to correctly regress the z-axis of the grasp pose of an unseen instance when the object height differs a lot from the trained objects. In contrast, EDFs rarely fail under such height differences due to the $SE(3)$-equivariance. For the distracting objects, we presume that the reason why EDFs perform better is that the point cloud inputs provide better geometric information than the orthographic RGB-D inputs that $SE(3)$-TNs take.

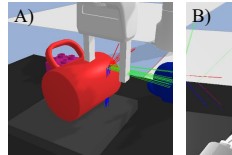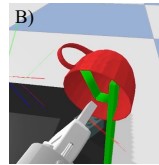

Figure 7: A) **SE(3)-TNs** fail to pick the object in an unseen pose due to the lack of $SE(3)$-equivariance. B) **Type-0 only descriptors** fail to place the object in a proper orientation due to the lack of orientational sensitivity.

The ablation study shows that using higher-type descriptors significantly increases generalization performance when the number of query points is highly limited due to computational constraints. As can be seen in Table 2, EDFs outperform the ablated model that only uses type-0 descriptors as NDFs. As was discussed in Section 4.1, we presume that this is due to the orientational insensitivity of type-0 descriptors. As type-0 descriptors cannot represent orientations alone, query points are crucial in representing orientations for the ablated model. In contrast, the higher-type descriptors of EDFs can represent the orientation without the help of query points. As a result, EDFs can maintain orientational accuracy in unseen situations in which low-quality query points are expected. We illustrate the failure case of the ablated model in Figure 7.

## 7 DISCUSSION AND CONCLUSION

There are several limitations to EDFs that should be resolved in future works. First, faster sampling methods are required for real-time manipulations. Cooperative learning (Xie et al., 2018a; 2021b; 2022) and amortized sampling (Wang & Liu, 2016; Xie et al., 2021c) can be applied to accelerate the MCMC sampling. In addition, EDFs are not intended for tasks with significant occlusions to the target object. Future work may also encompass 3D reconstruction methods. Lastly, EDFs cannot solve problems at the *trajectory* level, which is a shared problem with NDFs. Future work should define the adequate equivariance condition for full trajectory-level manipulation tasks.

To summarize, we introduce EDFs and the corresponding energy-based models, which are $SE(3)$-equivariant end-to-end models for robotic manipulations. We propose novel bi-equivariant energy-based models, which provably allow highly sample efficient and generalizable learning. We show by experiment that our method is highly sample efficient and generalizable to unseen object instances, unseen object poses, and unseen distracting objects. Lastly, we show by the ablation study that higher-degree equivariance (type1 or higher) is important for generalizability.

**Acknowledgement** This work was supported by the National Research Foundation of Korea (NRF) grant funded by the Korea government (MSIT) (No. 2021R1A2B5B01002620). This work was partially supported by the Korea Institute of Science and Technology (KIST) intramural grants (2E31570).

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

## A  BI-INVARIANT VOLUME FORM

A *bi-invariant volume form* $dg$ of a $n$-dimensional Lie group $G$ is a differential $n$-form that satisfies

$$dg = d(hg) = d(gh) \quad \forall h \in G \tag{23}$$

such that for all Borel subsets $\Omega \subseteq G$ and for all well-behaved function $f(g) : G \to \mathbb{R}$

$$\int_{g \in \Omega} dg f(g) = \int_{hg \in h\Omega} d(hg) f\left(h^{-1}(hg)\right) \quad \forall h \in G \qquad \text{(Left invariance)}$$

$$\int_{g \in \Omega} dg f(g) = \int_{gh \in \Omega h} d(gh) f\left((gh)h^{-1}\right) \quad \forall h \in G \qquad \text{(Right invariance)} \tag{24}$$

where $h\Omega = \{hg | g \in \Omega\}$ and $\Omega h = \{gh | g \in \Omega\}$. Let $\mathbf{z} = \phi(g)$ be some coordinatization of $G$ that for some function $J(\mathbf{z}) : \mathbb{R}^n \to \mathbb{R}$, $dg$ can be explicitly written as

$$dg = J(\mathbf{z}) d^n \mathbf{z}$$

Let the left/right group translations in coordinates be $\mathbf{z}^{(l)}(\mathbf{z}) = \phi(hg(\mathbf{z}))$ and $\mathbf{z}^{(r)}(\mathbf{z}) = \phi(g(\mathbf{z})h)$. The bi-equivariance condition in Equation (23) can then be expressed in the coordinate form as

$$d(hg) = J(\mathbf{z}^{(l)}) d^n \mathbf{z}^{(l)} = J(\mathbf{z}^{(l)}) \det \left[\frac{\partial \mathbf{z}^{(l)}}{\partial \mathbf{z}}\right] d^n \mathbf{z} = J(\mathbf{z}) d^n \mathbf{z} = dg$$

$$d(gh) = J(\mathbf{z}^{(r)}) d^n \mathbf{z}^{(r)} = J(\mathbf{z}^{(r)}) \det \left[\frac{\partial \mathbf{z}^{(r)}}{\partial \mathbf{z}}\right] d^n \mathbf{z} = J(\mathbf{z}) d^n \mathbf{z} = dg \tag{25}$$

thus leading to the following equations:

$$J(\mathbf{z}^{(l)}) \det \left[\frac{\partial \mathbf{z}^{(l)}}{\partial \mathbf{z}}\right] = J(\mathbf{z}) = J(\mathbf{z}^{(r)}) \det \left[\frac{\partial \mathbf{z}^{(r)}}{\partial \mathbf{z}}\right] \tag{26}$$

Detailed introduction to invariant volume forms on Lie groups can be found in (Chirikjian, 2011; Zee, 2016). Readers interested in differential forms may find differential geometry textbooks (Spivak, 2018; Nakahara, 2018) useful.

For example, the translation group $(\mathbb{R}, +)$ admits a bi-invariant volume form $dx$ because

$$\int_{x \in \Omega} dx f(x) = \int_{x+\epsilon \in \Omega + \epsilon} d(x + \epsilon) \frac{\cancel{dx}}{\cancel{d(x + \epsilon)}} f((x + \epsilon) - \epsilon) = \int_{x+\epsilon \in \Omega + \epsilon} d(x + \epsilon) f((x + \epsilon) - \epsilon) \tag{27}$$

Note that the right invariance in Equation (27) is sufficient to prove the bi-invariance of $dx$ because $(\mathbb{R}, +)$ is commutative, that is $x + \epsilon = \epsilon + x \; \forall x, \epsilon \in \mathbb{R}$.

On the other hand, for the multiplicative group $(\mathbb{R}_{\neq 0}, \times)$, $dx$ is not a bi-invariant volume form:

$$\int_{x \in \Omega} dx f(x) = \int_{\epsilon x \in \epsilon \Omega} d(\epsilon x) \frac{dx}{d(\epsilon x)} f(\epsilon^{-1}(\epsilon x))$$

$$= \frac{1}{\epsilon} \int_{\epsilon x \in \epsilon \Omega} d(\epsilon x) f(\epsilon^{-1}(\epsilon x)) \neq \int_{\epsilon x \in \epsilon \Omega} d(\epsilon x) f(\epsilon^{-1}(\epsilon x)) \qquad \forall \epsilon \in \mathbb{R}_{\neq 0} \tag{28}$$

However, $d(\log |x|) = dx/x$ is a bi-invariant volume form:

$$\int_{x \in \Omega} \frac{dx}{x} f(x) = \int_{\epsilon x \in \epsilon \Omega} \frac{d(\epsilon x)}{x} \frac{dx}{d(\epsilon x)} f(\epsilon^{-1}(\epsilon x)) \int_{\epsilon x \in \epsilon \Omega} \frac{d(\epsilon x)}{\epsilon x} f(\epsilon^{-1}(\epsilon x)) \tag{29}$$

Again, we only show the left invariance because the multiplicative group is commutative.

Note that not every Lie group does admit bi-invariant volume form. Nevertheless, the Lie groups that we are concerned in this paper, the $SO(3)$ group and the $SE(3)$ group, have bi-invariant volume forms. We reproduce here the bi-invariant volume forms of $SO(3)$ and $SE(3)$ in coordinate forms provided in Chirikjian (2011). The bi-invariant volume form on $SO(3)$ can be written in Euler angles $\alpha, \beta,$and $\gamma$ as

$$d\mathbf{R} = \frac{1}{8\pi^2} \sin \beta \, d\alpha \, d\beta \, d\gamma$$

The bi-invariant volume form on $SE(3)$ can be written in rotation-translation coordinate as

$$dT = d\mathbf{R}d^3\mathbf{v}$$

where $\mathbf{v} \in \mathbb{R}^3$ denotes the translation vector with respect to the space frame. Lastly, the bi-invariant volume form or bi-invariant integral measure of $SE(3)$ should not be confused with the bi-invariant metric, which does not exist for $SE(3)$ (Chirikjian, 2015).

# B  QUERY MODELS

## B.1  EQUIVARIANT QUERY POINTS

We use Stein variational gradient descent (SVGD) (Liu & Wang, 2016; Jaini et al., 2021) method to equivariantly draw query points $[\mathbf{q}_{i;\boldsymbol{\theta}}(Y)]_{i=1}^{i=N_q}$ in Equation (13) from the query weight field $w_{\boldsymbol{\theta}}(\mathbf{x}|Y)$. In this case, $w_{\boldsymbol{\theta}}(\mathbf{x}|Y)$ can be interpreted as an unnormalized probability distribution on $\mathbb{R}^3$. The SVGD equation (Liu & Wang, 2016) is given by

$$\mathbf{q}_i^{t+1} = \mathbf{q}_i^t + \epsilon \frac{1}{N_q} \sum_{j=1}^{N_q} \left[ k(\mathbf{q}_j^t, \mathbf{q}_i^t) \, \nabla_{\mathbf{x}} \log w_{\boldsymbol{\theta}}(\mathbf{x}|Y)|_{\mathbf{x}=\mathbf{q}_j^t} + \nabla_{\mathbf{x}} k(\mathbf{x}, \mathbf{q}_i^t)|_{\mathbf{x}=\mathbf{q}_j^t} \right] \tag{30}$$

$$k(\mathbf{x}, \mathbf{x}') = \exp\left[ -\frac{1}{h}\|\mathbf{x} - \mathbf{x}'\|^2 \right] \tag{31}$$

Note that SVGD is fully deterministic given initial points $\mathbf{q}_i^{t=0}$. We use $h_t = \text{med}_t^2 / \log N_q$ as (Liu & Wang, 2016) where $\text{med}_t$ denotes the median of the distances between all the points in $\left\{ \mathbf{q}_1^t, \mathbf{q}_2^t, \cdots, \mathbf{q}_{N_q}^t \right\}$. We take the final output of SVGD as the query points, that is

$$\mathbf{q}_i(Y) = \mathbf{q}_i^{t=t_{fin}} \tag{32}$$

for some $t_{fin} \geq 1$. We take $\epsilon$ and $t_{fin}$ as hyperparameters. In our work, we uses $\epsilon = 0.005$ and $t_{fin} = 100$.

We now show that the query point $\mathbf{q}_i(Y)$ in Equation (32) is $SE(3)$-equivariant.

**Proposition 8.** *The query points* $\{\mathbf{q}_i(Y)\}_{i=1}^{N_q}$ *are* $SE(3)$-*equivariant if the initial query points* $\left\{\mathbf{q}_i^{t=0}(Y)\right\}_{i=1}^{N_q}$ *are* $SE(3)$-*equivariant, that is*

$$T\mathbf{q}_i^{t=0}(Y) = \mathbf{q}_i^{t=0}(T \circ Y) \quad \forall i \quad \Rightarrow \quad T\mathbf{q}_i(Y) = \mathbf{q}_i(T \circ Y) \quad \forall i \tag{33}$$

To prove Proposition 8, we first explicitly denote the query points' dependence on $Y$ as $\mathbf{q}_i^t = \mathbf{q}_i^t(Y)$. We then propose the following lemma.

**Lemma 1.** *If* $\mathbf{q}_i^t(T \circ Y) = T\mathbf{q}_i^t(Y) \; \forall T \in SE(3)$, *the following equations hold:*

$$\begin{aligned} &k(\mathbf{q}_j^t(Y), \mathbf{q}_i^t(Y))\mathbf{R} \, \nabla_{\mathbf{x}} \log w_{\boldsymbol{\theta}}(\mathbf{x}|Y)|_{\mathbf{x}=\mathbf{q}_j^t(Y)} \\ &= k(\mathbf{q}_j^t(T \circ Y), \mathbf{q}_i^t(T \circ Y)) \, \nabla_{\mathbf{x}} \log w_{\boldsymbol{\theta}}(\mathbf{x}|T \circ Y)|_{\mathbf{x}=\mathbf{q}_j^t(T \circ Y)} \end{aligned} \tag{34}$$

$$\mathbf{R} \, \nabla_{\mathbf{x}} k(\mathbf{x}, \mathbf{q}_i^t(Y))|_{\mathbf{x}=\mathbf{q}_j^t(Y)} = \nabla_{\mathbf{x}} k(\mathbf{x}, \mathbf{q}_i^t(T \circ Y))|_{\mathbf{x}=\mathbf{q}_j^t(T \circ Y)} \tag{35}$$

Since $\|\mathbf{x} - \mathbf{x}'\|^2 = \|T\mathbf{x} - T\mathbf{x}'\|^2 \; \forall T \in SE(3)$, it is straightforward to prove that

$$k(\mathbf{x}, \mathbf{x}') = k(T\mathbf{x}, T\mathbf{x}') \quad \forall T \in SE(3) \tag{36}$$

To prove the equivariance of the gradient terms in Equation (34) and Equation (35), we prove the following lemma.

**Lemma 2.** *If* $f(\mathbf{x}|Y) = f(T\mathbf{x}|T \circ Y)$ *for some function* $f : \mathbb{R}^3 \times \mathcal{P} \to \mathbb{R}$, *the following holds.*

$$\nabla_{\mathbf{x}} f(\mathbf{x}|T \circ Y)|_{\mathbf{x}=T\mathbf{x}_0} = \mathbf{R} \, \nabla_{\mathbf{x}} f(\mathbf{x}|Y)|_{\mathbf{x}=\mathbf{x}_0} \tag{37}$$

*Proof.*

$$\nabla_{\mathbf{x}} f(\mathbf{x}|T \circ Y)|_{\mathbf{x}=T\mathbf{x}_0} = \nabla_{\mathbf{x}} f(T^{-1}\mathbf{x}|Y)\big|_{\mathbf{x}=T\mathbf{x}_0} \qquad (\because f(\mathbf{x}|Y) = f(T\mathbf{x}|T \circ Y))$$
$$= \nabla_{\mathbf{x}} f(\mathbf{x}'|Y)|_{\mathbf{x}=T\mathbf{x}_0} \qquad \text{(Change of variables } \mathbf{x}' = T^{-1}\mathbf{x})$$
$$= \mathbf{R} \, \nabla_{\mathbf{x}'} f(\mathbf{x}'|Y)|_{\mathbf{x}'=\mathbf{x}_0} = \mathbf{R} \, \nabla_{\mathbf{x}} f(\mathbf{x}|Y)|_{\mathbf{x}=\mathbf{x}_0} \qquad (\mathbf{x}' \to \mathbf{x})$$

where in the last line we used

$$\nabla_{\mathbf{x}} = \left( \frac{\partial \mathbf{x}'}{\partial \mathbf{x}} \right)^T \nabla_{\mathbf{x}'} = \left( \frac{\partial \left( \mathbf{R}^{-1}\mathbf{x} - \mathbf{R}^{-1}\mathbf{v} \right)}{\partial \mathbf{x}} \right)^T \nabla_{\mathbf{x}'} = (\mathbf{R}^{-1})^T \nabla_{\mathbf{x}'} = \mathbf{R} \nabla_{\mathbf{x}'}$$

$\square$

One can prove Lemma 1 using Lemma 2 and Equation (36).

We now prove Proposition 8 using Lemma 1. Let the sum on the right-hand side of Equation (30) be $\mathbf{s}(Y)$ such that

$$\mathbf{s}(Y) = \sum_{j=1}^{N_q} \left[ k(\mathbf{q}_j^t(Y), \mathbf{q}_i^t(Y)) \, \nabla_{\mathbf{x}} \log w_{\boldsymbol{\theta}}(\mathbf{x}|Y)|_{\mathbf{x}=\mathbf{q}_j^t(Y)} + \nabla_{\mathbf{x}} k(\mathbf{x}, \mathbf{q}_i^t(Y))|_{\mathbf{x}=\mathbf{q}_j^t(Y)} \right] \qquad (38)$$

We first consider the case with $t = 0$. One can prove that $\mathbf{s}(T \circ Y) = \mathbf{R}\mathbf{s}(Y)$ using Lemma 1 and the equivariance of the initial points $T\mathbf{q}_i^{t=0}(Y) = \mathbf{q}_i^{t=0}(T \circ Y)$, which was assumed in Proposition 8. It is then straightforward to prove that $\mathbf{q}_i^{t=1}(Y)$ is also equivariant.

*Proof.*

$$\mathbf{q}_i^{t=1}(T \circ Y) = \mathbf{q}_i^{t=0}(T \circ Y) + \epsilon \mathbf{s}(T \circ Y)$$
$$= T\mathbf{q}_i^{t=0}(Y) + \epsilon \mathbf{R}\mathbf{s}(Y)$$
$$= \mathbf{R}\mathbf{q}_i^{t=0}(Y) + \mathbf{v} + \epsilon \mathbf{R}\mathbf{s}(Y) \qquad (39)$$
$$= \mathbf{R}(\mathbf{q}_i^{t=0}(Y) + \epsilon \mathbf{s}(Y)) + \mathbf{v}$$
$$= T\mathbf{q}_i^{t=1}(Y)$$

$\square$

We now recursively apply this relation to $t = 1, 2, 3, \cdots, t_{fin}$ to conclude that the final query point is also equivariant, that is $\mathbf{q}_i^{t=t_{fin}}(Y) = \mathbf{q}_i^{t=t_{fin}}(T \circ Y)$. Therefore, the only requirement for our query points to be equivariant is the equivariance of the initial points: $T\mathbf{q}_i^{t=0}(Y) = \mathbf{q}_i^{t=0}(T \circ Y)$. We provide a simple (and clearly not the best) deterministic method that we used to sample the initial point $\mathbf{q}_i^{t=0}(Y)$ in Algorithm 1.

---

**Algorithm 1** Simple algorithm for deterministic and equivariant initial point sampling

---

**Input:** $Y = \{(\mathbf{y}_1, \mathbf{c}_1), \cdots, (\mathbf{y}_M, \mathbf{c}_M)\}$, $w(\mathbf{x}|Y)$, $N_{max}$, $r_{cluster}$
**Output:** $\mathbf{q}_1, \mathbf{q}_2, \cdots$
   $i \leftarrow 1$
   $Q \leftarrow \{\mathbf{y}_1, \cdots, \mathbf{y}_M\}$                 $\triangleright$ Initialize set $Q$ with the set of all the points in $Y$
   **while** $i \leq N_{max}$ **do**
      **if** $Q$ is not empty **then**
         $\mathbf{q}_i \leftarrow \arg\max_{\mathbf{y} \in Q} w(\mathbf{y}|Y)$           $\triangleright$ Take the point with largest weight in $Q$
         $Q \leftarrow Q - \left\{ \mathbf{z} \in Q \, \big| \, \|\mathbf{z} - \mathbf{q}_i\|^2 \leq r_{cluster} \right\}$     $\triangleright$ Remove the neighbors from $Q$
      **end if**
      $i \leftarrow i + 1$
   **end while**

---

### B.2 SURROGATE QUERY MODEL

Let $A(r)$ be the set of all the indices of the query points whose shortest distance to the point cloud $X$ is farther than some radius $r$ such that $A(r) = \{i \in \{1, 2, \cdots, N_q\} | d_{min}(T\mathbf{q}_i, X) \geq r\}$. The Kullback-Leibler divergence term in Equation (22) can be calculated as follows:

$$
\begin{aligned}
& D_{KL}(H(\mathbf{w}, \mathbf{Q}|X, Y, T) \| \hat{P}(\mathbf{w}, \mathbf{Q}|Y)) \\
&= \left[ \prod_{i=1}^{N_Q} \int_{\mathbb{R}^+} dw_i \int_{\mathbb{R}^3} d\mathbf{q}_i H_i(w_i, \mathbf{q}_i|X, Y, T) \right] \sum_{j=1}^{N_Q} \log \frac{H_j(w_j, \mathbf{q}_j|X, Y, T)}{\hat{P}_j(w_j, \mathbf{q}_j|Y)} \\
&= \sum_{i \in A(r)} \int_{\mathbb{R}^+} dw_i \int_{\mathbb{R}^3} d\mathbf{q}_i H_i(w_i, \mathbf{q}_i|X, Y, T) \log \frac{H_i(w_i, \mathbf{q}_i|X, Y, T)}{\hat{P}_i(w_i, \mathbf{q}_i|Y)} \\
&= \sum_{i \in A(r)} \int_{\mathbb{R}^+} \cancel{dw_i} \frac{dl_i}{\cancel{dw_i}} \int_{\mathbb{R}^3} d\mathbf{q}_i \cancel{\delta^{(3)}(\mathbf{q}_i - \mathbf{q}_i(Y))} \mathcal{N}(l_i; \alpha, \sigma_H) \\
& \qquad \times \log \frac{\mathcal{N}(l_i; \alpha, \sigma_H) \cancel{\delta^{(3)}(\mathbf{q}_i - \mathbf{q}_i(Y))}}{\mathcal{N}(l_i; \log w(\mathbf{q}_i|Y), \sigma_H) \cancel{\delta^{(3)}(\mathbf{q}_i - \mathbf{q}_i(Y))}} \\
&= \sum_{i \in A(r)} \int_{\mathbb{R}} dl_i \mathcal{N}(l_i; \alpha, \sigma_H) \log \frac{\mathcal{N}(l_i; \alpha, \sigma_H)}{\mathcal{N}(l_i; \log w(\mathbf{q}_i(Y)|Y), \sigma_H)} \\
&= \sum_{i \in A(r)} \int_{\mathbb{R}} dl_i \mathcal{N}(l_i; \alpha, \sigma_H) \left[ -\frac{1}{2\sigma_H^2} \left\{ (l_i - \alpha)^2 - (l_i - \log w(\mathbf{q}_i(Y)|Y))^2 \right\} \right] \\
&= \sum_{i \in A(r)} \mathbb{E}_{\epsilon \sim \mathcal{N}_{0,1}} \left[ -\frac{1}{2\sigma_H^2} \left\{ (\epsilon \sigma_H)^2 - (\epsilon \sigma_H + \alpha - \log w(\mathbf{q}_i(Y)|Y))^2 \right\} \right] \\
&= \sum_{i \in A(r)} \frac{1}{2} \left( \frac{\log w(\mathbf{q}_i(Y)|Y) - \alpha}{\sigma_H} \right)^2
\end{aligned}
\tag{40}
$$

where in the third line we used

$$
\log \frac{H_j(w_j, \mathbf{q}_j|X, Y, T)}{\hat{P}_j(w_j, \mathbf{q}_j|Y)} = \log \frac{\cancel{\hat{P}_j(w_j, \mathbf{q}_j|Y)}}{\cancel{\hat{P}_j(w_j, \mathbf{q}_j|Y)}} = 0 \quad \forall j \notin A(r)
$$

and in the last line, we used the reparameterization trick for Gaussian distributions (Kingma & Welling, 2013).

### B.3 QUERY ATTENTION

Due to the computational limitations, it is desirable to have as few query points as possible during the inference time. Therefore, instead of directly taking $w_i = w_{\boldsymbol{\theta}}(\mathbf{q}_{i;\boldsymbol{\theta}}(Y))$, we normalize the query weights by taking

$$
w_i = \frac{w_{\boldsymbol{\theta}}(\mathbf{q}_i(Y))}{\sum_{j=1}^{N_q} w_{\boldsymbol{\theta}}(\mathbf{q}_j(Y))}
\tag{41}
$$

such that the query points compete with each other during the training. As a result of this competition, only a few query points have non-negligible weights. Therefore, during the inference time, we can calculate for only a few query points with non-negligible weights instead of calculating the whole query points to save the computation. Note that the normalized query weight is still equivariant because only scalar addition and division were used in Equation (41).

## C INTUITION BEHIND THE BI-EQUIVARIANCE CONDITION

To illustrate the bi-equivariance condition in Equation (6), consider an object placing task where $T_{go} \in SE(3)$ is the object pose ($o$) in the gripper frame ($g$) and $T_{sd} \in SE(3)$ is the desired object

A) Left Equivariance                    B) Right Equivariance

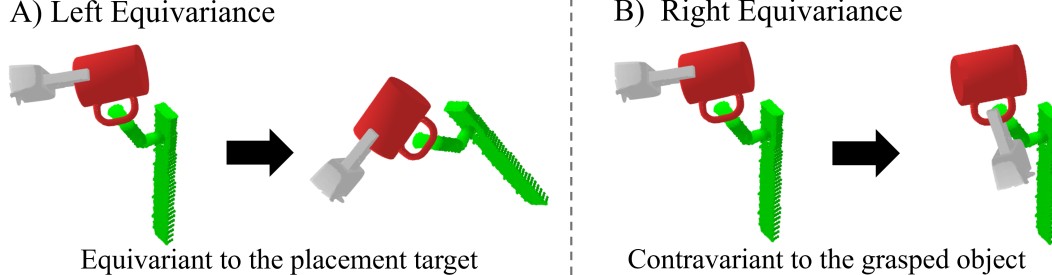

Equivariant to the placement target     Contravariant to the grasped object

Figure 8: A) The end-effector should follow the transformation of the placement target to keep the relative pose between the object and the placement target invariant. B) The end-effector should transform contravariantly to compensate for the transformation of the grasped object such that the relative pose between the object and the placement target is invariant.

pose ($d$) that is to be placed in the scene frame ($s$). Consequently, the gripper pose in the scene frame $T_{sg}$ should satisfy the following equation

$$T_{sd} = T_{sg}T_{go} \tag{42}$$

Now, let the desired pose of the object to be placed has been transformed as $T_{sd} \to T'_{sd} = ST_{sd}$ for some transformation $S \in SE(3)$. In order to keep the relation in Equation (42) invariant such that for the new gripper pose $T'_{sg}$ the equation $T'_{sd} = T'_{sg}T_{go}$ holds, it should be that $T'_{sg} = ST_{sg}$. Similarly, if the pose of the grasped object is transformed as $T_{go} \to T''_{go} = ST_{go}$, the gripper pose should also be transformed as $T_{sg} \to T''_{sg} = T_{sg}S^{-1}$ to keep the relation in Equation (42) invariant. Since $T_{sd}$ and $T_{go}$ are implicitly encoded in $X$ and $Y$ individually, one can naively substitute $T_{sd}$ and $T_{go}$ into $X$ and $Y$ to get the equation

$$P(T_{sg}|X, Y) = P(T'_{sg} = ST_{sg}|S \circ X, Y) = P(T''_{sg} = T_{sg}S|X, S^{-1} \circ Y)$$

This is the intuition behind Equation (6). We illustrate the bi-equivariance condition in Figure 8

## D  SAMPLING DETAILS

For the sampling, we use the *Metropolis-Hastings* (MH) algorithm (Hastings, 1970; Metropolis et al., 1953) and *Langevin algorithm* on the $SE(3)$ manifold (Brockett, 1997; Chirikjian, 2011; Davidchack et al., 2017). Unlike the MH, the Langevin algorithm does not suffer from high rejection ratios and converges with much fewer iterations. However, the Langevin algorithm requires the gradient of the energy function and thus is computationally inefficient. In addition, the time step for the Langevin algorithm cannot be arbitrarily high to maintain the precision of the dynamics. Therefore, we first run MH for rapid exploration and then run the Langevin algorithm using the MH samples as initial seeds. Note that the differential geometric aspects of the $SE(3)$ manifold must be considered in implementing these methods.

For the following sections, we provide details of the sampling methods that we used. We first explain the proposal distributions that we used to run the MH algorithm on the $SE(3)$ manifold. We then introduce the Langevin algorithm on $SE(3)$. We calculate the Langevin dynamics in quaternion-translation parameterization as Davidchack et al. (2017) to avoid singularity while benefiting from commonly used autograd packages.

### D.1  PROPOSAL DISTRIBUTION FOR MH

The Metropolis-Hastings (MH) algorithm (Hastings, 1970) is a propose-and-reject algorithm used for sampling from some probability distribution $dP(T)$. First, a proposal point $T_p$ is sampled from the proposal distribution $dQ(T_p|T_t)$. The proposed point $T_p$ is stochastically accepted or rejected by the acceptance ratio $A = \min\left[1, \frac{dP(T_p)dQ(T_t|T_p)}{dP(T_t)dQ(T_p|T_t)}\right]$. If the proposal is accepted, the next point is the proposed point, that is $T_{t+1} = T_p$. If rejected, the point remains the same, that is $T_{t+1} = T_t$. It is known that the steady-state distribution $dP_\infty(T_\infty)$ converges to $dP(T)$.

We decompose the proposal distribution $dQ(T|T_t) = Q(T|T_t)dT$ into 1) the orientation proposal distribution $Q_{\mathbf{R}}(\mathbf{R}|\mathbf{R}_t)d\mathbf{R}$ and 2) the position proposal distribution $Q_{\mathbf{v}}(\mathbf{v}|\mathbf{v}_t)d^3\mathbf{v}$ such that

$$Q(T|T_t)dT = Q_{\mathbf{R}}(\mathbf{R}|\mathbf{R}_t)d\mathbf{R} \times Q_{\mathbf{v}}(\mathbf{v}|\mathbf{v}_t)d^3\mathbf{v}$$

where $d^3\mathbf{v}$ is the Euclidean volume element and $dR$ is the bi-invariant volume form of $SO(3)$ (See Appendix A). We use Gaussian distribution for the position proposal, that is $Q_{\mathbf{v}}(\mathbf{v}_p|\mathbf{v}_t) = \mathcal{N}(\mathbf{v}_p; \mathbf{v}_t, \sigma\mathbf{I})$. For the orientation proposal $Q_{\mathbf{R}}(\mathbf{R}_p|\mathbf{R}_t)$, we used $\mathcal{IG}_{SO(3)}$ which is the normal distribution on $SO(3)$ (Nikolayev & Savyolov, 1970; Savyolova, 1994; Leach et al., 2022). Concrete calculation and sampling methods for $\mathcal{IG}_{SO(3)}$ are provided in Appendix D.2.

## D.2 NORMAL DISTRIBUTION ON SO(3)

We follow the method in Leach et al. (2022) to calculate and sample from $\mathcal{IG}_{SO(3)}$, the normal distribution on $SO(3)$ (Nikolayev & Savyolov, 1970; Savyolova, 1994). In the axis-angle parameterization, our orientation proposal distribution $Q_{\mathbf{R}}(\mathbf{R}_p|\mathbf{R}_t)d\mathbf{R}$, which is $\mathcal{IG}_{SO(3)}$, can be written as

$$Q_{\mathbf{R}}(\mathbf{R}_p|\mathbf{R}_t)d\mathbf{R} = (1 - \cos\omega)/\pi \times f_\epsilon(\omega)d\omega d\Omega$$

where $\omega \in [0, \pi)$ is the rotation angle, $d\Omega = \sin\theta/\pi \times d\theta d\phi$ is the uniform volume element over the sphere $S^2$ where the rotation axis lies, and $f_\epsilon(\omega)$ is as follows:

$$f_\epsilon(\omega) = \left[ \sum_{l=0}^{\infty} (2l+1)e^{-\epsilon l(l+1)} \frac{\sin((2l+1)\omega/2)}{\sin\omega/2} \right] \tag{43}$$

We approximate the infinite sum in Equation (43) by summing up to sufficiently high $l$. Note that the summand in Equation (43) decays exponentially fast to the square of $l$. Therefore, the approximation is justified. The rotation axis vector can be easily sampled by first sampling from three-dimensional Gaussian and then normalizing it. For the sampling of $\omega$, one may use numerical inverse transform sampling. As noted by Leach et al. (2022), the volume element $(1 - \cos\omega)/\pi$ should be multiplied to $f_\epsilon(\omega)$ for the inverse transform sampling.

## D.3 LANGEVIN MCMC ON SE(3)

Let $\mathcal{V}_i$ be the $i$-th basis of the Lie algebra of an unimodular Lie group $G$. Consider the following stochastic process $g(t) \in G$ generated by a Lie algebra $\delta X(t) = \sum_i \delta X_i(t)\mathcal{V}_i \in T_e G$ such that $g(t) = g(0) \exp[\delta X(0)] \exp[\delta X(dt)] \cdots \exp[\delta X(t - dt)]$. The Langevin dynamics for $G$ is then

$$\delta X_i(t) = -\mathcal{L}_{\mathcal{V}_i} [E(g)] dt + \sqrt{2} dw_i \tag{44}$$

where $dw_i \sim \mathcal{N}_{0;\sqrt{dt}}$ is the standard Wiener process and $\mathcal{L}_{\mathcal{V}} f = \left( \frac{d}{ds} f(g \exp[s\mathcal{V}]) \right)\big|_{s=0}$ is the (left) Lie derivative of a function $f$ on $G$ along $\mathcal{V}$. It is known that this process converges to

$$dP_\infty(g) \propto \exp[-E(g)] dg$$

when $t \to \infty$ where $dg$ is the (left) invariant volume form of $G$ (Brockett, 1997; Chirikjian, 2011).

Davidchack et al. (2017) provide concrete ways to calculate the Lie derivative and the Langevin dynamics on $SE(3)$ in quaternion-translation parameterization. Quaternion-translation parameterization is convenient because it has no singularities. Therefore, the gradients from commonly used autograd packages can be easily used to calculate the dynamics. For $SE(3)$, $\mathcal{V}_i$ is the Lie algebra basis of $SO(3)$ for $i = 1, 2, 3$ and the translation generator for $i = 4, 5, 6$. Let the quaternion-translation parameterization be $\mathbf{z} = (\mathbf{q}, \mathbf{v}) \in S^3 \times \mathbb{R}^3 \subset \mathbb{R}^7$. Let $\mathbf{L} \in \mathbb{R}^{7 \times 6}$ be the Lie derivative matrix whose $(\mu, i)$'th element is $[\mathbf{L}]^\mu_i = \mathcal{L}_{\mathcal{V}_i} z^\mu$. The matrix can be calculated as

$$\mathbf{L} = \begin{bmatrix} \mathbf{L}_{SO(3)} & \mathbf{0}_{4\times3} \\ \mathbf{0}_{3\times3} & \mathbf{I}_{3\times3} \end{bmatrix}$$

$$\mathbf{L}_{SO(3)} = \frac{1}{2} \begin{bmatrix} -q^2 & -q^3 & -q^4 \\ q^1 & -q^4 & q^3 \\ q^4 & q^1 & -q^2 \\ -q^3 & q^2 & q^1 \end{bmatrix} \tag{45}$$

where $\mathbf{q} = q^1 + q^2 i + q^3 j + q^4 k$. Derivations of Equation (45) can be found in (Davidchack et al., 2017). Since the chain rule holds for the Lie derivatives (Chirikjian, 2011), the Equation (44) can be written in the parameterized form as

$$dz = \begin{pmatrix} d\mathbf{q} \\ d\mathbf{v} \end{pmatrix} = -\mathbf{G}^{-1}\nabla_{\mathbf{z}}E(\mathbf{z})dt + \sqrt{2}\mathbf{L}d\mathbf{w} \qquad (46)$$

where $\mathbf{G}^{-1} = \mathbf{L}\mathbf{L}^T$. We calculate the gradient of the energy $\nabla_{\mathbf{z}}E(\mathbf{z})$ using typical autograd packages. Note that $d\mathbf{q}$ in Equation (46) satisfies the unit-quaternion constraint $\mathbf{q} \cdot d\mathbf{q} = 0$ such that $\mathbf{q} + d\mathbf{q} \in S^3$ (Davidchack et al., 2017). In practice, however, we reproject $\mathbf{q} + d\mathbf{q}$ onto $S^3$ by a normalization because of the inaccuracy in numerical integration.

## E    PICK-MODEL AND THE RELATIONSHIP TO NDFs

**Pick-model**    For the place model, the point cloud of the end-effector $Y$ is always different because of the grasped object. On the other hand, $Y$ is always the same for the pick-model because no object has been grasped yet. Therefore, we remove the $Y$-dependence of the query EDF and the query density by taking $\psi_{\boldsymbol{\theta}}(\mathbf{x}|Y)$ to $\psi_{\boldsymbol{\theta}}(\mathbf{x})$ and $\rho_{\boldsymbol{\theta}}(\mathbf{x}|Y)$ to $\rho_{\boldsymbol{\theta}}(\mathbf{x})$. In this case, the energy function $E_{\boldsymbol{\theta}}(T|X, Y)$ in Equation (14) becomes

$$E_{\boldsymbol{\theta}}(T|X, Y) = E_{\boldsymbol{\theta}}(T|X) = \sum_{i=1}^{N_q} w_i \|\boldsymbol{\varphi}_{\boldsymbol{\theta}}(T\mathbf{q}_i|X) - \mathbf{D}(\mathbf{R})\psi_i\|^2 \qquad (47)$$

where $w_i$, $\mathbf{q}_i$ and $\psi_i$ are not the outputs of some functions anymore but just parameters that are either predefined or learned.

**Relationship to NDFs**    We now illustrate the relation of Equation (47) to the energy function of NDFs (Simeonov et al., 2021). Let the energy function in Simeonov et al. (2021) be

$$E_{NDF}(T|X) = \sum_{i=1}^{N_q} \|\boldsymbol{\varphi}(T\mathbf{q}_i|X) - \psi_i\|_1 \qquad (48)$$

$$\psi_i = \frac{1}{N_{demo}} \sum_{n=1}^{N_{demo}} \boldsymbol{\varphi}(\hat{T}_n \mathbf{q}_i | \hat{X}_n) \qquad (49)$$

where $\hat{T}_n$ and $\hat{X}_n$ are the grasp pose and the point cloud input of the $n$'th demonstration. The Equation (48) can be understood as a special case of Equation (47) with (i) the $L_1$ error instead of the square error, (ii) all the query weights being constant $w_i = 1$, and (iii) all the components of the feature EDF $\boldsymbol{\varphi}(\mathbf{x}|X)$ being the *invariant* scalars (type-0 vectors) such that $\mathbf{D}(\mathbf{R}) = \mathbf{I}$.

**Relationship to other variants of NDFs**    Other recent works derived from NDFs (Simeonov et al., 2022; Chun et al., 2023) also are closely related to EDFs. *Relational Neural Descriptor Fields* (R-NDFs) (Simeonov et al., 2022) extend NDFs' fixed placement target tasks to object rearrangement tasks in which a placement target is also an object with varying poses. Instead of using fixed target descriptors as NDFs, R-NDFs utilize another descriptor field to represent the target placement object. The resulting R-NDFs' energy function is very similar to EDFs' bi-equivariant energy function. This is a natural consequence due to the bi-equivariant nature of object rearrangement tasks.

On the other hand, *Local Neural Descriptor Fields* (L-NDFs) (Chun et al., 2023) focus on imposing locality on the descriptor fields. While EDFs and L-NDFs both try to exploit locality, the motivation is largely different. Chun et al. (2023) focus on locality to improve generalization and transferability to novel objects. This is distinguished from the major motivation for imposing locality on EDFs: removing the necessity of object segmentation.

One thing to note is that these studies are not mutually exclusive but complement each other. EDFs can be pre-trained with a similar method to NDFs and R-NDFs. The strict $SE(3)$-equivariance of EDFs can be relaxed by imposing the $SE(3)$-equivariance on the loss function as L-NDFs instead of imposing it on the model itself. Conversely, these methods may benefit from the end-to-end trainability and generative nature of EDFs' energy-based model. The orientational sensitivity of higher-type (equivariant) descriptors should also be beneficial to these methods.

**Irrepwise $L_1$ Norm**  For closer analogy with the energy function of NDFs, we propose using *irrepwise $L_1$ norm*. Let an equivariant vector $\mathbf{f}$ be given by $\mathbf{f} = \bigoplus_{n=1}^{N} \mathbf{f}^{(n)}$ where $\mathbf{f}^{(n)}$ is a type-$l_n$ vector. We then define the irrepwise $L_1$ norm as

$$\|\mathbf{f}\|_1^I = \sum_{n=1}^{N} \|\mathbf{f}^{(n)}\|_2$$

If we use irrepwise $L_1$ norm in Equation (47) and confine all the vectors in EDFs to be of type-$0$, Equation (47) and Equation (48) are exactly identical. Although we did not use irrepwise $L_1$ norm in our work, we expect that this modification would be more robust to outliers than using the square error term.

## F  PROOFS

### F.1  PROOF OF PROPOSITION 1

We first prove the left equivariance.

*Proof.*

$$
\begin{aligned}
\int_{T \in S\Omega} dP(T|S \circ X, Y) &= \int_{T \in S\Omega} dT P(T|S \circ X, Y) \\
&= \int_{T \in S\Omega} dT P(S^{-1}T|X, Y) \qquad (\because P(ST|S \circ X, Y) = P(T|X, Y)) \\
&= \int_{S^{-1}T \in \Omega} d(S^{-1}T) P(S^{-1}T|X, Y) \qquad (\because \text{bi-invariance of } dT) \\
&= \int_{T \in \Omega} dT P(T|X, Y) \qquad\qquad\qquad\qquad (S^{-1}T \to T) \\
&= \int_{T \in \Omega} dP(T|X, Y)
\end{aligned}
$$

$\square$

The right equivariance can be similarly proved.

*Proof.*

$$
\begin{aligned}
\int_{T \in \Omega S} dP(T|X, S^{-1} \circ Y) &= \int_{T \in \Omega S} dT P(T|X, S^{-1} \circ Y) \\
&= \int_{T \in \Omega S} dT P(TS^{-1}|X, Y) \quad (\because P(TS|X, S^{-1} \circ Y) = P(T|X, Y)) \\
&= \int_{TS^{-1} \in \Omega} d(TS^{-1}) P(TS^{-1}|X, Y) \qquad (\because \text{bi-invariance of } dT) \\
&= \int_{T \in \Omega} dT P(T|X, Y) \qquad\qquad\qquad\qquad (TS^{-1} \to T) \\
&= \int_{T \in \Omega} dP(T|X, Y)
\end{aligned}
$$

$\square$

### F.2  PROOF OF PROPOSITION 2

Let the partition function (the denominator) of Equation (9) be $Z(X, Y)$.

**Lemma 3.** *For a bi-equivariant energy function $E(T|X, Y)$, the following equation holds.*

$$Z(X, Y) = Z(S \circ X, Y) = Z(X, S^{-1} \circ Y) \tag{50}$$

*Proof.*

$$Z(S \circ X, Y) = \int_{SE(3)} dT \exp\left[-E(T|S \circ X, Y)\right]$$

$$= \int_{SE(3)} dT \exp\left[-E(S^{-1}T|X, Y)\right] \qquad (\because E(ST|S \circ X, Y) = E(T|X, Y))$$

$$= \int_{SE(3)} d(S^{-1}T) \exp\left[-E(S^{-1}T|X, Y)\right] \qquad (\because \text{bi-invariance of } dT)$$

$$= \int_{SE(3)} dT \exp\left[-E(T|X, Y)\right] = Z(X, Y) \qquad (S^{-1}T \to T)$$

$$Z(X, S^{-1} \circ Y) = \int_{SE(3)} dT \exp\left[-E(T|X, S^{-1} \circ Y)\right]$$

$$= \int_{SE(3)} dT \exp\left[-E(TS^{-1}|X, Y)\right] \quad (\because E(TS|X, S^{-1} \circ Y) = E(T|X, Y))$$

$$= \int_{SE(3)} d(TS^{-1}) \exp\left[-E(TS^{-1}|X, Y)\right] \qquad (\because \text{bi-invariance of } dT)$$

$$= \int_{SE(3)} dT \exp\left[-E(T|X, Y)\right] = Z(X, Y) \qquad (TS^{-1} \to T)$$

$\square$

Now we prove the bi-equivariance of Equation (9) using Lemma 3.

*Proof.*

$$P(ST|S \circ X, Y) = \exp\left[-E(ST|S \circ X, Y)\right]/Z(S \circ X, Y)$$
$$= \exp\left[-E(T|X, Y)\right]/Z(X, Y)$$
$$= P(T|X, Y)$$
$$= \exp\left[-E(TS|X, S^{-1} \circ Y)\right]/Z(X, S^{-1} \circ Y)$$
$$= P(TS|X, S^{-1} \circ Y)$$

$\square$

### F.3 PROOF OF PROPOSITION 3

We first show that the energy function in Equation (10) satisfies $E(ST|S \circ X, Y) = E(T|X, Y)$ where $T = (\mathbf{R}, \mathbf{v}) \in SE(3)$ and $S = (\mathbf{R}_S, \mathbf{v}_S) \in SE(3)$.

*Proof.*

$$E(ST|S \circ X, Y)$$

$$= \int_{\mathbb{R}^3} d^3\mathbf{x}\rho(\mathbf{x}|Y)\|\boldsymbol{\varphi}(ST\mathbf{x}|S \circ X) - \mathbf{D}(\mathbf{R}_S\mathbf{R})\boldsymbol{\psi}(\mathbf{x}|Y)\|^2$$

$$= \int_{\mathbb{R}^3} d^3\mathbf{x}\rho(\mathbf{x}|Y)\|\mathbf{D}(\mathbf{R}_S)\boldsymbol{\varphi}(T\mathbf{x}|X) - \mathbf{D}(\mathbf{R}_S\mathbf{R})\boldsymbol{\psi}(\mathbf{x}|Y)\|^2 \qquad (\because \text{Equation (8)})$$

$$= \int_{\mathbb{R}^3} d^3\mathbf{x}\rho(\mathbf{x}|Y)\|\mathbf{D}(\mathbf{R}_S)\boldsymbol{\varphi}(T\mathbf{x}|X) - \mathbf{D}(\mathbf{R}_S)\mathbf{D}(\mathbf{R})\boldsymbol{\psi}(\mathbf{x}|Y)\|^2 \qquad (\because \text{Equation (1)})$$

$$= \int_{\mathbb{R}^3} d^3\mathbf{x}\rho(\mathbf{x}|Y)\|\boldsymbol{\varphi}(T\mathbf{x}|X) - \mathbf{D}(\mathbf{R})\boldsymbol{\psi}(\mathbf{x}|Y)\|^2 = E(T|X, Y)$$

where the orthogonality of the representation $\mathbf{D}(\mathbf{R})$ is used in the last line. Note that the inner product of two vectors is invariant to orthogonal transformations. $\square$

We now prove that $E(TS|X, S^{-1} \circ Y) = E(T|X, Y)$.

*Proof.*

$$
\begin{aligned}
&E(TS|X, S^{-1} \circ Y) \\
&= \int_{\mathbb{R}^3} d^3\mathbf{x}\rho(\mathbf{x}|S^{-1} \circ Y)\|\boldsymbol{\varphi}(TS\mathbf{x}|X) - \mathbf{D}(\mathbf{R}\mathbf{R}_S)\boldsymbol{\psi}(\mathbf{x}|S^{-1} \circ Y)\|^2 \\
&= \int_{\mathbb{R}^3} d^3\mathbf{x}\rho(\mathbf{x}|S^{-1} \circ Y)\|\boldsymbol{\varphi}(TS\mathbf{x}|X) - \mathbf{D}(\mathbf{R})\mathbf{D}(\mathbf{R}_S)\boldsymbol{\psi}(\mathbf{x}|S^{-1} \circ Y)\|^2 && (\because \text{Equation (1)}) \\
&= \int_{\mathbb{R}^3} d^3\mathbf{x}\rho(S\mathbf{x}|Y)\|\boldsymbol{\varphi}(TS\mathbf{x}|X) - \mathbf{D}(\mathbf{R})\boldsymbol{\psi}(S\mathbf{x}|Y)\|^2 \\
&= \int_{\mathbb{R}^3} d^3(S\mathbf{x})\rho(S\mathbf{x}|Y)\|\boldsymbol{\varphi}(TS\mathbf{x}|X) - \mathbf{D}(\mathbf{R})\boldsymbol{\psi}(S\mathbf{x}|Y)\|^2 && (\because d^3(T\mathbf{x}) = d^3\mathbf{x} \ \forall T \in SE(3)) \\
&= \int_{\mathbb{R}^3} d^3\mathbf{x}\rho(\mathbf{x}|Y)\|\boldsymbol{\varphi}(T\mathbf{x}|X) - \mathbf{D}(\mathbf{R})\boldsymbol{\psi}(\mathbf{x}|Y)\|^2 && (S\mathbf{x} \to \mathbf{x})
\end{aligned}
$$

In the fourth line, we used $\rho(T\mathbf{x}|T \circ Y) = \rho(\mathbf{x}|Y)$ and $\boldsymbol{\psi}(T\mathbf{x}|T \circ Y) = \mathbf{D}(\mathbf{R})\boldsymbol{\psi}(\mathbf{x}|Y)$ by the definition of the query density and the query EDF. Note that in the fifth line we used the $SE(3)$-invariance of the Euclidean volume element $d^3\mathbf{x}$, that is

$$
\begin{aligned}
d^3(T\mathbf{x}) &= \det\left[\partial(\mathbf{R}\mathbf{x} + \mathbf{v})/\partial\mathbf{x}\right]d^3\mathbf{x} \\
&= \det\left[\partial(\mathbf{R}\mathbf{x})/\partial\mathbf{x}\right]d^3\mathbf{x} \\
&= \underbrace{\det\mathbf{R}}\underbrace{\det\mathbf{I}}\, d^3\mathbf{x} = d^3\mathbf{x} \quad \forall T = (\mathbf{R}, \mathbf{v}) \in SE(3)
\end{aligned} \tag{51}
$$

$\square$

Therefore, the energy function $E(T|X, Y)$ in Equation (10) is indeed bi-equivariant.

### F.4 PROOF OF PROPOSITION 4

*Proof.* Let a query density satisfies Equation (11) such that $\rho(\mathbf{x}|Y) = \rho(T\mathbf{x}|T \circ Y) \ \forall T \in SE(3)$. If this query density is grasp-independent such that $\rho(\mathbf{x}|Y) = \rho(\mathbf{x})$, then $\rho(\mathbf{x}) = \rho(T\mathbf{x}) \ \forall T \in SE(3)$ by Equation (11). Since there always exists some $T \in SE(3)$ such that $T\mathbf{x} = \mathbf{x}'$ for any $\mathbf{x}' \in \mathbb{R}^3$, $\rho(\mathbf{x})$ must be a constant function. In other words, there exists no grasp-independent and non-constant query density that satisfies Equation (11). $\square$

### F.5 PROOF OF PROPOSITION 5

*Proof.*

$$
\begin{aligned}
\rho_{\boldsymbol{\theta}}(T\mathbf{x}|T \circ Y) &= \sum_{i=1}^{N_Q} w_{\boldsymbol{\theta}}\left(\mathbf{q}_{i;\boldsymbol{\theta}}(T \circ Y)|T \circ Y\right)\delta^{(3)}\left(T\mathbf{x} - \mathbf{q}_{i;\boldsymbol{\theta}}(T \circ Y)\right) \\
&= \sum_{i=1}^{N_Q} w_{\boldsymbol{\theta}}\left(T\mathbf{q}_{i;\boldsymbol{\theta}}(Y)|T \circ Y\right)\delta^{(3)}\left(T\mathbf{x} - T\mathbf{q}_{i;\boldsymbol{\theta}}(Y)\right) \\
&= \sum_{i=1}^{N_Q} w_{\boldsymbol{\theta}}\left(\mathbf{q}_{i;\boldsymbol{\theta}}(Y)|Y\right)\delta^{(3)}\left(T\mathbf{x} - T\mathbf{q}_{i;\boldsymbol{\theta}}(Y)\right) \\
&= \sum_{i=1}^{N_Q} w_{\boldsymbol{\theta}}\left(\mathbf{q}_{i;\boldsymbol{\theta}}(Y)|Y\right)\delta^{(3)}\left(\mathbf{x} - \mathbf{q}_{i;\boldsymbol{\theta}}(Y)\right) = \rho_{\boldsymbol{\theta}}(\mathbf{x}|Y)
\end{aligned} \tag{52}
$$

where Equation (13) was used in the second and the third lines. $\square$

F.6  PROOF OF PROPOSITION 6

Let the query model $P(\mathbf{w}, \mathbf{Q}|Y)$ be $SE(3)$-equivariant such that

$$P(\mathbf{w}, \mathbf{Q}|Y) = P(\mathbf{w}, S\mathbf{Q}|S \circ Y) \quad \forall S \in SE(3) \tag{53}$$

We first show that $P(T|X, Y, \mathbf{w}, \mathbf{Q})$ satisfies

$$\begin{aligned} P(T|X, Y, \mathbf{w}, \mathbf{Q}) &= P(ST|S \circ X, Y, \mathbf{w}, \mathbf{Q}) \\ &= P(TS|X, S^{-1} \circ Y, \mathbf{w}, S^{-1}\mathbf{Q}) \quad \forall S = (\mathbf{R}_S, \mathbf{v}_S) \in SE(3) \end{aligned} \tag{54}$$

To prove Equation (54), we first show that $\widetilde{E}(T|X, Y, w, \mathbf{q})$ in Equation (15) satisfies the following:

$$\widetilde{E}(ST|S \circ X, Y, w, \mathbf{q}) = \widetilde{E}(T|X, Y, w, \mathbf{q}) = \widetilde{E}(TS|X, S^{-1} \circ Y, w, S^{-1}\mathbf{q})$$

*Proof.* We first prove the left equivariance.

$$\begin{aligned} &\widetilde{E}_{\boldsymbol{\theta}}(ST|S \circ X, Y, w, \mathbf{q}) \\ &= w\|\boldsymbol{\varphi}_{\boldsymbol{\theta}}(ST\mathbf{q}|S \circ X) - \mathbf{D}(\mathbf{R}_S)\mathbf{D}(\mathbf{R})\boldsymbol{\psi}_{\boldsymbol{\theta}}(\mathbf{q}|Y)\|^2 \\ &= w\|\mathbf{D}(\mathbf{R}_S)\boldsymbol{\varphi}_{\boldsymbol{\theta}}(T\mathbf{q}|X) - \mathbf{D}(\mathbf{R}_S)\mathbf{D}(\mathbf{R})\boldsymbol{\psi}_{\boldsymbol{\theta}}(\mathbf{q}|Y)\|^2 &(\because \text{Equation (8)}) \\ &= w\|\boldsymbol{\varphi}_{\boldsymbol{\theta}}(T\mathbf{q}_i|X) - \mathbf{D}(\mathbf{R})\boldsymbol{\psi}_{\boldsymbol{\theta}}(\mathbf{q}_i|Y)\|^2 = \widetilde{E}_{\boldsymbol{\theta}}(T|X, Y, w, \mathbf{q}) &(\because \mathbf{D}(\mathbf{R})^T = \mathbf{D}(\mathbf{R})^{-1}) \end{aligned}$$

We now prove the right equivariance.

$$\begin{aligned} &\widetilde{E}_{\boldsymbol{\theta}}(TS|X, S^{-1} \circ Y, w, S^{-1}\mathbf{q}) \\ &= w\|\boldsymbol{\varphi}_{\boldsymbol{\theta}}(T\cancel{SS^{-1}}\mathbf{q}_i|X) - \mathbf{D}(\mathbf{R})\mathbf{D}(\mathbf{R}_S)\boldsymbol{\psi}_{\boldsymbol{\theta}}(S^{-1}\mathbf{q}_i|S^{-1} \circ Y)\|^2 \\ &= w\|\boldsymbol{\varphi}_{\boldsymbol{\theta}}(T\mathbf{q}_i|X) - \mathbf{D}(\mathbf{R})\cancel{\mathbf{D}(\mathbf{R}_S)}\cancel{\mathbf{D}(\mathbf{R}_S^{-1})}\boldsymbol{\psi}_{\boldsymbol{\theta}}(\mathbf{q}_i|Y)\|^2 &(\because \text{Equation (8)}) \\ &= w\|\boldsymbol{\varphi}_{\boldsymbol{\theta}}(T\mathbf{q}_i|X) - \mathbf{D}(\mathbf{R})\boldsymbol{\psi}_{\boldsymbol{\theta}}(\mathbf{q}_i|Y)\|^2 = \widetilde{E}_{\boldsymbol{\theta}}(T|X, Y, w, \mathbf{q}) &(\because \mathbf{D}(\mathbf{R}^{-1}) = \mathbf{D}(\mathbf{R})^{-1}) \end{aligned}$$

$\square$

One may simply replace the energy function $E(T|X, Y)$ in Appendix F.2 with the new energy function $E(T|X, Y, \mathbf{v}, \mathbf{Q}) = \sum_{i=1}^{N_q} \widetilde{E}(T|X, Y, w_i, \mathbf{q}_i)$ to find that Equation (54) indeed holds.

Now we show that the marginal PDF $P(T|X, Y)$ is bi-equivariant,

*Proof.*

$$\begin{aligned} &P(ST|S \circ X, Y) \\ &= \int d\mathbf{w}d\mathbf{Q} P(ST|S \circ X, Y, \mathbf{w}, \mathbf{Q})P(\mathbf{w}, \mathbf{Q}|Y) \\ &= \int d\mathbf{w}d\mathbf{Q} P(T|X, Y, \mathbf{w}, \mathbf{Q})P(\mathbf{w}, \mathbf{Q}|Y) = P(T|X, Y) \quad (\because \text{Equation (53) and Equation (54)}) \\ &= \int d\mathbf{w}d\mathbf{Q} P(TS|X, S^{-1} \circ Y, \mathbf{w}, S^{-1}\mathbf{Q})P(\mathbf{w}, S^{-1}\mathbf{Q}|S^{-1} \circ Y) \\ &= \int d\mathbf{w}d(S^{-1}\mathbf{Q}) P(TS|X, S^{-1} \circ Y, \mathbf{w}, S^{-1}\mathbf{Q})P(\mathbf{w}, S^{-1}\mathbf{Q}|S^{-1} \circ Y) \quad (\because \text{Equation (51)}) \\ &= \int d\mathbf{w}d\mathbf{Q} P(TS|X, S^{-1} \circ Y, \mathbf{w}, \mathbf{Q})P(\mathbf{w}, \mathbf{Q}|S^{-1} \circ Y) = P(TS|X, S^{-1} \circ Y) \quad (S^{-1}\mathbf{Q} \to \mathbf{Q}) \end{aligned}$$

In the fourth line, we used the $SE(3)$-invariance of the Eulcidean volume element in Equation (51):

$$\int d\mathbf{Q} = \prod_{i=1}^{N_q} \int_{\mathbb{R}^3} d^3\mathbf{q}_i = \prod_{i=1}^{N_q} \int_{\mathbb{R}^3} d^3(T\mathbf{q}_i) = \int d(T\mathbf{Q}) \quad \forall T \in SE(3) \tag{55}$$

$\square$

### F.7 PROOF OF PROPOSITION 7

We first show that $\hat{P}_i(w_i, \mathbf{q}_i|Y)$ in Equation (20) is $SE(3)$-equivariant:

$$\hat{P}_i(w_i, T\mathbf{q}_i|T \circ Y) = \hat{P}_i(w_i, \mathbf{q}_i|Y) \quad \forall T \in SE(3) \tag{56}$$

*Proof.*

$$\begin{aligned}
&\hat{P}_i(w_i, T\mathbf{q}_i|T \circ Y) \\
&= \frac{dl_i}{dw_i}\mathcal{N}(l_i; \log w(T\mathbf{q}_i|T \circ Y), \sigma_H)\delta^{(3)}(T\mathbf{q}_i - \mathbf{q}_i(T \circ Y)) \\
&= \frac{dl_i}{dw_i}\mathcal{N}(l_i; \log w(\mathbf{q}_i|Y), \sigma_H)\delta^{(3)}(\not{T}\mathbf{q}_i - \not{T}\mathbf{q}_i(Y)) \qquad (\because \text{Equation (13))} \\
&= \hat{P}_i(w_i, \mathbf{q}_i|Y)
\end{aligned}$$

$\square$

As a result, $\hat{P}(\mathbf{w}, \mathbf{Q}|Y) = \prod_{i=1}^{N_q} \hat{P}_i(w_i, \mathbf{q}_i|Y)$ in Equation (20) also satisfies

$$\hat{P}(\mathbf{w}, T\mathbf{Q}|T \circ Y) = \hat{P}(\mathbf{w}, \mathbf{Q}|Y) \quad \forall T \in SE(3) \tag{57}$$

We now show that $H_i(w_i, \mathbf{q}_i|X, Y, T)$ in Equation (21) satisfies the following equation:

$$H_i(w_i, \mathbf{q}_i|X, Y, T) = H_i(w_i, \mathbf{q}_i|S \circ X, Y, ST) = H_i(w_i, S^{-1}\mathbf{q}_i|X, S^{-1} \circ Y, TS) \tag{58}$$

*Proof.*

$$\begin{aligned}
&H_i(w_i, \mathbf{q}_i|S \circ X, Y, ST) \\
&= \begin{cases} \hat{P}_i(w_i, \mathbf{q}_i|Y) & \text{if } d_{min}(ST\mathbf{q}_i, S \circ X) < r \\ (dl_i/dw_i)\mathcal{N}(l_i; \alpha, \sigma_H)\delta^{(3)}(\mathbf{q}_i - \mathbf{q}_i(Y)) & \text{else} \end{cases} \\
&= \begin{cases} \hat{P}_i(w_i, \mathbf{q}_i|Y) & \text{if } d_{min}(T\mathbf{q}_i, X) < r \\ (dl_i/dw_i)\mathcal{N}(l_i; \alpha, \sigma_H)\delta^{(3)}(\mathbf{q}_i - \mathbf{q}_i(Y)) & \text{else} \end{cases} \tag{A} \\
&= H_i(w_i, \mathbf{q}_i|X, Y, T) \\
&= \begin{cases} \hat{P}_i(w_i, S^{-1}\mathbf{q}_i|S^{-1} \circ Y) & \text{if } d_{min}\left(T(SS^{-1})\mathbf{q}_i, X\right) < r \\ (dl_i/dw_i)\mathcal{N}(l_i; \alpha, \sigma_H)\delta^{(3)}(S^{-1}\mathbf{q}_i - S^{-1}\mathbf{q}_i(Y)) & \text{else} \end{cases} \tag{B} \\
&= \begin{cases} \hat{P}_i(w_i, S^{-1}\mathbf{q}_i|S^{-1} \circ Y) & \text{if } d_{min}\left((TS)(S^{-1}\mathbf{q}_i), X\right) < r \\ (dl_i/dw_i)\mathcal{N}(l_i; \alpha, \sigma_H)\delta^{(3)}(S^{-1}\mathbf{q}_i - \mathbf{q}_i(S^{-1} \circ Y)) & \text{else} \end{cases} \tag{C} \\
&= H_i(w_i, S^{-1}\mathbf{q}_i|X, S^{-1} \circ Y, TS)
\end{aligned}$$

We used $d_{min}(T\mathbf{x}, T \circ X) = d_{min}(\mathbf{x}, X) \ \forall T \in SE(3)$ in (A). This is because the Euclidean distance is preserved under $SE(3)$ transformations. We used $\delta^3(\mathbf{x}_1 - \mathbf{x}_2) = \delta^3(T\mathbf{x}_1 - T\mathbf{x}_2) \ \forall T \in SE(3)$ and Equation (56) in (B). Lastly, we used Equation (13) in (C). $\square$

Therefore, $H(\mathbf{w}, \mathbf{Q}|X, Y, T) = \prod_{i=1}^{N_q} H_i(w_i, \mathbf{q}_i|X, Y, T)$ also satisfies

$$H(\mathbf{w}, \mathbf{Q}|X, Y, T) = H(\mathbf{w}, \mathbf{Q}|SX, Y, ST) = H(\mathbf{w}, S^{-1}\mathbf{Q}|X, S^{-1} \circ Y, TS) \tag{59}$$

We now propose the following lemma.

**Lemma 4.** *Let a scalar function $f(T|X, Y)$ be defined as follows:*

$$f(T|X, Y) = \int d\mathbf{w} \int d\mathbf{Q} \ h_1(T, X, Y, \mathbf{w}, \mathbf{Q})h_2(T, X, Y, \mathbf{w}, \mathbf{Q})$$

*$f(T|X, Y)$ is bi-equivariant if $h_1(T, X, Y, \mathbf{w}, \mathbf{Q})$ and $h_2(T, X, Y, \mathbf{w}, \mathbf{Q})$ satisfies*

$$\begin{aligned}
h_i(T, X, Y, \mathbf{w}, \mathbf{Q}) &= h_i(ST, S \circ X, Y, \mathbf{w}, \mathbf{Q}) \\
&= h_i(TS, X, S^{-1} \circ Y, \mathbf{w}, S^{-1}\mathbf{Q}) \quad \forall S \in SE(3), i \in \{1, 2\}
\end{aligned} \tag{60}$$

*Proof.*

$$f(ST|SX, Y)$$

$$= \int d\mathbf{w} \int d\mathbf{Q} \, h_1(ST, SX, Y, \mathbf{w}, \mathbf{Q}) h_2(ST, SX, Y, \mathbf{w}, \mathbf{Q})$$

$$= \int d\mathbf{w} \int d\mathbf{Q} \, h_1(T, X, Y, \mathbf{w}, \mathbf{Q}) h_2(T, X, Y, \mathbf{w}, \mathbf{Q}) = f(T|X, Y) \qquad (\because \text{Equation (60)})$$

$$= \int d\mathbf{w} \int d\mathbf{Q} \, h_1(TS, X, S^{-1} \circ Y, \mathbf{w}, S^{-1}\mathbf{Q}) h_2(TS, X, S^{-1} \circ Y, \mathbf{w}, S^{-1}\mathbf{Q}) \quad (\because \text{Equation (60)})$$

$$= \int d\mathbf{w} \int d(S^{-1}\mathbf{Q}) \, h_1(TS, X, S^{-1} \circ Y, \mathbf{w}, S^{-1}\mathbf{Q}) h_2(TS, X, S^{-1} \circ Y, \mathbf{w}, S^{-1}\mathbf{Q})$$
$$\qquad\qquad\qquad (\because \text{Equation (55)})$$

$$= \int d\mathbf{w} \int d\mathbf{Q} \, h_1(TS, X, S^{-1} \circ Y, \mathbf{w}, \mathbf{Q}) h_2(TS, X, S^{-1} \circ Y, \mathbf{w}, \mathbf{Q}) \qquad (S^{-1}\mathbf{Q} \to \mathbf{Q})$$

$$= f(TS|X, S^{-1}Y)$$

$\square$

We now prove the bi-equivariance of $\mathcal{L}_{\boldsymbol{\theta}}(T|X, Y)$ in Equation (22).

*Proof.* We first define $h(T, X, Y, \mathbf{w}, \mathbf{Q})$ as follows:

$$h(T, X, Y, \mathbf{w}, \mathbf{Q}) = \log P_{\boldsymbol{\theta}}(T|X, Y, \mathbf{w}, \mathbf{Q}) + \hat{P}_{\boldsymbol{\theta}}(\mathbf{w}, \mathbf{Q}|Y) - H_{\boldsymbol{\theta}}(\mathbf{w}, \mathbf{Q}|X, Y, T) \qquad (61)$$

Using Equation (54), Equation (57) and Equation (59), one can prove that $h(T, X, Y, \mathbf{w}, \mathbf{Q})$ satisfies Equation (60). In addition, $H_{\boldsymbol{\theta}}(\mathbf{w}, \mathbf{Q}|X, Y, T)$ satisfies Equation (60) as was shown in Equation (59). Because $\mathcal{L}_{\boldsymbol{\theta}}(T|X, Y)$ can be written as

$$\mathcal{L}_{\boldsymbol{\theta}}(T|X, Y) = \mathbb{E}_{\mathbf{w}, \mathbf{Q} \sim H_{\boldsymbol{\theta}}} \left[ \log P_{\boldsymbol{\theta}}(T|X, Y, \mathbf{w}, \mathbf{Q}) \right] - D_{KL} \left[ H_{\boldsymbol{\theta}}(\mathbf{w}, \mathbf{Q}|X, Y, T) \, \Big\| \, \hat{P}_{\boldsymbol{\theta}}(\mathbf{w}, \mathbf{Q}|Y) \right]$$
$$= \int d\mathbf{w} \int d\mathbf{Q} \, H_{\boldsymbol{\theta}}(\mathbf{w}, \mathbf{Q}|X, Y, T) h(T, X, Y, \mathbf{w}, \mathbf{Q})$$
$$(62)$$

we prove the bi-equivariance of $\mathcal{L}_{\boldsymbol{\theta}}(T|X, Y)$ in Equation (22) using Lemma 4. $\square$

## G  EQUIVARIANT GRAPH NEURAL NETWORKS

Graph neural networks are often used to model point cloud data (Wang et al., 2019; Te et al., 2018; Shi & Rajkumar, 2020). $SE(3)$-equivariant graph neural networks (Thomas et al., 2018; Fuchs et al., 2020; Liao & Smidt, 2022) exploit the roto-translation symmetry of graphs with spatial structures. In this work, we use Tensor Field Networks (TFNs) (Thomas et al., 2018) and the SE(3)-transformers (Fuchs et al., 2020) as the backbone networks for our models.

**Tensor Product and Spherical Harmonics**  Given two vectors $\mathbf{u}$ and $\mathbf{v}$ of type-$l_1$ and -$l_2$, the tensor product $\mathbf{u} \otimes \mathbf{v}$ transforms according to a rotation $\mathbf{R} \in SO(3)$ as

$$\mathbf{u} \otimes \mathbf{v} \to (\mathbf{D}_{l_1}(\mathbf{R})\mathbf{u}) \otimes (\mathbf{D}_{l_2}(\mathbf{R})\mathbf{v}) \qquad (63)$$

Tensor products are important because they can be used to construct new vectors of different types. By a change of basis the tensor product $\mathbf{u} \otimes \mathbf{v}$ can be decomposed into the direct sum of type-$l$ vectors using the *Clebsch-Gordan coefficients* (Thomas et al., 2018; Zee, 2016; Griffiths & Schroeter, 2018). Let this type-$l$ vector be $(\mathbf{u} \otimes \mathbf{v})^{(l)}$. The $m$'th components of this vector is calcuated as:

$$(\mathbf{u} \otimes \mathbf{v})_m^{(l)} = \sum_{m_1 = -l_1}^{l_1} \sum_{m_2 = -l_2}^{l_2} C_{(l_1, m_1)(l_2, m_2)}^{(l, m)} u_{m_1} v_{m_2} \qquad (64)$$

where $C_{(l_1, m_1)(l_2, m_2)}^{(l, m)}$ is the Clebsch-Gordan coefficients in real basis, which can be nonzero only for $|l_1 - l_2| \le l \le l_1 + l_2$.

The *(real) spherical harmonics* $Y_m^{(l)}(\mathbf{x}/\|\mathbf{x}\|)$ are orthonormal functions that form the complete basis of the Hilbert space on the sphere $S^2$. $l \in \{0, 1, 2, \cdots\}$ is called the *degree* and $m \in \{-l, \cdots, l\}$ is called the *order* of the spherical harmonic function.

Consider the following $(2l + 1)$-dimensional vector field $\mathbf{Y}^{(l)} = (Y_{m=-l}^l, \cdots, Y_{m=l}^l)$. By a 3-dimensional rotation $\mathbf{R} \in SO(3)$, $\mathbf{Y}^{(l)}$ transforms like a type-$l$ vector field such that

$$\mathbf{Y}^{(l)}(\mathbf{R}(\mathbf{x}/\|\mathbf{x}\|)) = \mathbf{D}_l(\mathbf{R})\mathbf{Y}^{(l)}(\mathbf{x}/\|\mathbf{x}\|) \tag{65}$$

**Tensor Field Networks**  Tensor field networks (TFNs) (Thomas et al., 2018) are $SE(3)$-equivariant models for generating representation-theoretic vector fields from a point cloud input. TFNs construct equivariant output feature vectors from equivariant input feature vectors and spherical harmonics. Spatial convolutions and tensor products are used for the equivariance.

Consider a featured point cloud input with $M$ points given by $X = \{(\mathbf{x}_1, \mathbf{f}_1), \cdots, (\mathbf{x}_M, \mathbf{f}_M)\}$ where $\mathbf{x}_i \in \mathbb{R}^3$ is the position and $\mathbf{f}_i$ is the equivariant feature vector of the $i$-th point. Let $\mathbf{f}_i$ be decomposed into $N$ vectors such that $\mathbf{f}_i = \bigoplus_{n=1}^N \mathbf{f}_i^{(n)}$, where $\mathbf{f}_i^{(n)}$ is a type-$l_n$ vector, which is $(2l_n + 1)$ dimensional. Therefore, we define the action of $T = (\mathbf{R}, \mathbf{v}) \in SE(3)$ on $X$ as

$$T \circ X = \{(T\mathbf{x}_1, \mathbf{D}(\mathbf{R})\mathbf{f}_1), \cdots, (T\mathbf{x}_M, \mathbf{D}(\mathbf{R})\mathbf{f}_M)\}$$

where $\mathbf{R} \in SO(3), \mathbf{v} \in \mathbb{R}^3$ and $\mathbf{D}(\mathbf{R}) = \bigoplus_{n=1}^N \mathbf{D}_{l_n}(\mathbf{R})$.

Consider the following input feature field $\mathbf{f}_{(in)}(\mathbf{x}|X)$ generated by the point cloud input $X$ as

$$\mathbf{f}_{(in)}(\mathbf{x}|X) = \sum_{j=1}^M \mathbf{f}_j \delta^{(3)}(\mathbf{x} - \mathbf{x}_j) \tag{66}$$

where $\delta^{(3)}(\mathbf{x} - \mathbf{y}) = \prod_{\mu=1}^3 \delta(x_\mu - y_\mu)$ is the three-dimensional Dirac delta function centered at $\mathbf{x}_j$. Note that this input feature field is an $SE(3)$-equivariant field, that is:

$$\mathbf{f}_{(in)}(T\mathbf{x}|T \circ X) = \mathbf{D}(\mathbf{R})\mathbf{f}_{(in)}(\mathbf{x}|X) \quad \forall\, T = (\mathbf{R}, \mathbf{v}) \in SE(3)$$

Now consider the following output feature field by a convolution

$$\begin{aligned} \mathbf{f}_{(out)}(\mathbf{x}|X) &= \bigoplus_{n'=1}^{N'} \mathbf{f}_{(out)}^{(n')}(\mathbf{x}|X) \\ &= \int d^3\mathbf{y}\, \mathbf{W}(\mathbf{x} - \mathbf{y})\mathbf{f}_{(in)}(\mathbf{y}|X) = \sum_j \mathbf{W}(\mathbf{x} - \mathbf{x}_j)\mathbf{f}_j \end{aligned} \tag{67}$$

with the convolution kernel $\mathbf{W}(\mathbf{x} - \mathbf{y}) \in \mathbb{R}^{dim(\mathbf{f}_{(out)}) \times dim(\mathbf{f}_{(in)})}$ whose $(n', n)$-th block $\mathbf{W}^{n'n}(\mathbf{x} - \mathbf{y}) \in \mathbb{R}^{(2l_{n'}+1) \times (2l_n+1)}$ is defined as follows:

$$\left[\mathbf{W}^{n'n}(\mathbf{x})\right]_{m'm} = \sum_{J=|l_{n'}-l_n|}^{l_{n'}+l_n} \phi_J^{n'n}(\|\mathbf{x}\|) \sum_{k=-J}^J C_{(J,k)(l_n,m)}^{(l_{n'},m')} Y_k^{(J)}(\mathbf{x}/\|\mathbf{x}\|) \tag{68}$$

Here, $\phi_J^{n'n}(\|\mathbf{x}\|) : \mathbb{R} \to \mathbb{R}$ is some learnable radial function. The output feature field $\mathbf{f}_{(out)}(\mathbf{x}|X)$ in Equation (67) is proven to be $SE(3)$-equivariant (Thomas et al., 2018; Fuchs et al., 2020).

**SE(3)-Transformers**  The $SE(3)$-Transformers (Fuchs et al., 2020) are variants of TFNs with self-attention. Consider the case in which the output field is also a featured sum of Dirac deltas

$$\mathbf{f}_{(out)}(\mathbf{x}|X) = \sum_{j=1}^M \mathbf{f}_{(out),j} \delta^{(3)}(\mathbf{x} - \mathbf{x}_j) \tag{69}$$

where $\mathbf{x}_i$ is the same point as that of the point cloud input $X$. The SE(3)-Transformers apply type-0 (scalar) self-attention $\alpha_{ij}$ to Equation (67):

$$\mathbf{f}_{(out),i} = \sum_{j \neq i} \alpha_{ij} \mathbf{W}(\mathbf{x} - \mathbf{x}_j)\mathbf{f}_j + \bigoplus_{n'}^{N'} \sum_{n=1}^N \mathbf{W}_{(S)}^{n'n} \mathbf{f}_j^{(n)} \tag{70}$$

where the $\mathbf{W}_{(S)}^{n'n}$ term is called the *self-interaction* (Thomas et al., 2018). $\mathbf{W}_{(S)}^{n'n}$ is nonzero only when $l'_n = l_n$. The self-interaction occurs where $i = j$ such that $\mathbf{W}(x_i - x_j) = W(0)$. The self-interaction term is needed because $\mathbf{W}$ is a linear combination of the spherical harmonics, which are not well defined in $\mathbf{x} = 0$. Details about the calculation of the self-attention $\alpha_{ij}$ can be found in Fuchs et al. (2020).

## H    EXPERIMENTAL DETAILS

For the test environment, we use PyBullet (Coumans & Bai, 2016–2021) simulator for the experiments. We use the Franka Panda manipulator with a custom end-effector. We use IKFast (Diankov, 2010) with Pybullet-Planning (Garrett, 2018) for the inverse kinematics. Three simulated depth cameras are used to observe the point cloud of the scene. Six simulated depth cameras are used to observe the point cloud of the grasp. We downsample the point clouds using a voxel filter. We illustrate the downsampled point clouds in Figure 9. Since motion planning is not in the scope of our work, we assume no collision between the environment and the robot links except for the hand link. We also allow the robot to teleport to reach pre-grasp and pre-place poses to eliminate the unnecessary influence of the motion planners. However, we fully simulate the trajectories of all the task-relevant primitives (e.g., grasping, releasing, lifting).

We experiment with three tasks, the mug-hanging task and the bowl/bottle pick-and-place task. For the mug hanging task and the bowl pick-and-place task, we demonstrate with a single target object instance in upright poses only. For the bottle pick-and-place task, we use five object instances in upright poses only. For evaluations, we tested in four different unseen setups: (A) Unseen instances, (B) Unseen poses, (C) Unseen distractors, and (D) Unseen poses, instances, and distractors. Note that in the unseen poses setup (B), we only use lying poses, which were not presented during the training. On the other hand, in the unseen poses, instances, and distractors setup (D), we both use lying and upright poses but in unseen elevations. The reason why we also use upright poses in setup (D) is that the baseline model already completely fails in setup (B). Therefore, the result would be trivial if we only use lying poses in (D). Therefore, we mix both upright and lying poses in setup (D). Note that the upright poses are also unseen poses because of the unseen elevations.

For the inference, we run the MH for 1000 steps and then run the Langevin algorithm for 300 steps. Lastly, we optimize the samples for 100 steps using Equation (46) but without noise. We empirically find that only at most three query points have significant weights after training. Therefore, we only use three query points with the highest weights to save computations. Instead of directly taking the lowest-energy sample pose, we check the feasibility of the pose before going into action. For example, if a collision is found or no inverse kinematics solution can be found for the sample pose, we deny that pose and move to the next best sample. We provide the details in Algorithm 2.

Ten demonstrations are generated by a probabilistic oracle for each task, with the default instances in upright poses, with only the $x, y$, and yaw being randomized. In the unseen poses setup, the default (trained) instances are provided in unseen (lying) poses. The poses are completely randomized, including the elevation $z$. In the unseen instances setup, ten unseen instances of target objects are provided in the trained poses (upright), again with only the $x, y$, and yaw being randomized. In the unseen distractors setup, four unseen visual distractors are located near the target objects. We randomize the poses and the colors of these distractors. To separate the effect of motion planning, we disable the collision between the distractors and the robot. Lastly, in the unseen instances, poses, and distractors setup, we combine all the prior setups. We experiment with ten unseen instances in unseen poses (50% upright and 50% lying, arbitrary elevation) with four randomized visual distractors. In this setup, we use supports to give arbitrary elevation to the target objects. Note that we both used upright and lying poses, unlike the unseen poses setup. This is to test the case with unseen distractors and unseen instances for not only the lying poses but the upright poses as well. Note that upright poses are also unseen poses because we give arbitrary elevations. For the mug and bowl tasks, only a single instance was used as the default instance for training. For the bottle task, five instances were used due to the high variance of the shape. All the models are sufficiently trained such that the total success rate in the trained setup (no unseen situations) exceed at least 90% for all three tasks. The experimental settings for the three tasks are illustrated in Figure 10. We also illustrate the ten unseen instances of mug, bowl, and bottle in Figures 11, 12, and 13, respectively.

---

**Algorithm 2** Pick-and-place algorithm

---

**Input:** Pick-model $P_{pick}(T|X)$, Place-model $P_{place}(T|X, Y)$, Number of iterations $N$

    $X \leftarrow \text{observe\_scene}()$                     ▷ Observe the point cloud of the scene

    $[T_i]_{i=1}^N \leftarrow \text{sample}(P_{pick}(\cdot|X), N)$         ▷ Sample $N$ poses from the pick-model

    $[T_i]_{i=1}^N \leftarrow \text{sort}\left([T_i]_{i=1}^N, P_{pick}(\cdot|X)\right)$   ▷ Sort samples by their probabilities (descending order)

    **for** $T$ **in** $[T_i]_{i=1}^N$ **do**

        **if** feasible$(T)$ **then**

            pick$(T)$                          ▷ Pick if the configuration is feasible

            **break**

        **end if**

    **end for**

    $X \leftarrow \text{observe\_scene}()$                     ▷ Observe the point cloud of the scene

    $Y \leftarrow \text{observe\_gripper}()$                ▷ Observe the point cloud of the gripper

    $[T_i]_{i=1}^N \leftarrow \text{sample}(P_{place}(\cdot|X, Y), N)$     ▷ Sample $N$ poses from the place-model

    $[T_i]_{i=1}^N \leftarrow \text{sort}\left([T_i]_{i=1}^N, P_{place}(\cdot|X, Y)\right)$ ▷ Sort samples by their probabilities (descending order)

    **for** $T$ **in** $[T_i]_{i=1}^N$ **do**

        **if** feasible$(T)$ **then**

            place$(T)$                      ▷ Place if the configuration is feasible

            **break**

        **end if**

    **end for**

---

For EDFs, we use a single query point for the pick inference and three query points for the place inference. For the ablated model, we use ten query points for the pick inference and five query points for the place inference. The reason we use more query points for the ablated model is that the type-$0$ descriptors cannot encode orientations alone. Unlike EDFs, type-$0$ descriptor fields (the ablated model) require at least three non-collinear query points and much more in practice to determine orientations. This is in direct comparison with EDFs, which can determine the orientations even with a single point. The computational benefit of using only type-$0$ descriptors is compensated by the increased number of query points. We set the number of query points to make the inference time of the ablated model to be similar to or slightly longer than EDFs. We run all the experiments on an Nvidia RTX3090 GPU and an Intel i9-12900k CPU with 16Gb RAM. We turned off all the E-cores of the CPU and only used P-cores with a fixed clock of 5100Mhz. We found that turning off the E-core is crucial for the inference speed.

The models were trained for 600 steps (60 epochs) using Adam optimizer (Kingma & Ba, 2014) where the learning rates range from $0.005$ to $0.001$. We randomly perturb the target pose and apply jitters on input point clouds to augment the training data. It takes around 5.5 hours to train the pick-model and 8.5 hours to train the place-model, where most of the time is spent on MCMC sampling. We run 10000 iterations of the MH and 3000∼6000 iterations (linearly increasing as training proceeds) of the Langevin algorithm to draw negative samples for the training.

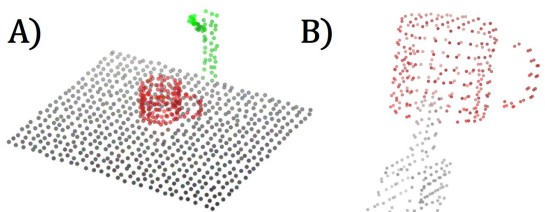

Figure 9: A) Downsampled point cloud of the scene. B) Downsampled point cloud of the gripper with a grasped object.

A) Demonstrations  B) Unseen Instances  C) Unseen Poses  D) Unseen Distractors  E) Unseen Instances, Poses, & Distractors

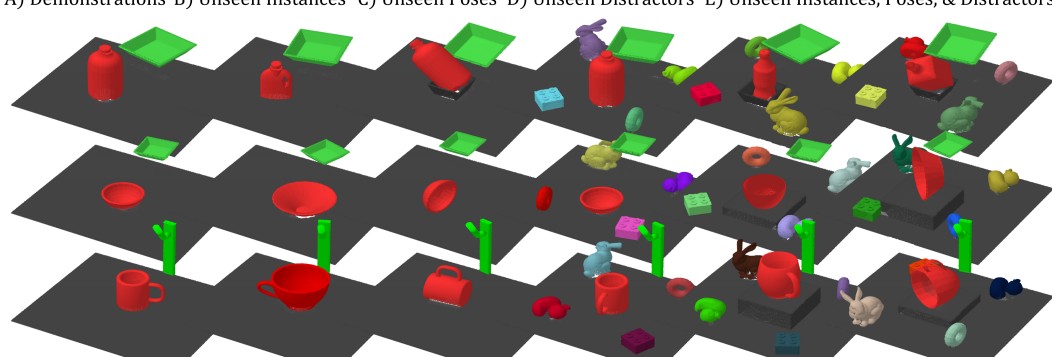

Figure 10: Test environment for the mug-hanging task, the bowl pick-and-place task, and the bottle pick-and-place task with various unseen setups.

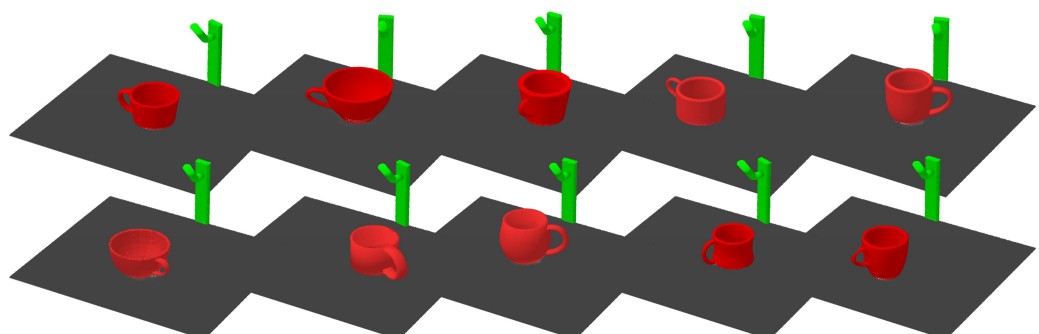

Figure 11: The ten mug instances that were used as unseen mug instances are illustrated.

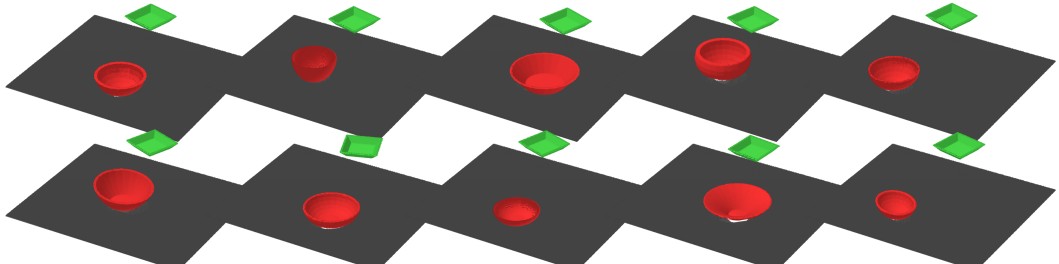

Figure 12: The ten bowl instances that were used as unseen bowl instances are illustrated.

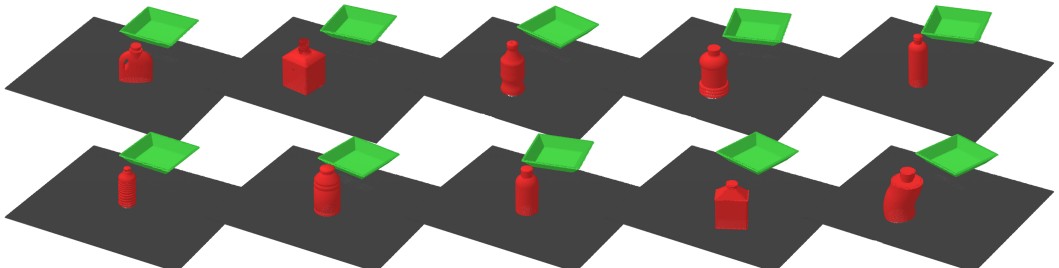

Figure 13: The ten bottle instances that were used as unseen bottle instances are illustrated.

# I  ADDITIONAL EXPERIMENTAL RESULTS

Table 3: Success rate of EDFs for mug-hanging task with different demonstrations

| | Low Var. & Unimodal Demo | | | Mixed Grasp (Handle & Rim) | | |
|---|---|---|---|---|---|---|
| Setup | Pick | Place | Total | Pick | Place | Total |
| Unseen Poses (P) | 1.00 | 0.96 | 0.96 | 1.00 | 0.99 | 0.99 |
| Unseen Instances (I) | 0.99 | 0.90 | 0.89 | 1.00 | 0.92 | 0.92 |
| Unseen Distractors (D) | 1.00 | 1.00 | 1.00 | 0.96 | 0.99 | 0.95 |
| Unseen P+I+D | 0.99 | 0.83 | 0.82 | 0.90 | 0.89 | 0.80 |

In Table 3, we list the success rates of EDFs for (1) low variance and unimodal demonstrations and (2) highly multimodal demonstrations for the mug-hanging task. In the highly multimodal demonstrations, the mug is picked both using the rim grasp and the handle grasp. The experimental results indicate that EDFs are both robust to low variance or high variance demonstrations, whereas $SE(3)$-TNs can only be trained from low variance ones.

Table 4: Success rate and inference time of the ablated model and EDFs. All the evaluations are done in the *unseen instances, poses & distracting objects* setting.

| Descriptor Type | Mug | | | Bowl | | | Bottle | | |
|---|---|---|---|---|---|---|---|---|---|
| (# Pick Query / # Place Query) | Pick | Place | Total | Pick | Place | Total | Pick | Place | Total |
| **Type-0 Only** (10 / 5) | | | | | | | | | |
| Inference Time | 5.7s | 8.6s | 14.3s | 6.1s | 9.9s | 16.0s | 5.8s | 17.3s | 23.0s |
| Success Rate | 0.84 | 0.77 | 0.65 | 0.60 | 0.95 | 0.57 | 0.66 | 0.95 | 0.63 |
| **Type-0 Only** (30 / 10) | | | | | | | | | |
| Inference Time | 10.0s | 14.5s | 24.5s | 13.9s | 19.4s | 33.4s | 10.9s | 24.2s | 35.1s |
| Success Rate | 0.90 | 0.90 | 0.81 | 0.72 | 0.99 | 0.71 | 0.76 | 0.96 | 0.73 |
| **EDFs** (1 / 3) | | | | | | | | | |
| Inference Time | 5.1s | 8.3s | 13.4s | 5.2s | 10.4s | 15.6s | 5.2s | 11.5s | 16.7s |
| Success Rate | **1.00** | **0.95** | **0.95** | **0.95** | **1.00** | **0.95** | **0.95** | **1.00** | **0.95** |

In Table 4, we compare EDFs with the ablated models with only type-0 descriptors. We evaluate the ablated model for different query point numbers. The result shows that EDFs still outperform the ablated model even though much more query points and longer inference time is used.

Table 5: Success rate of experimented methods in the trained setups

| | Mug | | | Bowl | | | Bottle | | |
|---|---|---|---|---|---|---|---|---|---|
| Method | Pick | Place | Total | Pick | Place | Total | Pick | Place | Total |
| EDFs | 1.00 | 0.99 | 0.99 | 1.00 | 1.00 | 1.00 | 0.98 | 1.00 | 0.98 |
| SE(3)-TNs (Zeng et al., 2020) | 1.00 | 0.91 | 0.91 | 1.00 | 1.00 | 1.00 | 0.99 | 0.93 | 0.92 |
| Type-0 Only (Fast) | 1.00 | 0.98 | 0.98 | 1.00 | 1.00 | 1.00 | 1.00 | 0.97 | 0.97 |
| Type-0 Only (Slow) | 1.00 | 0.98 | 0.98 | 1.00 | 1.00 | 1.00 | 1.00 | 1.00 | 1.00 |

Lastly, in Table 5, we list the success rate of all the methods in the trained setup. Note that all the methods used in the experiments were sufficiently trained to achieve at least 91% total success rate.

## J EXPERIMENTAL RESULTS ON NEURAL DESCRIPTOR FIELDS WITH UNSEGMENTED POINT CLOUD INPUTS

In this section, we show by experiment that object segmentation is critical to the performance of Neural Descriptor Fields (NDFs) (Simeonov et al., 2021). We compare the success rate of NDFs with unsegmented point cloud input to the same NDFs with segmented point cloud input. All the experiments are done using the official implementations of Simeonov et al. (2021). The results are summarized in Table 6. As can be seen in Table 6, the performances significantly drop for both tasks when NDFs are provided with unsegmented inputs. We provide qualitative examples in Figure 14. Therefore, it can be concluded that object segmentation is essential for NDFs.

Table 6: Success rate of NDFs with and without object segmentation

| | Bottle | | | Mug | | |
|---|---|---|---|---|---|---|
| **Object Segmentation** | Pick | Place | **Total** | Pick | Place | **Total** |
| With Segmentation | 0.94 | 0.86 | 0.81 | 0.97 | 0.72 | 0.70 |
| Without Segmentation | 0.00 | 0.00 | 0.00 | 0.37 | 0.00 | 0.00 |
| **Difference** | 0.94 | 0.86 | 0.81 | 0.60 | 0.72 | 0.70 |

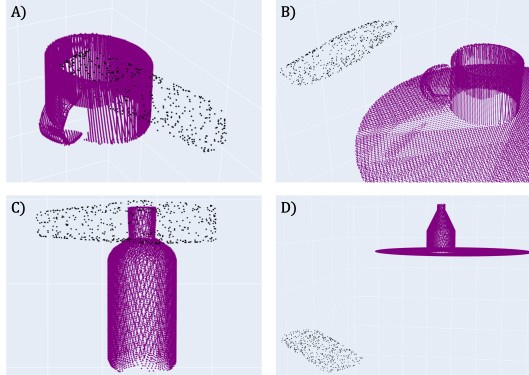

Figure 14: A) NDFs successfully infer the pick position of a well-segmented mug point cloud. B) NDFs fail to successfully infer the pick position of an unsegmented mug point cloud. C) NDFs successfully infer the pick position of a well-segmented bottle point cloud. D) NDFs fail to successfully infer the pick position of an unsegmented bottle point cloud. The black dots are the query points attached to the gripper.

