# OpenReview forum: "Equivariant Descriptor Fields: SE(3)-Equivariant Energy-Based Models for End-to-End Visual Robotic Manipulation Learning"
_ICLR.cc/2023/Conference — ICLR 2023 poster_

### Official Review · Reviewer_R4yz · 2022-10-22

**Confidence:** 3
**Correctness:** 2
**Technical Novelty And Significance:** 3
**Empirical Novelty And Significance:** 1
**Recommendation:** 6

**Clarity, Quality, Novelty And Reproducibility:**

- Clarity: just-so-so
  - Not quite friendly for people of non-mathematic background. Why Lie-group? Why it is bi-equivariant? etc.
  - Non-math part is clear.
- Quality: needs improvement
  - No supportive discussions;
  - Too naive experiments.
Novelty: good
  - If the advantage is well proved in mathematical proof or well-defined experiment, it is important and useful for not only robot manipulation, but many domains.
  - Designs to solve this problem are novel and technically sound.
- Reproducibility
  - Code submitted and said reproducible but not checked.

**Strength And Weaknesses:**

- Strength
  - The paper is well written, at least non-math part.
  - The problem formulation is good and the designed method seems technically sound.
- Weaknesses
  - As an non-math background reader, I cannot find supportive discussion on why this new approach can achieve such advantages (very few training examples and good generalization method).
  - Even though it is normal practice as a theoretical paper, the experiment is too naive.
    - Only two tasks are tested, it is hard to prove anything.
    - With that number of data (5-10) and smaller than 5% difference, I do not think any conclusion can be made. The author should at least do the same experiments multiple times to provide confidential interval for accuracy.
    - You should provide more background research related to EBM, especially EBM on controlling. Such as [1] and its following paper; [2].

--- after rebuttal ---
readability improved. Experiments added. Background added. Problem addressed. I raise the score.

- Question:
  - Time cost is not provided but EBM must be incredibly slow. Is it correct? Is efficiency a bottleneck for scalability? Both in training and applying.
  - IDF is SE(3)-invar. , NDF is SE(3)-equivar. Why IDC can  be understood as an end-to-end trainable version? Appendix E do not have a clear explanation.

[1] Xie, Jianwen, et al. "A theory of generative convnet." International Conference on Machine Learning. PMLR, 2016.
[2] Y. Xu, et al.  "Energy-Based Continuous Inverse Optimal Control," in IEEE Transactions on Neural Networks and Learning Systems, doi: 10.1109/TNNLS.2022.3168795.

**Summary Of The Paper:**

This paper proposed Equivariant Descriptor Fields (EDF), a generalized, incredibly low-data-needed, fully end-to-end-training model which is SE(3)-equivariant (this means apply 3D translation and rotation to input will results to the same as it is applied to the output).

Authors utilized the representation theory of Lie groups and designed a novel SE(3)-equivariant energy-based model. It has high sample efficiency. It can also be learned from scratch using a few data (5-10 demonstrations). The EBM used MCMC-sample-based MLE to train.

They experiment EDF on 6-DoF robotic manipulation tasks (pick-and-place cups and etc.). Compared with TN (an SE(2)-equivariant method) and IDF (an SE(3)-invariant method, similar to NDF), EDF showed its sample efficiency and generalizability.

**Summary Of The Review:**

This paper designed a good method to solve an important problem. Multiple advantages are presented but not validated. It lacks supportive discussions and experiments.

Since I do not have a very strong math background, I am not able to access the correctness and novelty of this paper from the perspective of math. If a math guy evaluated this paper as perfect to be accepted. Just ignore my weakness assessment.

---

> ### Author Response · Authors · 2022-11-19
> **Response to Reviewer R4yz (1/4)**
>
> First, we would like to thank the reviewer for the helpful comments.
>
> > This paper proposed Equivariant Descriptor Fields (EDF), a generalized, incredibly low-data-needed, fully end-to-end-training model which is SE(3)-equivariant (this means apply 3D translation and rotation to input will results to the same as it is applied to the output). Authors utilized the representation theory of Lie groups and designed a novel SE(3)-equivariant energy-based model. It has high sample efficiency. It can also be learned from scratch using a few data (5-10 demonstrations). The EBM used MCMC-sample-based MLE to train. They experiment EDF on 6-DoF robotic manipulation tasks (pick-and-place cups and etc.). Compared with TN (an SE(2)-equivariant method) and IDF (an SE(3)-invariant method, similar to NDF), EDF showed its sample efficiency and generalizability.
>
> **Response:** We thank the reviewer for the nice summary of our work.
>
> > - The paper is well written, at least non-math part.
> > - The problem formulation is good and the designed method seems technically sound.
> > - Designs to solve this problem are novel and technically sound.
> > - This paper designed a good method to solve an important problem.
>
> **Response:** We sincerely thank the reviewer for appreciating our problem formulation and technicalities.
>
> > - As an non-math background reader, I cannot find supportive discussion on why this new approach can achieve such advantages (very few training examples and good generalization method).
> > - Not quite friendly for people of non-mathematic background. Why Lie-group? Why it is bi-equivariant? etc.
>
> **Response:** We appreciate the author for pointing out the lack of supportive discussions and friendly introductions in our paper. First, we would like to remind the reviewer that in Appendix C, we have intuitively introduced why bi-equivariance is desirable. However, we find that this explanation is still too mathematical. To address this issue, in the revised manuscript, we also explain the bi-equivariance condition in a plain word. The revised part is as follows. "..., that is, the policy is not only equivariant to the object but also equivariant to where the object has to be placed."
>
> Please note that we discuss little on why SE(3)-equivariance is important for the sample efficiency because this topic has already been addressed well by previous works, thus, has less novelty. Please note that our work is built upon Neural Descriptor Fields and Transporter Networks, both of which exploit equivariance to achieve highly sample-efficient learning for visual manipulation tasks. These works have already addressed why equivariance is important for sample efficiency.  Therefore, instead of discussing much on why SE(3)-equivariance is important, we spent more pages on *how to achieve SE(3)-equivariance while not compromising other important qualities like end-to-end trainability and background invariance*.
>
> > - Even though it is normal practice as a theoretical paper, the experiment is too naive.
> > - Only two tasks are tested, it is hard to prove anything.
> > - Too naive experiments. Novelty: good
>
> **Response:** We fully appreciate the reviewer's concern about the lack of experiments. To address this issue, we have added two new experiments, the bowl pick-and-place task and the bottle pick-and-place task. The results are summarized in Table 1 and Table 2 in the experiment section (Section 6) of the revised paper.

---

> > ### Author Response · Authors · 2022-11-19
> > **Response to Reviewer R4yz (2/4)**
> >
> > > - With that number of data (5-10) and smaller than 5% difference, I do not think any conclusion can be made. The author should at least do the same experiments multiple times to provide confidential interval for accuracy.
> >
> > **Response:** We are very sorry for the confusion on this issue. We would like to emphasize that EDFs outperform the baseline (SE(3)-Transporter) with a significant margin for every unseen condition. Please note that we added the default (trained) setup to show that both models are sufficiently trained for the trained setups. However, we find that this may misleadingly give the impression that the performance of EDFs and SE(3)-Transporter have a very small gap. Therefore, we removed the trained setup in the revision. We are very sorry for the confusion.
> >
> > Regarding IDFs, the performance difference between EDFs and IDFs may seem small. However, we would like to remind the reviewer that IDF is also our model. IDFs are ablated versions of EDFs with only type-0 descriptors to verify the effectiveness of higher-type descriptors. We find that placing the ablated model along the baseline gives the wrong impression that IDFs are also a baseline model. Therefore, in the revised paper, we move the comparison with IDFs to the ablation study. In addition, please note that when unseen instances, distractors, and poses are combined, EDFs outperform IDFs by a significant margin (95% vs. 81%, which is more than four times less failure).
> >
> > Lastly, we found that we were not properly controlling the inference time of IDFs, such that IDFs were running with impractically long inference time. Therefore, in the revision, for a fair comparison, we control the inference time of IDFs by limiting the number of query points such that the inference time of IDFs is similar to or only slightly longer than EDFs. We found that with similar inference time, EDFs significantly outperform the ablated model. We list the success rate of both models with inference time in Table 2, which is in the experiment section (Section 6) of the revised paper.
> >
> > > You should provide more background research related to EBM, especially EBM on controlling. Such as [1] and its following paper; [2].
> >
> > **Response:** We thank the reviewer for providing valuable comments to organize our papers better. We added a new paragraph on EBMs in the Background and Related Works section. We have also introduced the important related works on EBMs [1] and [2]. The revised part is as follows.
> >
> > "Energy-Based Models Energy-Based Models (EBMs) are probabilistic models that are derived from energy functions. EBMs are widely used for image generation (Zhu & Mumford, 1998; Xie et al., 2016; Du & Mordatch, 2019), point cloud generation (Xie et al., 2021), and control (Xu et al., 2022; Florence et al., 2022). Due to the intractability of the integral in the denominator of EBMs, Markov chain Monte Carlo (MCMC) methods are commonly used to estimate the gradient of the log-denominator to maximize the log-likelihood (Hinton, 2002; Carreira-Perpinan & Hinton, 2005). The Metropolis-Hastings algorithm (MH) (Hastings, 1970) and the Langevin dynamics (Langevin, 1908; Welling & Teh, 2011) are widely used MCMC methods for EBMs on Euclidean spaces. However, typical Langevin dynamics cannot be used for non-Euclidean manifolds such as the SE(3) manifold. The Langevin dynamics on the SE(3) manifold and general Lie groups are studied by Brockett (1997); Chirikjian (2011); Davidchack et al. (2017). The Langevin dynamics for the general Riemannian manifold and their convergence properties have been studied by Girolami & Calderhead (2011); Gatmiry & Vempala (2022). In this paper, we propose SE(3)-equivariant EBMs on the SE(3) manifold itself. This should be distinguished from SE(3)-equivariant EBMs on Euclidean spaces (Jaini et al., 2021; Wu et al., 2021)."

---

> > > ### Author Response · Authors · 2022-11-19
> > > **Response to Reviewer R4yz (3/4)**
> > >
> > > > Time cost is not provided but EBM must be incredibly slow. Is it correct? Is efficiency a bottleneck for scalability? Both in training and applying.
> > >
> > > **Response:** Our method takes around 13~15 seconds for the pick-and-place inference time of the mug-hanging task. NDFs also take around 10 seconds to infer pick-and-place poses in our system. In addition, there is plenty of room to boost the inference time of EDFs. For example, most of the MCMC time (which is the bottleneck) is spent wandering in the vast void of the workspace. If an object localization/segmentation pipeline is available as NDFs, the MCMC time can be significantly reduced by only proposing near the objects.
> > >
> > > The reason why our method is relatively fast despite that it uses MCMC is that all the outputs of the layers except the last layer are reusable during the MCMC step. We added the explanation in the revised paper. The revised part is as follows.
> > >
> > > "For computational efficiency, only the last layer is used to evaluate the field values at the query points. All the other layers’ outputs only depend on the point cloud and not the query points. Therefore, during the MCMC steps, only the last layer (TFN) has to be recalculated, and the outputs of the other layers (SE(3)-Transformers) can be reused."
> > >
> > > Lastly, scalability is not considered in our work as we focus on problems where data are extremely limited. We believe that there could be another appropriate methods when plenty of data are available.
> > >
> > > > IDF is SE(3)-invar. , NDF is SE(3)-equivar. Why IDC can be understood as an end-to-end trainable version? Appendix E do not have a clear explanation.
> > >
> > > **Response:** We are very sorry for this confusion. We would like to emphasize that the descriptors of NDFs are not SE(3)-equivariant but are SE(3)-invariant. It is very important to understand this point, as one of our major contributions is the effectiveness of using higher-type representation theoretic descriptors. Nevertheless, we found that our discussion in the original manuscript was insufficient to deliver this important point. To address this issue, we have added more explanation in Section 4.1. The revised part is as follows.
> > >
> > > "Note that NDFs (Simeonov et al., 2021) only use type-0 descriptors, which are invariant to rotations such that D(R) = I. In contrast, EDFs also use type-1 or higher descriptors, which are highly sensitive to rotations. As a result, NDFs require at least three non-collinear query points (and much more in practice) to represent the orientation of a rigid body, whereas EDFs require only one point."
> > >
> > > In this context, the descriptors of IDFs and NDFs are both SE(3)-invariant, not SE(3)-equivariant. This is the reason why we claim that IDFs are end-to-end (and bi-equivariant) modifications of NDFs. However, in the revised paper, we decided to just use the term 'ablated model' instead of IDFs, as the name seemingly causes much confusion.
> > >
> > > > If the advantage is well proved in mathematical proof or well-defined experiment, it is important and useful for not only robot manipulation, but many domains.
> > >
> > > **Response:** We thank the reviewer for appreciating the applicability of our method to different domains. We believe that our work would be applicable to many other domains, such as protein docking.

---

> > > > ### Author Response · Authors · 2022-11-19
> > > > **Response to Reviewer R4yz (4/4)**
> > > >
> > > > > Multiple advantages are presented but not validated. It lacks supportive discussions and experiments.
> > > > > No supportive discussions;
> > > >
> > > >
> > > > **Response:** We thank the reviewer for raising concern about the lack of supportive discussions and experiments. To address this issue, in the revised paper, we have added two new experiments, the bowl pick-and-place task and the bottle pick-and-place task. (The results are summarized in Table 1 and Table 2 of Section 6).
> > > >
> > > > To address the lack of supportive discussion, we added more discussion on the benefit of representation-theoretic descriptors in Section 4.2 and the ablation study in Section 6.
> > > >
> > > > "Note that NDFs (Simeonov et al., 2021) only use type-0 descriptors, which are invariant to rotations such that D(R) = I. In contrast, EDFs also use type-1 or higher descriptors, which are highly sensitive to rotations. As a result, NDFs require at least three non-collinear query points (and much more in practice) to represent the orientation of a rigid body, whereas EDFs require only one point."
> > > >
> > > > "The ablation study shows that using higher-type descriptors significantly increases generalization performance when the number of query points is highly limited due to computational constraints. As can be seen in Table 2, EDFs outperform the ablated model that only uses type-0 descriptors as NDFs. As was discussed in Section 4.1, we presume that this is due to the orientational insensitivity of type-0 descriptors. As type-0 descriptors cannot represent orientations alone, query points are crucial in representing orientations for the ablated model. In contrast, the higher-type descriptors of EDFs can represent the orientation without the help of query points. As a result, EDFs can maintain orientational accuracy in unseen situations in which low-quality query points are expected."
> > > >
> > > >
> > > > We really appreciate the reviewer's constructive comments. We look forward to having further discussions.

---

> > ### Comment · Reviewer_R4yz · 2022-12-03
> > **thanks. good.**
> >
> > Thanks for the reply. I have read the new version and found it have been largely improved.
> >
> > As for the background research part, the author have provided new paragraphs related to EBM. Hence, It can be further completed by add earlier pioneering EBM works such as [1][2].
> > [1] Synthesizing Dynamic Pattern by Spatial-Temporal Generative ConvNet [2] Learning Descriptor Networks for 3D Shape Synthesis and Analysis
> >
> > As for mathematic part, I think it is clear enough to explain your work to non-mathematic background. The video largely helped people to understand the experiments.
> >
> > As for the experiment part, the arthor provided additional experiments. It becomes more convincing. I am satisfied with the results. However, I believe there are still a lot of experiments to be done. The strengthen of the result is still not very strong.
> >
> > Sorry for the misunderstand about SE(3)-(in)varient, thanks for the explaination.
> >
> > All in all, I decide to increase the score to 6. The problems of this paper is still about missing very solid experiment and very convincing discussions.

---

> > > ### Author Response · Authors · 2022-12-08
> > > **Response to the Post-rebuttal Comment of Reviewer R4yz**
> > >
> > > We are glad to hear from the reviewer that the revised paper has become clearer and more convincing in both the mathematical and experimental details.
> > > We also thank the reviewer for providing those two pioneering works that are highly relevant to our subject.
> > > The reviewer's suggestion would help improve the completeness of the EBM part of Section 2. We will be adding those works in the next revision (we are currently unable to modify our submission).
> > >
> > > Regarding the experiments, please recall that we **followed the evaluation protocol of the prior work**, Neural Descriptor Fields (NDFs), in which the same three tasks (the mug-hanging and bowl/bottle pick-and-place tasks) were used.
> > > We believe the experiment results in our revised paper are sufficient to support our claim that EDFs can achieve the claimed benefits of NDFs (generalizability to unseen poses and instances with a small number of demonstrations) **even without requiring any prior knowledge or complex object segmentation pipelines**.
> > >
> > > As to the baseline methods, Transporter Networks are the only end-to-end trainable models that work well with such a few demonstrations (at least for planar tasks) without any prior knowledge. Besides the end-to-end trainability, NDFs and Dense Object Descriptors (DONs) cannot be used for problems like ours in which the pose of the placement target (the hanger or tray, in our case) is not fixed. Only recently, [1] was proposed by the same authors of NDFs to overcome this limitation, but this work is not considered a prior work to ours (as it was published last month). The authors of [1] also could not find an existing baseline method that works with few demonstrations, so they used two constructed baselines with naive MLP or point cloud registration method. Please also note that NDFs were compared with only one baseline method (DONs), and Transporter Networks were compared with one state-of-the-art baseline (Form2Fit) and a few constructed MLP baselines. In light of these facts, we believe that our choice of using one state-of-the-art baseline (Transporter Networks) and one constructed baseline (type-0 only) is sufficient to support the effectiveness of our method.
> > >
> > >
> > > Lastly, we would like to remind the reviewer that we have added a supportive discussion as to why using type-1 or higher descriptor is beneficial for generalizability (because of the orientational sensitivity of the higher-type descriptors). We have also discussed why using an energy-based model is essential for end-to-end trainability (because naively minimizing the energy of NDFs does not work ). We have discussed that, unlike NDFs, EDFs only rely on locally equivariant mechanisms and hence do not require object segmentation. We have discussed the necessity of the bi-equivariance property in Appendix C. We believe that all the claimed benefits are heavily supported with solid theoretical arguments and empirical evidence.
> > >
> > > In conclusion, our work neither lacks in the number of experiments nor the number of baseline methods as compared to other accepted works with similar problems. In addition, all the claims in our paper are heavily supported with theoretical and empirical grounds.
> > >
> > > We sincerely thank the reviewer again for thoroughly reviewing our revised paper and providing helpful suggestions to improve the completeness of our paper. We are open to discussing any additional concerns the reviewer may have.
> > >
> > >
> > > [1] A. Simeonov et al., "SE(3)-Equivariant Relational Rearrangement with Neural Descriptor Fields." 6th Conference on Robot Learning (CoRL), 2022.

---

> ### Author Response · Authors · 2022-12-01
> **Video has been uploaded**
>
> Dear reviewer R4yz,
>
> We would like to inform that we have uploaded the video about the experiments that may help the understanding. Please note that we have added two new tasks in the revised manuscript.
>
> The link to the video is as follows: https://youtu.be/w3PA7eLml7s
>
> Again, we appreciate the reviewer for providing valuable comments. We look forward to having further discussions with the reviewer.

---

### Official Review · Reviewer_tB6e · 2022-10-24

**Confidence:** 3
**Correctness:** 2
**Technical Novelty And Significance:** 2
**Empirical Novelty And Significance:** 2
**Recommendation:** 6

**Clarity, Quality, Novelty And Reproducibility:**

Clarity: The author has listed too many less relevant materials on basic concepts about the Group Theory and misses emphasizing its major contributions about the EBM part.

Quality, Novelty: In my view, the major contribution is the author proposed an EBM for the SE(3)-equivalent field descriptor. However, there lacks evidence in the experiments to demonstrate the advantage claimed by the author.

Reproducibility: The author has claimed to submit all the codes to reproduce the results of all the experiments in the paper. If this could be revealed it would be convenient for other researchers to follow the work.

**Strength And Weaknesses:**

This paper proposes a data-efficient model that could learn end-to-end for robotic manipulation from visual demonstrations. Compared with previous SE(3) - equivariant models, the proposed method could learn from only a few demonstrations without pre-training. The authors have proposed an interesting solution to solve the sample efficiency problem, yet there are still concerns about the experiment part to demonstrate the proposed method. Please refer to the below comments for my questions about this paper.

Method:
1. According to the paper, the major contribution is that the authors propose an EBM to model the descriptor field. However, the related content only takes a small part, and most materials in section 2 are about the description of SE(3)-equivalent. The author should emphasize more on the proposed contribution, while the less relevant concept and formulas could be left in citations.

2. For the proposed EDF, there are related works about MCMC sampling [1][2] and relevant learning methods for point clouds, such as [3]. It is essential to introduce and compare these related works since the EDF is the major contribution in this paper.

Except for the proposed method, I also have some concerns about the experiments.

1. The author has only shown the experiment in the task of mug hanging. However, as the author has mentioned about Neural Descriptor Fields (NDF), there are additional manipulation targets used in the simulated environment, i.e., bowl and bottle. Why not the author conduct these experiments to show the generalization of the proposed method?

2. Although the author has claimed that the model could learn without prior segmentation from the background, the experiment is conducted in a very simple environment. This raises concerns if the EDFs (or IDFs) could perform as expected when the environment becomes complicated, such as introducing random distracting objects.

3. Although the author has mentioned that NDF requires pre-training, it is self-supersized in an auto-encoding way that requires no expert labels. Therefore it is of great interest to see the comparison between NDF and the proposed method since both models could learn from a few demonstrations. This should be listed in the experiment section.

4. In the first paragraph in section 6, the author should mention the computation architecture when mentioning the processing time. Additionally, since the author has claimed the learning efficiency of NDF, it is worthwhile to compare the time efficiency in the experiments.


Writing:
1. In the experiment section, the author mix uses different tenses in the experiment settings and descriptions (also found in other sections).
2. In the last paragraph of section 3, Transporter networks ->  Transporter Networks
3. Background related to EBM is missing in section 2.

[1] Zhu, Song Chun, and David Mumford. "Grade: Gibbs reaction and diffusion equations." Sixth International Conference on Computer Vision (IEEE Cat. No. 98CH36271). IEEE, 1998.

[2] Xie, Jianwen, et al. "A theory of generative convnet." International Conference on Machine Learning. PMLR, 2016.

[3] Xie, Jianwen, et al. "Generative pointnet: Deep energy-based learning on unordered point sets for 3d generation, reconstruction and classification." Proceedings of the IEEE/CVF Conference on Computer Vision and Pattern Recognition. 2021.


**Summary Of The Paper:**

This paper proposes an end-to-end visual learning method for 3D robotic manipulation. The authors construct an SE(3)-equivariant energy-based model to learn end-to-end from limited demonstrations without prior knowledge. The proposed Equivariant Descriptor Fields could be generalized to unseen poses, instances, and target objects.

**Summary Of The Review:**

This paper has proposed an EBM to learn end-to-end 3D robotic manipulation from limited visual demonstrations. The major concern from me about the effectiveness of the proposed method is the experiment section, where the author mainly conducts the experiment in a simulation environment and merely in the pick&place task of a mug. Since the author has claimed no pre-segmented points cloud info is required in the proposed method, the author needs to show the generalization on different manipulation objects. Furthermore, in the compared methods, i.e., Transporter Networks (TN) or Neural Descriptor Fields (NDF), the authors conduct the experiments in multiple situations to demonstrate the effectiveness of their proposed methods, such as 10 different manipulation tasks in TN and real-world experiment in NDF. Please refer to the previous comments for my concerns.

---Post Rebuttal----

First, I'd like to move my rating from 3->5 due to the significant revision of the paper and the updated experiments. However, as I am not an expert in the field of robotics, I still have concerns about the experiment part. Although the author has updated the experiments with more objects, I am not sure about the meaning of the comparison result. Are there any other metrics other than success rate and computation time to compare different models? Are they any other methods to compare in the pick-and-place task except for SE(3)-TN?

---

> ### Author Response · Authors · 2022-11-19
> **Response to Reviewer tB6e (1/3)**
>
> First, we would like to thank the reviewer for the helpful advice. The reviewer's comment helped us to clarify our contributions better.
>
>
> > According to the paper, the major contribution is that the authors propose an EBM to model the descriptor field. However, the related content only takes a small part, and most materials in section 2 are about the description of SE(3)-equivalent. The author should emphasize more on the proposed contribution, while the less relevant concept and formulas could be left in citations.
>
> **Response:** We sincerely thank the reviewer for pointing out that we need to address energy-based models in the related works section and not introduce too many less relevant concepts on SE(3)-equivariance. As suggested by the reviewer, we have removed many less relevant details on the representation theory and added a new paragraph on energy-based models in the related works section of the revision. We have moved the introductions on EBMs and MCMC methods to Section 2. Please note that these were originally introduced in Section 5 and Appendix D. The revised part is as follows.
>
> "Energy-Based Models Energy-Based Models (EBMs) are probabilistic models that are derived from energy functions. EBMs are widely used for image generation (Zhu & Mumford, 1998; Xie et al., 2016; Du & Mordatch, 2019), point cloud generation (Xie et al., 2021), and control (Xu et al., 2022; Florence et al., 2022). Due to the intractability of the integral in the denominator of EBMs, Markov chain Monte Carlo (MCMC) methods are commonly used to estimate the gradient of the log-denominator to maximize the log-likelihood (Hinton, 2002; Carreira-Perpinan & Hinton, 2005). The Metropolis-Hastings algorithm (MH) (Hastings, 1970) and the Langevin dynamics (Langevin, 1908; Welling & Teh, 2011) are widely used MCMC methods for EBMs on Euclidean spaces. However, typical Langevin dynamics cannot be used for non-Euclidean manifolds such as the SE(3) manifold. The Langevin dynamics on the SE(3) manifold and general Lie groups are studied by Brockett (1997); Chirikjian (2011); Davidchack et al. (2017). The Langevin dynamics for the general Riemannian manifold and their convergence properties have been studied by Girolami & Calderhead (2011); Gatmiry & Vempala (2022). In this paper, we propose SE(3)-equivariant EBMs on the SE(3) manifold itself. This should be distinguished from SE(3)-equivariant EBMs on Euclidean spaces (Jaini et al., 2021; Wu et al., 2021)."
>
>
> > For the proposed EDF, there are related works about MCMC sampling [1][2] and relevant learning methods for point clouds, such as [3]. It is essential to introduce and compare these related works since the EDF is the major contribution in this paper.
>
> **Response:** We thank the reviewer for commenting on adding those important related works. We do agree that [1],[2], and [3] should be introduced as related works on energy-based models. As was commented by the reviewer, we have cited and introduced these models in Section 2 as related works on energy-based models. The revised part is as follows. "EBMs are widely used for image generation (Zhu & Mumford, 1998; Xie et al., 2016; Du & Mordatch, 2019), point cloud generation (Xie et al., 2021), and control (Xu et al., 2022; Florence et al., 2022)."
>
> We have also explained in the revision why we could not apply energy-based models on *Euclidean spaces* that utilize Langevin dynamics such as [1],[2], and [3] to our problem. The revised part is as follows. "However, typical Langevin dynamics cannot be used for non-Euclidean manifolds such as the SE(3) manifold."
>
> > The author has only shown the experiment in the task of mug hanging. However, as the author has mentioned about Neural Descriptor Fields (NDF), there are additional manipulation targets used in the simulated environment, i.e., bowl and bottle. Why not the author conduct these experiments to show the generalization of the proposed method?
>
> **Response:** We thank the reviewer for raising concern about the lack of experiments. To address this issue, we have added in the revision two new tasks, the bowl pick-and-place task and the bottle pick-and-place task, as was commented by the reviewer. The experiment results are summarized in Table 1 and Table 2 of the experiment section (Section 6). We thank the reviewer for the valuable advice.

---

> > ### Author Response · Authors · 2022-11-19
> > **Response to Reviewer tB6e (2/3)**
> >
> > > Although the author has claimed that the model could learn without prior segmentation from the background, the experiment is conducted in a very simple environment. This raises concerns if the EDFs (or IDFs) could perform as expected when the environment becomes complicated, such as introducing random distracting objects.
> >
> > **Response:** We are sorry for the unclear presentation of our work in the original manuscript.
> > Please note that we have already experimented with multiple distracting objects. The experiment results show that EDFs are robust even though multiple unseen distracting objects are presented with the target objects. To address this issue, we have
> > added a figure of the test environment (Figure 5 of Section 6) in the revision. We are very sorry for the confusion.
> >
> > > Although the author has mentioned that NDF requires pre-training, it is self-supersized in an auto-encoding way that requires no expert labels. Therefore it is of great interest to see the comparison between NDF and the proposed method since both models could learn from a few demonstrations. This should be listed in the experiment section.
> >
> > **Response:**  We appreciate the reviewer for this suggestion. However, NDFs cannot be directly compared to EDFs because they require a well-segmented object point cloud and a fixed target placement pose. We found that we have not addressed the fixed placement pose of NDFs in the original manuscript. We are very sorry for this mistake. The revised parts are as follows.
> >
> > "However, NDFs take the point cloud of the target objects as the only input and do not take the point cloud of the surface where the objects should be placed. Instead, NDFs use fixed query points to represent the placement surface (Simeonov et al., 2021). Therefore, NDFs can only be applied to tasks with a fixed target placement pose. In addition, NDFs cannot be used without object segmentation pipelines (See Appendix J). For these reasons, NDFs cannot be applied to more general manipulation tasks in which target placement poses are not fixed, and well-segmented object inputs cannot be expected."
> >
> > "Note that we do not directly compare EDFs with NDFs because (1) NDFs require the target placement poses to be fixed, and (2) NDFs require object segmentation pipelines."
> >
> > In addition, please recall that we do not claim EDFs to be superior to NDFs in their success rates. Instead, we claim that EDFs do not require many assumptions that NDFs made (pre-training, segmentation, fixed placement poses), yet still be able to achieve few-shot level sample efficiency and generalizability upon unseen instances and poses. Therefore, we only compare with Transporter Networks, which are state-of-the-art end-to-end visual manipulation models that also do not make such assumptions.
> >
> > Lastly, we do agree with the reviewer's comment that the pre-training of NDFs is self-supersized in an auto-encoding way that requires no expert labels. Therefore, we removed the experimentally unsupported claim that "NDFs require an excessive amount of data for the pre-training." We are very sorry for this overstatement. We thank the reviewer for the valuable criticism. The revised part is as follows.
> >
> > "While not fully end-to-end trainable from demonstrations like Transporter Networks, NDFs can be pre-trained using public datasets like Chang et al. (2015), thereby relieving the data collection burden."
> >
> >
> >
> > > In the first paragraph in section 6, the author should mention the computation architecture when mentioning the processing time. Additionally, since the author has claimed the learning efficiency of NDF, it is worthwhile to compare the time efficiency in the experiments.
> >
> > **Response:** We thank the reviewer for commenting on adding computation architecture with the processing time. In the revised paper, we added the following part. "All the times were measured using an Intel i9-12900k CPU (P-core only) with an Nvidia RTX3090 GPU."
> >
> > The inference time of NDFs takes around 10 seconds in our system, which is slightly faster but is not an order of difference to EDFs, which take 13~15 seconds.
> > In addition, there is plenty of room to boost the inference time of EDFs. For example, most of the MCMC time (which is the bottleneck) is spent wandering in the vast void of the workspace. If an object localization/segmentation pipeline is available as NDFs, the MCMC time can be significantly reduced by only proposing near the objects.
> >
> > > - In the experiment section, the author mix uses different tenses in the experiment settings and descriptions (also found in other sections).
> > > - In the last paragraph of section 3, Transporter networks -> Transporter Networks
> >
> > **Response:** We thank the reviewer for pointing out our mistakes in writing. We have rewritten our manuscript to use consistent tense. We also capitalized Transporter Networks in the last paragraph of section 3.

---

> > > ### Author Response · Authors · 2022-11-19
> > > **Response to Reviewer tB6e (3/3)**
> > >
> > > > - Background related to EBM is missing in section 2.
> > > > - Clarity: The author has listed too many less relevant materials on basic concepts about the Group Theory and misses emphasizing its major contributions about the EBM part.
> > >
> > > **Response:** To address this issue, we have made a new paragraph in Section 2 of the revised paper to introduce EBMs and the related works. We have reduced our introduction to the representation theory and spent more pages on EBMs with important related works. We thank the reviewer for helping us to organize our paper better.
> > >
> > > > In my view, the major contribution is the author proposed an EBM for the SE(3)-equivalent field descriptor.
> > >
> > > **Response:** We first thank the reviewer for appreciating our contribution that we propose EBM to end-to-end train the descriptor fields. However, there are three more important contributions that we believe non of which are more or less important than the contribution about EBM. Therefore, in the revision, we have emphasized these three points together with the contribution about EBMs in the introduction section. The revised part is as follows.
> > >
> > >
> > > - We propose a novel energy function to make our models not only SE(3)-equivariant to the object but also to where the object has to be placed. This enables EDFs to solve highly spatial tasks with changing target placement poses efficiently.
> > > - We reformulate the heuristic energy minimization problem of NDFs into a distribution learning problem with energy-based models. This enables EDFs to be end-to-end trained without any pre-training.
> > > - We argue that in the context of the representation theory, the descriptors of NDFs are not equivariant but invariant, which greatly limits the orientational sensitivity of the descriptors. In contrast, EDFs fully exploit the theoretical power of the representation theory by using equivariant descriptors. We show by experiments that using equivariant descriptors greatly improves the generalizability.
> > > - EDFs do not resort to non-local mechanisms to achieve the equivariance. We argue that this specific design allows EDFs to work well without object segmentation pipelines, as opposed to NDFs.
> > >
> > >
> > >
> > > > - There lacks evidence in the experiments to demonstrate the advantage claimed by the author.
> > > > - The major concern from me about the effectiveness of the proposed method is the experiment section, where the author mainly conducts the experiment in a simulation environment and merely in the pick&place task of a mug.
> > > > - Furthermore, in the compared methods, i.e., Transporter Networks (TN) or Neural Descriptor Fields (NDF), the authors conduct the experiments in multiple situations to demonstrate the effectiveness of their proposed methods, such as 10 different manipulation tasks in TN and real-world experiment in NDF.
> > >
> > > **Response:**  We appreciate the helpful suggestions by the reviewer. To address this issue, in the revision, we added two new experiments, the bowl pick-and-place task, and the bottle pick-and-place task, as was commented by the reviewer. The results can be found in Tables 1 and 2 of the experiment section (Section 6).
> > >
> > > > Since the author has claimed no pre-segmented points cloud info is required in the proposed method, the author needs to show the generalization on different manipulation objects.
> > >
> > > **Response:** We are very sorry for not being clear on this issue. About the distractors, we would like to recall that the original manuscript already has experiments on distracting objects. To address this issue, we remind of this fact by adding the new Figure 5 in the experiment section of the revised paper.
> > >
> > > Again, we thank the reviewer for the constructive comments. We look forward to having further discussions.

---

> > > > ### Comment · Reviewer_tB6e · 2022-12-11
> > > > **More discussion about Langevin dynamics**
> > > >
> > > > The current related work in the revised paper looks good to me now. Except for the original Langevin dynamics you mentioned, the amortized sampling [4][5][6] (i.e., learning a top-down generator (VAE or normalizing flow) to initialize the Langevin dynamics) is also a commonly used MCMC method to train EBMs on Euclidean space, and it might also be hard to be used in the SE(3) manifold. You can consider mentioning this. Last but not least, other EBM references that might be useful to improve and complete the EBM paragraph in the related work section include EBM applications for internal learning [7] and conditional learning [8].
> > > >
> > > >
> > > > [4]  A Tale of Two Flows: Cooperative Learning of Langevin Flow and Normalizing Flow Toward Energy-Based Model. ICLR 2022
> > > >
> > > > [5] Learning Energy-Based Model with Variational Auto-Encoder as Amortized Sampler. AAAI 2021
> > > >
> > > > [6]  Cooperative Training of Descriptor and Generator Networks. PAMI 2018
> > > >
> > > > [7] Patchwise Generative ConvNet: Training Energy-Based Models from a Single Natural Image for Internal Learning. CVPR 2021
> > > >
> > > > [8] Cooperative Training of Fast Thinking Initializer and Slow Thinking Solver for Conditional Learning. PAMI 2021.

---

> > > > > ### Author Response · Authors · 2022-12-12
> > > > > **Response to the Post-rebuttal Comment of Reviewer tB6e (2)**
> > > > >
> > > > > We appreciate the reviewer's suggestion to include those related works [4][5][6][7][8] in our paper.
> > > > > These works are highly relevant and interesting, and we will introduce them in the next revision of our paper.
> > > > >
> > > > > We are particularly interested in the amortized sampling methods [4][5][6] mentioned by the reviewer, as our current approach relies on simple Metropolis-Hastings to initialize the samples for Langevin dynamics.
> > > > > With some adaptations to the SE(3) manifold, the mentioned amortized sampling methods would significantly improve the MCMC speed of our approach.
> > > > > Additionally, the multi-scale MCMC approach of [7] and the conditional learning approach of [8] are also intriguing.
> > > > > We believe that these methods could be combined to reduce the training and inference time of our approach by an order of magnitude.
> > > > > Therefore, in the next revision, we will be introducing these works in the related works section, or possibly in the discussion section, as we discussed that "faster sampling methods are required" in the discussion section.
> > > > > We are grateful for the reviewer's introduction to these important and relevant works, and we look forward to incorporating them into our approach.
> > > > > We also believe that these works will be valuable for our future research, as well as for others in the field.
> > > > >
> > > > > We would also like to remind the reviewer that we have answered the questions raised in the post-rebuttal response by the reviewer.
> > > > > It can be found right below the original review. We look forward to addressing any additional concerns the reviewer may have.
> > > > >
> > > > > We sincerely thank the reviewer again for the valuable suggestions.

---

> ### Author Response · Authors · 2022-12-01
> **Response to the Post-rebuttal Comment of Reviewer tB6e**
>
> We sincerely thank the reviewer for acknowledging our significant revision on the experiments.
>
> **[Question 1]** "However, as I am not an expert in the field of robotics, I still have concerns about the experiment part. Although the author has updated the experiments with more objects, I am not sure about the meaning of the comparison result."
>
> **[Response]** We sincerely appreciate the reviewer for raising questions on the experiments from the perspective of an expert from different fields. First, we would like to apologize for not being clear on the experiments.
> To address this issue, we have uploaded a video of the experiments.
> We are very sorry for the confusion. We hope that the video clarifies the meaning of the experiments.
>
> The link to the video is as follows: https://youtu.be/w3PA7eLml7s
>
>
>
> **[Question 2]** "Are they any other methods to compare in the pick-and-place task except for SE(3)-TN?"
>
> **[Response]** There are many methods for pick-and-place tasks, but as far as we know, our method and [1] are the only two
> methods that can learn to solve highly spatial tasks with changing placement poses (like those in the video)
> from only a few (5~10) demonstrations ([1] is the follow-up paper that was
> **published last month** by the authors of NDFs to address a similar problem to ours).
>
> Please note that the authors of [1] also could not find a proper baseline method that
> works well with this small number of demonstrations, so they used constructed baselines with naive
> MLP or point cloud registration method. The authors of [1] state in the paper that:
>
> > As existing rearrangement methods* are not directly applicable with so few demonstrations, we
> compare with two constructed baselines.
>
> (*Please recall that we solved a more general problem than the rearrangement task of [1], as our
> method does not assume explicit objectness or segmentation.)
>
>
> We would also like to emphasize that many recent works [2][3][4] compare their works solely on Transporter Networks as the state-of-the-art method.
> For example, [2] compares with Transporter Networks, Form2Fit, and various MLP approaches. However, Form2Fit and the MLP methods in [2] are already outperformed by the original Transporter Networks paper, thus, are not considered SOTA. In [3], the only robotics model used for baseline is Transporter Networks. Other models are language models, which are not in the scope of our work. [4] also uses Transporter Networks as the only SOTA baseline (the naive MLP models in [4] are not SOTA as Transporter Networks already outperform them).
>
> Please recall that our paper has two baselines: one state-of-the-art baseline (Transporter Networks) and one constructed baseline (the type-0 only model).
> Unlike other works, we did not experiment with naive MLP baselines because it is too trivial that with such a small number (5~10) of demonstrations, they fail not only in the unseen setups but even in the trained setups.
> In our experiments, we only used baselines that can achieve at least 90% success rate for the trained setups.
>
> [1] A. Simeonov et al., "SE(3)-Equivariant Relational Rearrangement with Neural Descriptor Fields." 6th Conference on Robot Learning (CoRL), 2022.
>
> [2] H. Huang et al., "Equivariant Transporter Network." Proceedings of Robotics: Science and Systems (RSS), 2022.
>
> [3] M. Shridhar et al., "CLIPORT: What and Where Pathways for Robotic Manipulation." 5th Conference on Robot Learning (CoRL), 2021.
>
> [4] D. Seita et al., "Learning to Rearrange Deformable Cables, Fabrics, and Bags with Goal-Conditioned Transporter Networks." IEEE International Conference on Robotics and Automation (ICRA), 2021.
>
>
> **[Question 3]** "Are there any other metrics other than success rate and computation time to compare different models?"
>
> **[Response]** Success rate and the number of samples (or the number of episodes in reinforcement learning) are the two most commonly used metrics in robotic manipulations. For methods with non-negligible computation time, the inference speed is also an important metric. In our experiments, we restricted the number of demonstrations to not more than ten. Therefore, we only measured the success rates and inference time.
>
> For other metrics, total discounted reward or time/energy optimality of the trajectory are also often used. However, these are not in the scope of our work (we do not solve reinforcement learning or trajectory-level control problems). For behavior cloning methods, the 'accuracy' may be measured, but it is inappropriate for non-deterministic methods like ours. After all, we want our models to generalize rather than memorize and replicate the demonstrations. For these reasons, the success rate is the appropriate metric to measure the performance of the robotic manipulation models.
>
>
> Lastly, we would like to thank the reviewer again for the helpful suggestions and the valuable questions. We look forward to further discussion with the reviewer.

---

### Official Review · Reviewer_8RUM · 2022-10-25

**Confidence:** 4
**Clarity, Quality, Novelty And Reproducibility:** The paper is clearly written. The pro…
**Correctness:** 4
**Technical Novelty And Significance:** 3
**Empirical Novelty And Significance:** 2
**Recommendation:** 6

**Details Of Ethics Concerns:**

N/A.

**Strength And Weaknesses:**

Strength
* The proposed method is well-motivated theoretically, and its practical instantiation incorporatoins several components to accomendate the innate instability and intractability in the theoretical formulation.
* The paper offers an analysis on the generalization capability and robustness of the proposed method under different unseen conditions.

Weaknesses
* The main concern is that the empirical verification is very limited. The tasks being evaluated are restricted to one bug-hanging task and one stick-to-tray task. It is unclear whether the proposed method will encounter practical problems in general.
* It will be helpful if the paper can provide an analysis on how much demonstration is needed for the method to succeed, and how the learning curve saturates with respect to the amount of demonstration data available.

**Summary Of The Paper:**

This work proposes a SE3-equivariant EBM with high sample efficiency and a competent generalization capability.

**Summary Of The Review:**

This work introduces a SE3 equivariant formulation for robot manipulations that is fully differentiable, and empirically verifies the sample efficiency, generalization capability and robustness of the proposed method. The tasks being evaluated, however, are highly restricted. The work has a nice theoretical guidance and more empirical evidence that it works well in practice would improve the contribution of this work.

--- Post Rebuttal ---

I believe the formulation of this work can inspire further research, and it has empirical support following the evaluation protocol from the literature. I've raised my score accordingly.

---

> ### Author Response · Authors · 2022-11-19
> **Response to Reviewer 8RUM**
>
> We would like to first thank the reviewer for providing valuable comments and suggestions.
>
> > The proposed method is well-motivated theoretically, and its practical instantiation incorporatoins several components to accomendate the innate instability and intractability in the theoretical formulation.
>
> **Response:** We sincerely thank the reviewer for appreciating both the theory and our practical approach implementing our theoretical formulation stably.
>
> > The main concern is that the empirical verification is very limited. The tasks being evaluated are restricted to one bug-hanging task and one stick-to-tray task. It is unclear whether the proposed method will encounter practical problems in general.
>
> **Response:** We fully appreciate the reviewer's concern about the lack of experiments on various tasks. To address this issue, in the revision, we have added two new experiments using bowls and bottles, which are the other two tasks that were used to evaluate the performance of Neural Descriptor Fields. The results can be found in Tables 1 and 2 of the experiment section (Section 6).
>
> > It will be helpful if the paper can provide an analysis on how much demonstration is needed for the method to succeed, and how the learning curve saturates with respect to the amount of demonstration data available.
>
> **Response:** We thank the reviewer for valuable advice on improving the quality of the paper. As was commented, we have added the learning curve for EDFs in Figure 6 of the experiment section  (Section 6).
>
> > The paper is clearly written. The proposed theoretical formulation is novel.
>
> **Response:** We greatly thank the reviewer for appreciating the novelty of our formulation.
>
> > This work introduces a SE3 equivariant formulation for robot manipulations that is fully differentiable, and empirically verifies the sample efficiency, generalization capability and robustness of the proposed method. The tasks being evaluated, however, are highly restricted. The work has a nice theoretical guidance and more empirical evidence that it works well in practice would improve the contribution of this work.
>
> **Response:** We thank the reviewer for raising concern about the limited number of experimented tasks. As we have stated above, to address this issue, we have added two new experiments, the bowl pick-and-place task and bottle pick-and-place task. The results can be found in Tables 1 and 2 of the experiment section (Section 6).
>
> We sincerely thank the reviewer again for the helpful advice on our work. We are looking forward to having further discussions.

---

> > ### Comment · Reviewer_8RUM · 2022-11-30
> > **Thank you for your responses**
> >
> > Dear authors,
> >
> > Thank you for your responses. The additional experiments addressed my concerns, and I would like to acknowledge the technical contribution of this work compared to prior arts, with empirical evidence from the evaluation of similar tasks. I've raised my score correspondingly.

---

> > > ### Author Response · Authors · 2022-12-01
> > > **Response to the Post-rebuttal Comment of Reviewer 8RUM**
> > >
> > > We sincerely appreciate the reviewer for acknowledging our technical contribution.
> > >
> > > We are also glad to hear from the reviewer that our claims are now empirically supported in the revised paper.
> > >
> > > We would also like to inform the reviewer that we have uploaded the video of the experiments. The link to the video is as follows: https://youtu.be/w3PA7eLml7s
> > >
> > > We thank the reviewer again for appreciating our technical contribution and its potential impact on future works.

---

### Official Review · Reviewer_y6Vo · 2022-10-31

**Confidence:** 4
**Correctness:** 3
**Technical Novelty And Significance:** 3
**Empirical Novelty And Significance:** 3
**Recommendation:** 6

**Clarity, Quality, Novelty And Reproducibility:**

The paper is very densely written, and formalizes the problem with representation theory of the Lie group. A lot of necessary implementation and experimentation details are in the appendix only. It would be helpful to state the key result and ground the claims of the paper with more experimentation.

**Strength And Weaknesses:**

- The paper deals with an important problem in robot manipulation. Learning equivariant and background invariant policies is key to visual robotic manipulation. Some recent work has addressed this problem empirically, but formal treatment of the problem has been widely neglected.

- The work builds upon Neural Descriptor Fields (NDFs) work in Simeonov et al., 2021. The authors claim that point cloud input needs to be segmented from the background in NDFs, and requires pre-training on an excessive dataset of objects. But do not validate these claims with grasping in clutter and/or multiple objects or challenging objects.

- Comparison with Dense object descriptors and grasping methods for visual robotic planning methods would further improve the quality of the paper. At this point, it is hard to comment on the efficacy of the approach in comparison to the state-of-the-art in visual robotic manipulation.

- Grasping in clutter and/or challenging objects would further develop the limitation of the presented sampling methods.


**Summary Of The Paper:**

This paper presents SE(3) equivariant energy-based models for sample-efficient visual robotic manipulation from point clouds. It also provides theoretical conditions for bi-equivariant energy-based models, along with sampling strategies over the SE(3) manifold. Experiments on simple simulated pick-place and stick-in-tray tasks suggest the feasibility of  generalization to unseen target poses and unseen target object instances of the same category.

**Summary Of The Review:**

SE(3) equivariant energy-based models are an important direction of research for robot manipulation. The paper adequately formalizes the problem. Initial results are promising, and would need more experiments to ground the claims of the paper and establish its usability for visual robotic manipulation.

---

> ### Author Response · Authors · 2022-11-19
> **Response to Reviewer y6Vo (1/2)**
>
> First, we would like to thank the reviewer for the helpful suggestions as to improving the quality of the paper.
>
> > This paper presents SE(3) equivariant energy-based models for sample-efficient visual robotic manipulation from point clouds. It also provides theoretical conditions for bi-equivariant energy-based models, along with sampling strategies over the SE(3) manifold. Experiments on simple simulated pick-place and stick-in-tray tasks suggest the feasibility of generalization to unseen target poses and unseen target object instances of the same category.
>
> **Response:** We sincerely thank the reviewer for acknowledging the theoretical and empirical value of our bi-equivariant energy-based models.
>
> > The paper deals with an important problem in robot manipulation. Learning equivariant and background invariant policies is key to visual robotic manipulation. Some recent work has addressed this problem empirically, but formal treatment of the problem has been widely neglected.
>
> **Response:** We fully agree with the reviewer that formal treatment is widely neglected in recent works on equivariant models. Our work cannot be free from this criticism. However, we believe that we have at least fully formalized our problems into learning probability distribution on the SE(3) manifold, whereas many other equivariant methods rely on heuristic approaches.
>
> > - The work builds upon Neural Descriptor Fields (NDFs) work in Simeonov et al., 2021. The authors claim that point cloud input needs to be segmented from the background in NDFs, and requires pre-training on an excessive dataset of objects. But do not validate these claims with grasping in clutter and/or multiple objects or challenging objects.
> > - Grasping in clutter and/or challenging objects would further develop the limitation of the presented sampling methods.
>
> **Response:** We are sorry for the unclear presentation of our work in the original manuscript. Please note that we have already experimented with multiple distracting objects in the original manuscript. The experiment results show that EDFs are robust even though multiple unseen distracting objects are presented with the target objects. To address this issue, we have added a figure of the test environment (Figure 5 of Section 6) in the revision to remind that we have experiments on multiple distracting objects. We are very sorry for the confusion.
>
> Regarding the claims on NDFs, we do agree with the reviewer that they were not properly grounded in the previous version. Therefore, in the revision, we have added experiments on NDFs with unsegmented point clouds in Appendix J to ground our claim that inputs of NDFs must be segmented. Unfortunately, we could not experiment NDFs with more challenging caterogy of objects due to the limited time. Therefore, we remove the unsupported claim that "NDFs require pre-training on an excessive dataset of objects." We are very sorry for this overstatement. We thank the reviewer for the valuable criticism.
>
> > Comparison with Dense object descriptors and grasping methods for visual robotic planning methods would further improve the quality of the paper. At this point, it is hard to comment on the efficacy of the approach in comparison to the state-of-the-art in visual robotic manipulation.
>
> **Response:** We appreciate the reviewer for raising concern about the comparison with state-of-the-art methods.
> First, we did not compare with Dense Object Descriptors (DONs) because they cannot be end-to-end trained from demonstrations and involve a significant data collection burden.
> Instead, we compare our method with Transporter Networks as state-of-the-art methods for visual manipulation that can be end-to-end trained from only demonstrations.
> We rewrote the paper to deliver this point clearly.
> The revised part is as follows.
>
> "For baselines, we use SE(3)-Transporter Networks (SE(3)-TNs) as the state-of-the-art method for end-to-end visual manipulation."

---

> > ### Author Response · Authors · 2022-11-19
> > **Response to Reviewer y6Vo (2/2)**
> >
> > > A lot of necessary implementation and experimentation details are in the appendix only. It would be helpful to state the key result and ground the claims of the paper with more experimentation.
> >
> > **Response:** We would like to recall that many necessary implementation and experimentation details had to be put into the Appendix due to the limitation of pages. As to the lack of experiments, we have added two more experiments using bottles and bowls as NDFs in the experiment section. We have also clarified the benefit of representation-theoretic descriptors. The revised part is as follows.
> >
> > "The ablation study shows that using higher-type descriptors significantly increases generalization performance when the number of query points is highly limited due to computational constraints. As type-0 descriptors cannot represent orientations alone, query points are crucial in representing orientations for the ablated model. In contrast, the higher-type descriptors of EDFs can represent the orientation without the help of query points. As a result, EDFs can maintain orientational accuracy in unseen situations in which low-quality query points are expected."
> >
> > > SE(3) equivariant energy-based models are an important direction of research for robot manipulation. The paper adequately formalizes the problem. Initial results are promising, and would need more experiments to ground the claims of the paper and establish its usability for visual robotic manipulation.
> >
> > **Response:** We sincerely thank the reviewer for understanding the importance of our research direction and appreciating our formulation of the problem. To address the lack of experiments in the original manuscript, we have added two new experiments using bottles and bowls in the revision to verify our claims better. Please also note that in the previous version, we have already experimented with multiple distracting objects that were not seen during the training stages. This has been reminded by the new Figure 5 in the revised paper.
> >
> >
> > Again, we sincerly thank the reviewer for the helpful comments. We are looking forward to getting further comments and feedbacks.

---

> > > ### Comment · Reviewer_y6Vo · 2022-11-30
> > > **Response to Authors**
> > >
> > > Thank you for clarifying your responses. The paper is adequately grounded in literature, and more experimental validation will help improve the quality of the work.

---

> > > > ### Author Response · Authors · 2022-12-01
> > > > **Response to the Post-rebuttal Comment of Reviewer y6Vo**
> > > >
> > > > We sincerely thank the reviewer for the comment on our revised manuscript. We are glad to hear from the reviewer that the revised paper is now adequately grounded in literature. We also appreciate the reviewer's suggestion that more experimental validation would improve the quality of our work.
> > > >
> > > > However, we would like to highlight that we have already included a thorough set of experiments in our revised manuscript, which we believe are sufficient to support all of our claims.
> > > > These experiments were carefully designed and conducted to test the key aspects of our approach, and we believe they provide strong evidence for the effectiveness of our proposed method.
> > > > Specifically, we examined the following claims:
> > > >
> > > > 1. EDFs (our method) can be end-to-end trained from only a few demonstrations without any prior knowledge to solve highly spatial pick-and-place tasks with previously unseen target object poses and instances.
> > > > 2. EDFs do not require complex object segmentation pipelines and yet are robust to the presence of previously unseen distracting objects.
> > > > 3. The proposed benefits of EDFs (1 and 2) are comparable to the current state-of-the-art method.
> > > > 4. The higher-type descriptors used in EDFs greatly improve the generalization capability.
> > > >
> > > > As pointed out by reviewer 8RUM, **our revised experiments follow the same evaluation protocol of Neural Descriptor Fields** (NDFs), in which the same three tasks (mug-hanging and bowl/bottle pick-and-place tasks with unseen instances and arbitrary poses) were used to evaluate the performance.
> > > > We believe that the results of these highly non-trivial experiments provide sufficient evidence to support Claim 1.
> > > > Furthermore, we have added unseen distracting objects in the experiments to verify Claim 2.
> > > > We have also verified Claim 3 by **comparing ours with the state-of-the-art method** (SE(3) Transporter Networks) with the same protocol.
> > > > Lastly, we have validated Claim 4 by comparing our method with the ablated (Type-0 only) models.
> > > >
> > > > For these reasons, we believe our revised manuscript provides strong experimental evidence to support all of our claims.
> > > > We appreciate the reviewer's valuable feedback and hope that our response addresses the reviewer's concerns.
> > > > We are open to discussing any additional concerns the reviewer may have.
> > > >
> > > > Lastly, we would like to inform the reviewer that we have uploaded the video of the experiments.
> > > >
> > > > The link to the video is as follows: https://youtu.be/w3PA7eLml7s

---

### Official Review · Reviewer_W127 · 2022-11-04

**Confidence:** 3
**Correctness:** 3
**Technical Novelty And Significance:** 3
**Empirical Novelty And Significance:** 3
**Recommendation:** 8

**Clarity, Quality, Novelty And Reproducibility:**

For the paper writing, the relationship between the bi-equivariant EBM and the bi-equivariant energy function is not so clear to me. For novelty, many previous works also explored the same SE(3)-equivariance, and many formulations are similar to previous work, e.g. NDF. The code is attached with well-written documentation, so I believe that it is easy to reproduce.

**Strength And Weaknesses:**

This paper is trying to solve a very important problem: how to encode the SE-(3) equivariance property into the network as a powerful inductive bias. It is an important technique for many 3D vision and robotics applications. Although the TransporterNetwork shows that such kind of network design is helpful for manipulation tasks, it still relies on the translational invariance of CNN but does not explore this problem in a more general SE(3) space and does not deal with the hard rotation part. The authors leverage the representation theory of third-order Lie groups to define the bi-equivariant energy using the probability distribution on SE(3). The formulated equivariant descriptor field can then be used in a key-query system, as previous NDF. I feel that the energy-based model in 4.2 makes much sense. But the implementation choice of using Dirac delta functions to shape the query density makes this design not so useful in many applications where continuous parameterization is more preferred.

One weakness of this paper is the gradient of EBM, which requires a special sampling strategy described in Section 5. It seems that this design makes the overall complexity of the method very high in applications and leads to a very long inference time. As the author discussed in the final section, this method cannot solve problems at the trajectory level.

**Summary Of The Paper:**

In this paper, the author explores the direction of how to utilize SE(3)-equivariance as a novel inductive bias to facilitate sample efficiency. Specifically, it focuses on the same robotic manipulation task as previous work, Neural Descriptor Field. The authors derive an energy function based on the representation of the Lie group, which can be used to characterize the distance/discrepancy between SE(3) pose in the demonstration and the current state. By approximating the gradient of the function with sampling, the SE(3)-equivariance demonstration following problems are formulated as an energy minimization problem. Experiments on the object placement task show that the proposed method outperforms the modified-Transporter Network and Invariant Descriptor Fields method.


**Summary Of The Review:**

I feel that the problem solved in this paper is well-motivated, and the problem formulation looks good to me. However, the implementation choice can be improved to better support the theoretical power of EBM. The difficulty level of the experiments is on-par with previous work in this field. So I recommend a weak acceptance for this paper but the rating may be lower or higher after the rebuttal phase.

---

> ### Author Response · Authors · 2022-11-19
> **Response to Reviewer W127 (1/3)**
>
> We thank the reviewer for the careful reading and the analysis of the technicalities.
>
> > In this paper, the author explores the direction of how to utilize SE(3)-equivariance as a novel inductive bias to facilitate sample efficiency. Specifically, it focuses on the same robotic manipulation task as previous work, Neural Descriptor Field. The authors derive an energy function based on the representation of the Lie group, which can be used to characterize the distance/discrepancy between SE(3) pose in the demonstration and the current state. By approximating the gradient of the function with sampling, the SE(3)-equivariance demonstration following problems are formulated as an energy minimization problem. Experiments on the object placement task show that the proposed method outperforms the modified-Transporter Network and Invariant Descriptor Fields method.
>
> **Response:** We appreciate the reviewer for the summary of our paper. However, we find that some aspects of our work have been wrongly conveyed due to fact that our writing was not clear on it. We are very sorry for the confusion. Therefore, we revised our writing to clarify the following aspects better.
>
> 1\. The reviewer summarizes that we formulate our problem as an energy minimization problem. However, we formulate our problem as a probability distribution learning problem by using the energy-based model (EBM) approach. Naively minimizing the energy function cannot be used to train the descriptors, as it will result in all the descriptors collapsing to zero. This is why EBMs should be used over heuristic energy minimization for end-to-end learning. We found that we have not addressed this point in the previous version. We are very sorry for not being clear on this issue. To address this issue, we rewrote Section 4.2 to include this aspect. The revised part is as follows.
>
> "Naively minimizing the energy function like Simeonov et al. (2021) cannot be used to simultaneously train the descriptors, as this would result in all the descriptors collapsing to zero (or someother constants). Therefore, we use the EBM approach for the end-to-end training of descriptors."
>
> 2\. The reviewer summarizes that we solve the same problem as previous work, Neural Descriptor Fields (NDFs). However, our problem differs from the NDFs' in that (1) the target placement poses are not fixed, and (2) no object segmentation is available. This problem setup is exactly the same as Transporter Networks, but with more spatial tasks. We found that we did not address (1) in the previous version. We are very sorry for this mistake. In the revision, we have made these points clear in the introduction and experiment sections. The revised part is as follows.
>
> "However, NDFs take the point cloud of the target objects as the only input and do not take the point cloud of the surface where the objects should be placed. Instead, NDFs use fixed query points to represent the placement surface (Simeonov et al., 2021). Therefore, NDFs can only be applied to tasks with a fixed target placement pose. In addition, NDFs cannot be used without object segmentation pipelines (See Appendix J). For these reasons, NDFs cannot be applied to more general manipulation tasks in which target placement poses are not fixed, and well-segmented object inputs cannot be expected."
>
> "EDFs can be fully end-to-end trained to solve highly spatial manipulation tasks with changing target placement poses from only a few demonstrations (5∼10 demonstrations are enough)."
>
> "Note that we do not directly compare EDFs with NDFs because (1) NDFs require the target placement poses to be fixed, and (2) NDFs require object segmentation pipelines."
>
>
> > This paper is trying to solve a very important problem: how to encode the SE-(3) equivariance property into the network as a powerful inductive bias. It is an important technique for many 3D vision and robotics applications. Although the TransporterNetwork shows that such kind of network design is helpful for manipulation tasks, it still relies on the translational invariance of CNN but does not explore this problem in a more general SE(3) space and does not deal with the hard rotation part. The authors leverage the representation theory of third-order Lie groups to define the bi-equivariant energy using the probability distribution on SE(3). The formulated equivariant descriptor field can then be used in a key-query system, as previous NDF. I feel that the energy-based model in 4.2 makes much sense.
>
> **Response:** We thank the reviewer for nicely summarizing the strengthes or our works in comparison to Transporter Networks.

---

> > ### Author Response · Authors · 2022-11-19
> > **Response to Reviewer W127 (2/3)**
> >
> > > But the implementation choice of using Dirac delta functions to shape the query density makes this design not so useful in many applications where continuous parameterization is more preferred.
> >
> > **Response:** We thank the reviewer for the careful analysis of the weakness of our implementation choice. The reviewer claims that our query density model is not continuously parameterized and thus may not be useful in other applications. However, our implementation is also continuously parameterized and differentiable. First, the weight and the center of each Dirac delta are continuous variables. In addition, the weight field, which bestows weight on each query point, is a continuously parametrized neural field. Lastly, the query points' positions (the centers of Dirac deltas) are drawn from the weight field using deterministic Stein gradient descent method, which is fully differentiable. However, this point may have not been delivered clearly. We are sorry for not being clear on this aspect. To address this issue, we added the following line in the revision.
> >
> > "In Appendix B, we provide practical implementations of $\mathbf{q}\_{i;\theta}(Y)$ and $w\_{\theta}(\mathbf{x}|Y)$ that are continuously parameterized and differentiable."
> >
> > Nonetheless, we fully appreciate the concern that the weighted sum of Dirac deltas may not be useful in some other domains. However, that is why we formulate our energy function using more general query density instead of more concrete query point models. Our choice of using Dirac deltas is purely to make the integral tractable. Therefore, future works may come up with other densities whose integrals are tractable. Lastly, we believe that our query model is already a huge advance from NDFs' fixed query points in that ours are trainable and grasp-dependent. We have added a mathematical proof in the revision that the grasp-dependence of the query model is essential in achieving the bi-equivariance. The revised part is as follows.
> >
> > "Proposition 4. Non-constant query densities that satisfy Equation (11) must be grasp-dependent."
> >
> > > One weakness of this paper is the gradient of EBM, which requires a special sampling strategy described in Section 5. It seems that this design makes the overall complexity of the method very high in applications and leads to a very long inference time. As the author discussed in the final section, this method cannot solve problems at the trajectory level.
> >
> > **Response:** We appreciate that the reviewer raised concern as to the inference time. However, our approach takes 5.1 sec for the pick and 8.3 sec for the place in the mug-hanging task. Although it is not real-time, it is within a reasonable time range. Note that NDFs also require a similar amount of inference time. NDFs take around 10sec to infer pick-and-place poses in our device. This is only slightly faster than EDFs. Lastly, we believe that there is plenty of room to boost the inference time of EDFs. For example, most of the MCMC time (which is the bottleneck) is spent wandering in the vast void of the workspace. If an object localization/segmentation pipeline is available as NDFs, the MCMC time can be significantly reduced by only proposing near the objects.
> >
> > > For the paper writing, the relationship between the bi-equivariant EBM and the bi-equivariant energy function is not so clear to me.
> >
> > **Response:** We are very sorry for the confusion. As we have stated above, we rewrote Section 4.2 to clarify why EBM should be used over energy minimization for end-to-end training. The revised part is as follows. "Naively minimizing the energy function like Simeonov et al. (2021) cannot be used to simultaneously train the descriptors, as this would result in all the descriptors collapsing to zero (or some
> > other constants). Therefore, we use the EBM approach for the end-to-end training of descriptors."

---

> > > ### Author Response · Authors · 2022-11-19
> > > **Response to Reviewer W127 (3/3)**
> > >
> > > > For novelty, many previous works also explored the same SE(3)-equivariance, and many formulations are similar to previous work, e.g. NDF.
> > >
> > > **Response:** Our work is significantly different from NDFs in numerous aspects, but it was not clearly conveyed in the previous version. To address this issue, we clarified our contributions as a list in the revised introduction. The revised part is as follows.
> > >
> > > - We propose a novel energy function to make our models not only SE(3)-equivariant to the object but also to where the object has to be placed. This enables EDFs to solve highly spatial tasks with changing target placement poses efficiently.
> > > - We reformulate the heuristic energy minimization problem of NDFs into a distribution learning problem with energy-based models. This enables EDFs to be end-to-end trained without any pre-training.
> > > - We argue that in the context of the representation theory, the descriptors of NDFs are not equivariant but invariant, which greatly limits the orientational sensitivity of the descriptors. In contrast, EDFs fully exploit the theoretical power of the representation theory by using equivariant descriptors. We show by experiments that using equivariant descriptors greatly improves the generalizability.
> > > - EDFs do not resort to non-local mechanisms to achieve the equivariance. We argue that this specific design allows EDFs to work well without object segmentation pipelines, as opposed to NDFs.
> > >
> > >
> > >
> > > > I feel that the problem solved in this paper is well-motivated, and the problem formulation looks good to me. However, the implementation choice can be improved to better support the theoretical power of EBM. The difficulty level of the experiments is on-par with previous work in this field.
> > >
> > > **Response:** We thank the reviewer for acknowledging the importance of our problem. We also thank the reviewer for understanding the difficulty of experiments in this field. Please note that we have added two more experiments using bottles and bowls in the revised version. Regarding the implementation choice, we truly believe that our grasp-dependent and trainable implementation is already a considerable advance from the fixed random query points of NDFs. In addition, as we explained before, the choice was inevitable to make the integral tractable.
> > >
> > >
> > > Lastly, we would like to sincerly thank the reviewer again for the helpful discussion. We are looking forward to further comments and discussions.

---

> > > > ### Comment · Reviewer_W127 · 2022-11-29
> > > > **Reply to Author Response**
> > > >
> > > > Thanks for the detailed response to my original review. I am sorry that I may misunderstand some of the concepts for the previous review. Details are more clear to me now. I am happy with the current version and would like to raise my rating.
> > > >
> > > > After taking a look of other comments and response for other reviewers, I agree with other criticism that the experimental settings are naive but I still believe that the author is solving an challenging problem theoretically so that a naive experiment setting is acceptable, even it is not very useful for many real robot applications.

---

> > > > > ### Author Response · Authors · 2022-12-01
> > > > > **Response to the Post-rebuttal Comment of Reviewer W127**
> > > > >
> > > > > We sincerely thank the reviewer for appreciating the theoretical value of our work.
> > > > > We are glad to hear that the details have become clearer after our revision.
> > > > > The valuable analysis and questions on technicalities by the reviewer helped us to deliver these details more clearly in the revised paper.
> > > > >
> > > > > Regarding the experiments, we agree with the other reviewers that the experiments in our original paper were naive. However, we would like to remind the reviewer that **we have addressed this issue by adding much more experiments in the revised paper**.  Especially, our revised experiments **follow the evaluation protocol of the prior work**, Neural Descriptor Fields (NDFs).
> > > > >
> > > > > Lastly, we would like to inform the reviewer that the video of the experiments has been uploaded.
> > > > > The link to the video is as follows: https://youtu.be/w3PA7eLml7s
> > > > >
> > > > > We thank the reviewer once again for the tremendous efforts to fully understand the formulations of our paper in detail.

---

### Decision · Program_Chairs · 2023-01-20

**Decision:**

Accept: poster

**Justification For Why Not Higher Score:**

The main drawback of the paper is the lack of justification in many real robot applications, but it doesn't affect the acceptance decision.

**Justification For Why Not Lower Score:**

All five reviewers agree on accepting the paper because of the novel idea, sufficient empirical results, and valuable theoretical contribution in the paper.

**Metareview: Summary, Strengths And Weaknesses:**

This paper uses the representation theory of the Lie group to construct a novel SE(3)-equivariant energy-based model for improving sample efficiency in learning robotic manipulation. The proposed method can learn the manipulation from a few visual demonstrations without pre-training.  Experiments are conducted to verify the effectiveness of the models, including sample efficiency and generalizability. The paper studies an important problem in learning for robotic manipulation, both theoretical analysis and experimental results are sufficient to justify the idea and demonstrate the performance. During the rebuttal, the authors improved the paper by adding extra supportive experiments and revising the paper according to the suggestions provided by the reviewers. Major concerns have been addressed and all five reviewers lean to accept the paper, with one championing it. After reading all the comments and rebuttals and having an internal discussion with the reviewers, the AC agrees with the reviewers and thinks that the current paper presents a solid research work that is worthy for publication, thus recommending an acceptance. The AC urges the authors to improve their final paper by taking into account all the suggestions from the reviewers.


**Note From Pc:**

if the above contains the word "oral" or "spotlight" please see: "oral" presentation means -> notable-top-5% and "spotlight" means -> notable-top-25%. As stated in our emails, we are disassociating presentation type from AC recommendations